# DiffSampling: Enhancing Diversity and Accuracy in Neural Text Generation

**Giorgio Franceschelli**  *giorgio.franceschelli@unibo.it*
*Alma Mater Studiorum Università di Bologna, Bologna, Italy*

**Mirco Musolesi**  *m.musolesi@ucl.ac.uk*
*University College London, London, United Kingdom*
*Alma Mater Studiorum Università di Bologna, Bologna, Italy*

**Reviewed on OpenReview:** *https://openreview.net/forum?id=kXjHbMvdIi*

## Abstract

Despite their growing capabilities, language models still frequently reproduce content from their training data, generate repetitive text, and favor common grammatical patterns and vocabulary. A possible cause is the decoding strategy: the most common strategies either consider only the most probable tokens, which reduces output diversity, or increase the likelihood of unlikely tokens, compromising output accuracy and correctness. In this paper, we propose *DiffSampling*, a new decoding method that leverages a mathematical analysis of the token probability distribution to ensure the generation of contextually appropriate text. In particular, the difference between consecutive, sorted probabilities can be used to truncate incorrect tokens. In addition, we also propose two variations of the proposed method that aim to correct the subtle inconsistencies of common sampling strategies. Experiments involving four different text-generation tasks demonstrate that our approach consistently performs at least on par with the existing methods it builds upon in terms of quality, despite sampling from a larger set of tokens.

## 1 Introduction

In recent years, large language models (LLMs) have demonstrated remarkable performance (Bubeck et al., 2023), driven by the availability of large-scale datasets, advances in computational power (Bommasani et al., 2021), and the development of innovative learning strategies (e.g., Stiennon et al., 2020; Rafailov et al., 2023). While training provides LLMs with the information and skills required to process natural language, another aspect plays a key role at generation time: the decoding strategy, that is, the method used to extract text sequences from the model. The choice of decoding scheme significantly impacts the generated output, as there is a pronounced trade-off between quality and diversity (Ippolito et al., 2019). The most straightforward strategies, such as greedy decoding (selecting the highest-probability token) or sampling, tend to repeat the same tokens multiple times (Su et al., 2022), reproduce training data (Carlini et al., 2021), or flatten the lexicon in favor of the most common

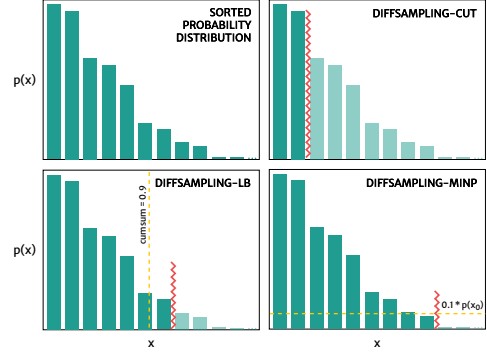

Figure 1: In the top-left square, the original distribution. In the top-right square, *DiffSampling-cut* truncates after the minimum discrete derivative. In the bottom-left square, *DiffSampling-lb* also imposes a total probability lower bound $p_{lb} = 0.9$. In the bottom-right square, *DiffSampling-minp* applies truncation only among tokens with a probability less than $p_{min} = 0.1$ times the highest probability.

grammatical structures and words (Fleisig et al., 2024; Reviriego et al., 2024). Although the temperature parameter may increase the likelihood of less frequent tokens, it also raises the chance of syntactically incorrect ones by flattening their probabilities, regardless of their actual ranking. An ideal solution should concentrate on where the *critical mass* of the probability distribution resides. More precisely, with critical mass, we refer here to the portion of the distribution that collectively accounts for the majority of the probability mass of the tokens. In this direction, a common approach is nucleus sampling (Holtzman et al., 2020), which removes the tail of the distribution by focusing on the smallest subset of tokens whose global probability exceeds a given threshold. However, a key issue remains: it can either preserve incorrect tokens or exclude appropriate ones, depending on whether the critical mass is smaller or larger than the threshold, respectively. As suggested by Hewitt et al. (2022), the learned probability distribution can be viewed as a mixture of the true distribution, which assigns a non-zero probability only to appropriate tokens (the critical mass), and a smoothing distribution, which assigns a small but non-zero probability to incorrect tokens for learning purposes.

To address the subtle inconsistencies in existing truncation strategies, we introduce a family of decoding strategies called *DiffSampling*, based on the analysis of the probability distribution of the tokens, and in particular, on the minimum discrete derivative (i.e., the largest difference between consecutive probabilities in a sorted distribution). We propose a method for excluding incorrect tokens introduced by the smoothing distribution, along with two relaxed variants designed to promote output diversity by *correcting* standard methods (see Figure 1). We then provide a comprehensive evaluation of them under four different tasks[1], namely mathematical problem-solving tasks, extreme summarization, the divergent association task[2], and story generation against the most common baselines, and discuss their advantages and limitations. We show that *DiffSampling* consistently performs at least on par with the standard methods they aim to correct, while enhancing output diversity, especially in longer-form text generation.

## 2 Background

### 2.1 Language Modeling

An autoregressive language model (LM) is a probability distribution $p_{\boldsymbol{\theta}}(\mathbf{x})$ parameterized by $\boldsymbol{\theta}$ over a variable-length text sequence $\mathbf{x} = (x_1 \ldots x_T)$, where $T$ is the sequence length and each token $x_t$ is in a finite vocabulary $\mathcal{V}$. The probability distribution is factorized as $p_{\boldsymbol{\theta}}(\mathbf{x}) = \prod_{t=1}^{T} p_{\boldsymbol{\theta}}(x_t|x_1 \ldots x_{t-1})$, and the LM is usually trained to maximize the likelihood of the true distribution $p_{\star}(\mathbf{x})$ for any $\mathbf{x}$ from a reference dataset (the training set). In other words, given as input $x_1 \ldots x_t$, the model learns to approximate the probability of each token from $\mathcal{V}$ being $x_{t+1}$. While this makes the model immediately capable of scoring the probability of a given text, it also allows for the generation of new sentences. Given a commonly human-written prefix (also known as a prompt) $\mathbf{x} = (x_1 \ldots x_P)$ of length $P$, we can decode a continuation $\hat{\mathbf{x}} = (x_{P+1} \ldots x_{T+P})$ from the LM through its factorized representation introduced before. However, we must remember that the model is trained to score and not to generate sentences. A given text might have zero probability for generation purposes (e.g., the text is syntactically incorrect), but non-zero probability for ranking purposes (Hewitt et al., 2022).

### 2.2 Decoding Strategies

The decoding of tokens from the probability distribution learned by a neural language model can occur in several ways. The greedy strategy involves selecting the most probable token each time. However, this can lead to a consistent lack of diversity and several repetitions. The standard approach involves sampling from the probability distribution, which can be transformed through a *temperature* parameter $\tau$. The temperature scales the differences among the various probabilities: a temperature lower than 1 will increase the probability of the most-probable tokens (a zero temperature degenerates to greedy strategy), while a temperature higher than 1 will increase the probability of the least-probable tokens, allowing for more diversity in generation

---

[1]The code and results are available at: `https://github.com/giorgiofranceschelli/DiffSampling-tmlr`

[2]A common task in creativity research that evaluates the ability to generate semantically unrelated concepts (Olson et al., 2021).

(Peeperkorn et al., 2024). However, this might lead to the selection of tokens that are not syntactically appropriate for the current input. Selective sampling (Troshin et al., 2025) dynamically switches between greedy and high-temperature sampling based on the likelihood of output errors estimated by a lightweight, ad-hoc classifier. Alternatively, top-$k$ sampling (Fan et al., 2018) reduces the token space to the $k$ most probable ones. To generate more natural and coherent solutions, contrastive search (Su et al., 2022) employs a greedy strategy over the combination of a top-$k$ truncation and a degeneration penalty. This promotes the selection of tokens that differ from those already generated, enhancing the diversity and quality of the output. Nevertheless, limiting the number of selected tokens *a priori* can lead to the exclusion of meaningful tokens or the inclusion of inappropriate ones. A possible solution is to set $k$ dynamically, as in Mirostat (Basu et al., 2021): to maintain the perplexity of generated text at a desired value, the $k$ parameter is actively tuned based on the current cross-entropy.

Alternatively, nucleus (or top-$p$) sampling (Holtzman et al., 2020) reduces the token space to the smallest subset of tokens with a total probability no less than $p$. To restrict the nucleus to tokens whose information content is close to the expected one given prior context, locally typical sampling (Meister et al., 2023) focuses on the tokens with negative log-probability within a certain absolute range from the conditional entropy (and a total probability no less than $p$). Finally, Hewitt et al. (2022) assert that a language model learns a mixture of the true token distribution and a smoothing distribution to avoid infinite perplexity. For *de-smoothing* the distribution, they propose $\epsilon$- and $\eta$-sampling, which truncate tokens with a probability smaller than a threshold set *a priori* or dynamically through the entropy of the distribution, respectively. This threshold can also be set according to the magnitude of the highest probability, as in min-$p$ (Minh et al., 2025), or based on the logit rather than the probability distribution (Tang et al., 2025). However, such strategies do not guarantee the exclusion of the smoothing-induced tail. Contrastive decoding (Li et al., 2023) leverages the difference in likelihood between a large language model and a smaller, less capable one to prioritize tokens with sufficiently high probability under the expert model. However, it requires access to a smaller model with an identical vocabulary, which is not always available. While conceptually aligned, our method simplifies the threshold computation and provides more intuitive guarantees on the suitability of selected tokens.

## 3 DiffSampling

Given the probability distribution of the next token, let us imagine sorting it to have tokens in descending order based on their probability. Following Hewitt et al. (2022), only the first $D$ tokens have a positive probability under the true token distribution, while the remaining $|\mathcal{V}| - D$ tokens receive a non-zero final probability solely due to the smoothing distribution, which prevents infinite perplexity. To generate correct text, we need to limit our sampling among the first $D$ tokens, thus, we need to identify a cutting point that is as close as possible to the $D$-th token. We propose to achieve this by truncating after the largest difference between probabilities: the token to its left should be the least probable token that our model considers correct.

From a mathematical analysis perspective, this point is characterized simply and elegantly as the location where the derivative reaches its minimum. Let us consider a probability distribution $p(x_t)$ defined for a limited number of $x_t^{[1]} \dots x_t^{[N]}$, with $p()$ monotonically decreasing. According to the forward difference approximation, the discrete derivative of a function $f(x_t^{[n]})$ is defined as $\Delta f(x_t^{[n]}) = f(x_t^{[n+1]}) - f(x_t^{[n]})$, thus we have:

$$\Delta p(x_t^{[n]}) = \begin{cases} p(x_t^{[n+1]}) - p(x_t^{[n]}) & \text{if } n < N \\ -p(x_t^{[n]}) & \text{if } n = N \end{cases} \tag{1}$$

which is always non-positive. $\arg\min(\Delta p(x_t^{[n]}))$ represents the index of the last token before the point characterized by the largest difference between consecutive probabilities.

In particular, it seems plausible that $\arg\min(\Delta p(x_t^{[n]})) \leq D$, i.e., it either marks the point where the true distribution ends and smoothing begins to take effect, or a point within the true distribution that separates tokens with significantly higher probabilities from the rest. Indeed, due to the inner nature of smoothing, it seems unreasonable that the maximum difference is between tokens with zero probability under the true

---

**Algorithm 1** DiffSampling

---

    **Input:** probabilities $\mathsf{probs} = [p_t^{[1]} \dots p_t^{[N]}]$, lower bound $\mathsf{p\_lb} = p_{lb}$, upper bound $\mathsf{p\_min} = p_{min}$, temperature $\mathsf{tau} = \tau$.
    $\mathsf{sorted\_probs}, \mathsf{indices} = \textbf{sort}(\mathsf{probs})$
    $\mathsf{fwd\_probs} = \mathsf{sorted\_probs}[1\!:] + [0.0]$
    $\mathsf{delta\_probs} = \mathsf{fwd\_probs} - \mathsf{sorted\_probs}$
    **if** $\mathsf{p\_min} > 0.0$ **then**
        $\mathsf{lim} = \textbf{argmin}(\mathsf{sorted\_probs} > \mathsf{p\_min} \cdot \mathsf{sorted\_probs}[0]) - 1$
        $\mathsf{delta\_probs}[:\mathsf{lim}] = 0.0$
    **else**
        $\mathsf{nucleus} = \textbf{cumsum}(\mathsf{sorted\_probs}) < \mathsf{p\_lb}$
        $\mathsf{delta\_probs}[\mathsf{nucleus}] = 0.0$
    **end if**
    $\mathsf{cut\_idx} = \textbf{argmin}(\mathsf{delta\_probs})$
    $\mathsf{sorted\_probs}[\mathsf{cut\_idx}+1\!:] = 0.0$
    $\mathsf{probs} = \textbf{sort\_by\_idx}(\mathsf{sorted\_probs}, \mathsf{indices})$
    $\mathsf{logits} = \textbf{log}(\mathsf{probs}/\textbf{sum}(\mathsf{probs}))/\mathsf{tau}$
    $\mathsf{probs} = \textbf{softmax}(\mathsf{logits})$
    **Output:** $\mathsf{probs}$.

---

distribution, and thus only because of the smoothing distribution (see Appendix A for a formal analysis on when $\arg\min(\Delta p(x_t^{[n]}))$ is provably $\leq D$).

Building on this intuition, we propose *DiffSampling*, a family of three decoding strategies (the full algorithm is reported in Algorithm 1). The first one, which we call *DiffSampling-cut*, leverages the aforementioned property and consists of cutting the distribution tail at the right side of the minimum discrete derivative, i.e., sampling among the tokens $x_i, i \leq \arg\min(\Delta p(x_t^{[n]}))$. Due to the guarantee of selecting a correct token, which prioritizes reliability over aggressiveness, this approach can be seen as an improved greedy strategy: when the model has high confidence in a single token, it degenerates into the greedy strategy; otherwise, it preserves other appropriate tokens, increasing diversity. The next section provides a toy example to showcase this relation.

Since the minimum discrete derivative should guarantee the correctness of the truncation, *any* preserved token should come from the true distribution: we can sample at a higher temperature to foster diversity without the usual trade-off with quality. Note that although temperature scaling is typically applied before truncation, doing so alters the probability distribution, potentially shifting the minimum of the discrete derivative forward - possibly into the region of tokens that have zero probability under the true distribution. To preserve the mathematical properties discussed above, we instead apply temperature scaling *after* truncation.

However, as previously discussed, this cutoff point can fall within the true distribution, thereby excluding tokens that are still correct; a frequent scenario consists of the first token minimizing $\Delta p(x_t^{[n]})$, but still having a quite low probability. To address this issue, we propose two relaxations to *right-move* the truncation. The first one builds upon top-$p$ sampling and introduces a lower bound on the saved mass probability. *DiffSampling-lb* considers truncating based on $\Delta p(x_t^{[n]})$ in such a way that the resulting tokens have a total probability at least equal to the lower bound $p_{lb}$. In other words, given $k$ cardinality of the smallest subset of tokens whose total probability is not lower than $p_{lb}$, it computes the $\arg\min(\Delta p(x_t^{[n]}))$ for $n \geq k$ (i.e., the cutting point is between tokens not included in the top-$p$ nucleus). This approach can be seen as an improved top-$p$ sampling: it *corrects* the $p$ parameter via our derivative-based approach to include appropriate tokens after the selected nucleus.

Alternatively, we can build upon min-$p$ sampling by introducing a dynamic upper bound on the probability of truncated tokens. *DiffSampling-minp* considers truncating based on $\Delta p(x_t^{[n]})$ in such a way that no discarded tokens have a probability greater than $p_{min} \cdot \max_{v \in \mathcal{V}} p(v)$. In other words, given $j$ index of the

**Prompt:** *Natural language generation (NLG) is the subfield of artificial intelligence and computational linguistics that is concerned with the construction of computer systems that can _______*

| Token | Prob | Top-$p$ | Min-$p$ | D-cut | D-lb | D-minp |
|---|---|---|---|---|---|---|
| generate | 37.326 | 41.366 | 50.872 | 59.886 | 40.929 | 47.537 |
| produce | 25.002 | 27.709 | 34.076 | **40.114** | 27.416 | 31.842 |
| understand | 7.295 | 8.084 | 9.942 | - | 7.999 | 9.290 |
| create | 3.749 | 4.154 | 5.109 | - | 4.110 | 4.774 |
| naturally | 2.797 | 3.100 | - | - | 3.067 | **3.562** |
| perform | 2.352 | 2.606 | - | - | 2.579 | **2.995** |
| reason | 1.067 | 1.182 | - | - | 1.170 | - |
| be | 0.956 | 1.060 | - | - | 1.048 | - |
| ... | ... | ... | - | - | ... | - |
| recognize | 0.350 | 0.388 | - | - | 0.384 | - |
| , | 0.339 | 0.375 | - | - | 0.371 | - |
| read | 0.325 | - | - | - | **0.357** | - |
| respond | 0.321 | - | - | - | **0.352** | - |
| interpret | 0.318 | - | - | - | **0.348** | - |
| interact | 0.259 | - | - | - | - | - |

Table 1: Token probability comparison between top-$p$, min-$p$, and our methods, showing how they avoid treating tokens with very similar probabilities differently (reported in **bold**). The probabilities (in percentage) are taken from `SmolLM-135M-Instruct` (Ben Allal et al., 2024).

lowest-probable token with a probability greater than $p_{min} \cdot \max_{v \in \mathcal{V}} p(v)$, it computes the $\arg\min(\Delta p(x_t^{[n]}))$ for $n \geq j$. This approach can be seen as an improved min-$p$ sampling: if there are tokens after index $j$ with a probability very close to the threshold, it still preserves them.

Overall, *DiffSampling* can be seen as a sampling scheme governed by two parameters, i.e., the probability-mass lower bound $p_{lb}$ and the truncated probability upper bound $p_{min}$ (where *DiffSampling-cut* just assumes a value of 0.0 for the first and of 1.0 for the second), plus the additional temperature $\tau$.

## 4 Illustrative Example

To make it easier to understand the advantages of our methods, Table 1 presents an illustrative example comparing them with their most similar methods. For the sake of simplicity, top-$p$ and *DiffSampling-lb* consider the same $p = p_{lb} = 0.9$, while min-$p$ and *DiffSampling-minp* consider the same $p = p_{min} = 0.1$. As apparent, *DiffSampling-cut* improves upon the greedy strategy by also considering the second-most probable token, while both *DiffSampling-lb* and *DiffSampling-minp* improve upon top-$p$ and min-$p$ by not discarding tokens with very similar probability compared to preserved ones (for example, top-$p$ would discard the 'read' token while having only a 0.014% probability less than ','). Although the differences between standard methods and ours are often minimal (typically involving low-probability tokens), even a slight correction in the right direction, at the negligible computational cost of an $\arg\min$ function, can lead to meaningful improvements.

## 5 Experiments

To evaluate whether *DiffSampling* helps diversify outputs while maintaining a high accuracy, we test it on four case studies: math problem solving, text summarization, the divergent association task, and story generation. While slightly unconventional, these tasks are very different from each other, and provide meaningful ways to evaluate diversity *and* quality together, as they have quantifiable goals which can be reached in syntactically and semantically different ways.

## 5.1 Models and Baselines

In all our experiments, we start from a state-of-the-art language model and test various decoding strategies. For the math problem-solving tasks, we use the Llama2-based `MetaMath-7B-V1.0` model trained with self-supervised learning on MetaMathQA (Yu et al., 2024). For extreme text summarization and story generation, we utilize the `Llama-3.2-3B` model (Grattafiori et al., 2024), with both original and `-Instruct` versions. Finally, for the divergent association task, we consider `Meta-Llama-3-8B` (Grattafiori et al., 2024), using both pre-trained and DPO-tuned `-Instruct` versions. We study the performances of our three methods: *DiffSampling-cut*; *DiffSampling-lb* with $p_{lb} = 0.9$; and *DiffSampling-minp* with $p_{min} = 0.1$. While these values are sometimes sub-optimal (see Appendix G for a full ablation study), we chose to standardize their values to match those used for the top-$p$ and min-$p$ baselines. Indeed, we compare them with a total of 5 decoding-based baselines: greedy strategy; $\eta$-sampling (with $\eta = 0.0003$); locally typical sampling (with $p = 0.9$); top-$p$ sampling (with $p = 0.9$); and min-$p$ sampling (with $p = 0.1$). While other methods, such as selective sampling (Troshin et al., 2025), contrastive decoding (Li et al., 2023), and beam search (Roark, 2001), could also be considered, we restrict our analysis to sampling-based methods to ensure a fair comparison, selecting those with similar computational costs and operational principles to our approach.

## 5.2 Math Problem Solving

**Experimental Setup.** Solving math problems provides a useful case study for our decoding strategies, as it allows us to evaluate the correctness of solutions (as the percentage of correctly solved problems) and the diversity of procedures to arrive at the result. In particular, we consider the GSM8K (Cobbe et al., 2021) and MATH (Hendrycks et al., 2021) test sets; the relative prompts are reported in Appendix D. To avoid resource wasting, we focus on entries with a problem and a solution of no more than 512 tokens.

We evaluate the quality of solutions through the ratio of correctly solved problems, i.e., with pass@1. Instead, the diversity is computed according to two methods: expectation-adjusted distinct $N$-grams (EAD) (Liu et al., 2022) and sentence embedding cosine diversity (SBERT) (Hong et al., 2024), which should evaluate syntactic and semantic diversity, respectively (Kirk et al., 2024). EAD counts the number of distinct $N$-grams tokens (averaging over $N = 1 \dots 5$) and removes the bias toward shorter inputs by scaling the number of distinct tokens based on their expectations[3]. The SBERT metric is 1 minus the cosine similarity between the embeddings of the sentences. While originally based on Sentence-BERT (Reimers & Gurevych, 2019), we employ the more recent `all-mpnet-base-v2` to obtain the embeddings, as suggested by their developers[4]. Following Kirk et al. (2024), we compute *cross-input* EAD and SBERT, i.e., by considering the set of all outputs produced for a specific seed. In addition, we also compute *against-greedy* EAD and SBERT. Given each input, we compare the output with the greedy one by calculating the average expectation-adjusted distinct $N$-grams not present in the greedy response, and 1 minus the cosine similarity between the two outputs, respectively. We refer the interested reader to Appendix C for a formal definition of all used metrics. Finally, for a more fine-grained analysis, Appendix H reports a few examples of generated outputs.

**Experimental Results.** Table 2 (left side) reports the results for the GSM8K test set. The greedy strategy achieves the highest average accuracy, closely followed by *DiffSampling-cut*. Among the other baselines, only locally typical sampling performs comparably, while *DiffSampling-lb* and *DiffSampling-minp* do not substantially differ from top-$p$ and min-$p$ on any metric (but different $p_{lb}$ or $p_{min}$ values can significantly improve the accuracy at a very small cost in diversity; see Appendix G).

Table 2 (right side) reports the results for the MATH test set. Here, the highest accuracy is reached by *DiffSampling-cut*, which also improves on the greedy strategy in terms of diversity. By contrast, our other two methods offer limited improvements over the sampling-based baselines. Notably, all methods achieve the same cross-input SBERT score, i.e., the overall diversity between all outputs is always the same across

---

[3]Note that EAD is not upper-bounded. Moreover, it counts for distinct $N$-grams across all outputs, including inside the same output: the EAD of a set of equal, non-empty sentences is not 0, as each sentence will contain at least one distinct 1-gram. In general, an EAD score cannot be considered high or low per se, but it must be compared with other EAD scores from experiments under similar conditions.

[4]https://huggingface.co/sentence-transformers/bert-large-nli-stsb-mean-tokens

| Dataset: | GSM8K | | | | | MATH | | | | |
|---|---|---|---|---|---|---|---|---|---|---|
| Method | Accuracy | Cross-Input | | Against-Greedy | | Accuracy | Cross-Input | | Against-Greedy | |
| | | EAD | SBERT | EAD | SBERT | | EAD | SBERT | EAD | SBERT |
| Greedy | $66.44_{\pm.09}$ | $2.03_{\pm.00}$ | $0.64_{\pm.00}$ | - | - | $20.62_{\pm.01}$ | $5.65_{\pm.00}$ | $0.80_{\pm.00}$ | - | - |
| Top-$p$ | $65.00_{\pm.18}$ | $2.08_{\pm.01}$ | $0.64_{\pm.00}$ | $0.23_{\pm.00}$ | $0.03_{\pm.00}$ | $20.02_{\pm.12}$ | $6.08_{\pm.02}$ | $0.80_{\pm.00}$ | $0.36_{\pm.00}$ | $0.10_{\pm.00}$ |
| $\eta$-sampling | $65.05_{\pm.19}$ | $2.12_{\pm.00}$ | $0.64_{\pm.00}$ | $0.25_{\pm.00}$ | $0.04_{\pm.00}$ | $19.67_{\pm.20}$ | $6.36_{\pm.01}$ | $0.80_{\pm.00}$ | $0.39_{\pm.00}$ | $0.11_{\pm.00}$ |
| Locally typical | $66.29_{\pm.55}$ | $2.09_{\pm.00}$ | $0.64_{\pm.00}$ | $0.23_{\pm.00}$ | $0.03_{\pm.00}$ | $19.95_{\pm.26}$ | $6.06_{\pm.01}$ | $0.80_{\pm.00}$ | $0.36_{\pm.00}$ | $0.10_{\pm.00}$ |
| Min-$p$ | $65.76_{\pm.44}$ | $2.09_{\pm.00}$ | $0.64_{\pm.00}$ | $0.23_{\pm.00}$ | $0.03_{\pm.00}$ | $20.25_{\pm.09}$ | $6.09_{\pm.01}$ | $0.80_{\pm.00}$ | $0.36_{\pm.00}$ | $0.10_{\pm.00}$ |
| DiffS.-cut | $66.36_{\pm.23}$ | $2.04_{\pm.00}$ | $0.64_{\pm.00}$ | $0.14_{\pm.00}$ | $0.02_{\pm.00}$ | $21.38_{\pm.20}$ | $5.71_{\pm.01}$ | $0.80_{\pm.00}$ | $0.27_{\pm.00}$ | $0.07_{\pm.00}$ |
| DiffS.-lb | $65.18_{\pm.65}$ | $2.09_{\pm.01}$ | $0.64_{\pm.00}$ | $0.23_{\pm.00}$ | $0.03_{\pm.00}$ | $20.20_{\pm.08}$ | $6.11_{\pm.02}$ | $0.80_{\pm.00}$ | $0.37_{\pm.00}$ | $0.10_{\pm.00}$ |
| DiffS.-minp | $65.48_{\pm.60}$ | $2.09_{\pm.01}$ | $0.64_{\pm.00}$ | $0.23_{\pm.00}$ | $0.03_{\pm.00}$ | $20.18_{\pm.08}$ | $6.06_{\pm.00}$ | $0.80_{\pm.00}$ | $0.36_{\pm.00}$ | $0.10_{\pm.00}$ |

Table 2: Accuracy and diversity of results for the GSM8K and MATH test sets over 3 seeds. The mean and standard error of the final score for each run are reported for accuracy and cross-input diversity, whereas the mean and the 95% confidence interval for the full set of answers are reported for against-greedy diversity.

different methods, which might be due to the very similar levels of accuracy (and, therefore, due to the similar meaning of the proposed solutions).

## 5.3 Extreme Summarization

**Experimental Setup.** Summarizing paragraphs represents another meaningful case study since the same text can be correctly outlined in different ways. To keep the resource consumption as low as possible, we consider the eXtreme Summarization (XSum) dataset (Narayan et al., 2018), which contains pairs of documents and one-sentence summaries. In particular, we use the test partition (11334 entries) and exclude all entries with a tokenized document longer than 768, obtaining 9815 entries; then, we limit our experiment to 1000 random samples, and we use the prompt suggested by Chhabra et al. (2024) and reported in Appendix D. Again, we aim to verify whether the summaries generated with *DiffSampling* are both diverse and of high quality. For diversity, we consider the per-input EAD and SBERT metrics, computed over five outputs sampled from the same prompt (Kirk et al., 2024), along with the against-greedy EAD and SBERT diversity scores introduced above. For quality assessment, we use ROUGE-1 (R-1) (Lin, 2004), a standard metric for summarization that evaluates the ratio of 1-grams present in both the target and generated summaries, as well as the sentence embedding cosine similarity (SIM) between the generated and target summaries. In this way, we compute both syntactic and semantic quality metrics, as a good summary might use entirely different words while still preserving the original meaning. In addition, following Su et al. (2022), we compute the coherence (COH) between the generated output and the text to summarize through the cosine similarity between their SimCSE embeddings (Gao et al., 2021).

| Model: | RLHF-instructed | | | | | | | Pre-trained | | | | | | |
|---|---|---|---|---|---|---|---|---|---|---|---|---|---|---|
| Method | Quality | | | Per-Input | | Against-Greedy | | Quality | | | Per-Input | | Against-Greedy | |
| | R-1 | SIM | COH | EAD | SBERT | EAD | SBERT | R-1 | SIM | COH | EAD | SBERT | EAD | SBERT |
| Greedy | $0.23_{\pm.00}$ | $0.49_{\pm.01}$ | $0.63_{\pm.01}$ | $0.18_{\pm.00}$ | - | - | - | $0.22_{\pm.00}$ | $0.51_{\pm.00}$ | $0.74_{\pm.00}$ | $0.19_{\pm.00}$ | - | - | - |
| Top-$p$ | $0.21_{\pm.00}$ | $0.45_{\pm.01}$ | $0.59_{\pm.01}$ | $0.36_{\pm.01}$ | $0.47_{\pm.01}$ | $0.66_{\pm.01}$ | $0.41_{\pm.01}$ | $0.16_{\pm.00}$ | $0.34_{\pm.01}$ | $0.48_{\pm.01}$ | $0.72_{\pm.01}$ | $0.66_{\pm.01}$ | $0.77_{\pm.01}$ | $0.55_{\pm.01}$ |
| $\eta$-sampling | $0.20_{\pm.00}$ | $0.45_{\pm.01}$ | $0.58_{\pm.01}$ | $0.38_{\pm.01}$ | $0.49_{\pm.01}$ | $0.69_{\pm.01}$ | $0.43_{\pm.01}$ | $0.16_{\pm.00}$ | $0.34_{\pm.01}$ | $0.48_{\pm.01}$ | $0.75_{\pm.01}$ | $0.67_{\pm.01}$ | $0.80_{\pm.01}$ | $0.56_{\pm.01}$ |
| Locally typical | $0.21_{\pm.00}$ | $0.45_{\pm.01}$ | $0.59_{\pm.01}$ | $0.36_{\pm.01}$ | $0.47_{\pm.01}$ | $0.66_{\pm.01}$ | $0.41_{\pm.01}$ | $0.16_{\pm.00}$ | $0.34_{\pm.01}$ | $0.48_{\pm.01}$ | $0.72_{\pm.01}$ | $0.66_{\pm.01}$ | $0.77_{\pm.01}$ | $0.55_{\pm.01}$ |
| Min-$p$ | $0.22_{\pm.00}$ | $0.46_{\pm.01}$ | $0.61_{\pm.01}$ | $0.36_{\pm.01}$ | $0.43_{\pm.01}$ | $0.64_{\pm.01}$ | $0.38_{\pm.01}$ | $0.20_{\pm.00}$ | $0.44_{\pm.01}$ | $0.63_{\pm.01}$ | $0.65_{\pm.01}$ | $0.47_{\pm.01}$ | $0.62_{\pm.01}$ | $0.39_{\pm.01}$ |
| DiffS.-cut | $0.23_{\pm.00}$ | $0.48_{\pm.01}$ | $0.63_{\pm.01}$ | $0.35_{\pm.01}$ | $0.25_{\pm.01}$ | $0.45_{\pm.01}$ | $0.23_{\pm.01}$ | $0.21_{\pm.00}$ | $0.49_{\pm.00}$ | $0.73_{\pm.00}$ | $0.38_{\pm.01}$ | $0.19_{\pm.00}$ | $0.32_{\pm.01}$ | $0.17_{\pm.01}$ |
| DiffS.-lb | $0.21_{\pm.00}$ | $0.45_{\pm.01}$ | $0.59_{\pm.01}$ | $0.37_{\pm.01}$ | $0.47_{\pm.01}$ | $0.67_{\pm.01}$ | $0.41_{\pm.01}$ | $0.16_{\pm.00}$ | $0.34_{\pm.01}$ | $0.48_{\pm.01}$ | $0.72_{\pm.01}$ | $0.66_{\pm.01}$ | $0.77_{\pm.01}$ | $0.55_{\pm.01}$ |
| DiffS.-minp | $0.22_{\pm.00}$ | $0.46_{\pm.01}$ | $0.60_{\pm.01}$ | $0.35_{\pm.01}$ | $0.43_{\pm.01}$ | $0.64_{\pm.01}$ | $0.38_{\pm.01}$ | $0.20_{\pm.00}$ | $0.44_{\pm.01}$ | $0.62_{\pm.01}$ | $0.65_{\pm.01}$ | $0.47_{\pm.01}$ | $0.63_{\pm.01}$ | $0.39_{\pm.01}$ |

Table 3: Aggregate results over 5 outputs sampled for each of the 1000 prompts from the XSum dataset for the instructed model (left) and the pre-trained model (right). The mean and 95% confidence interval are reported for all the metrics.

**Experimental Results.** With respect to the instructed model, as reported in Table 3 (left), all methods achieve similar quality scores, with the greedy strategy and *DiffSampling-cut* performing slightly better, followed closely by min-$p$ and its relaxed variant, *DiffSampling-minp*. The remaining sampling methods yield comparable scores, with the notable exception of $\eta$-sampling, which achieves the highest against-greedy diversity but the lowest similarity and coherence scores, thus confirming the well-known quality–diversity trade-off (Ippolito et al., 2019).

On the other hand, as shown in Table 3 (right), the quality metrics exhibit greater variation for the non-instructed model: *DiffSampling-cut* outperforms all other sampling methods and performs on par with the greedy strategy. While it does not reach the same diversity scores as the other sampling methods, it provides consistent deviations from greedy. Just below, *DiffSampling-minp* and min-*p* obtain very similar scores across all metrics, while outperforming the other methods in terms of accuracy as they deviate less from greedy decoding. Finally, as expected, *DiffSampling-lb* closely aligns with top-*p*, with negligible differences in diversity.

### 5.4 Divergent Association Task

**Experimental Setup.** The third use case considers the divergent association task (DAT) (Chen & Ding, 2023). Building on the theory that creativity is related to the capability of generating more divergent ideas (Beaty et al., 2014), it requires participants to name unrelated words. In particular, the task is the following:

> Write 10 nouns in English that are as irrelevant from each other as possible, in all meanings and uses of the words. Please note that the words you write should have only single word, only nouns (e.g., things, objects, concepts), and no proper nouns (e.g., no specific people or places).

Then, their semantic distance can represent an objective measure of divergent thinking (Olson et al., 2021). DAT is a useful case study for decoding strategies as it constrains the generation to different nouns (thus, assuming an optimal probability distribution, the tail due to smoothing should contain everything else) and requires generating terms that are as different as possible, which is quite the opposite to what typically happens in language modeling: an optimal strategy should exclude non-appropriate tokens but also not limit too much the space of possible tokens. We strictly follow the setup proposed by Chen & Ding (2023). More concretely, given the embeddings of $n$ words, the DAT score is the average cosine distance between each pair of embeddings (then scaled as a percentage). We use GloVe embeddings (Pennington et al., 2014) and ask the model to generate a list of 10 nouns. We discard outputs without at least 7 distinct nouns and compute the DAT score for all other outputs over their first 7 nouns. For completeness, Appendix G.1 reports results obtained when considering all 10 nouns. We repeat the experiment 100 times for non-greedy strategies to mitigate the sampling stochasticity.

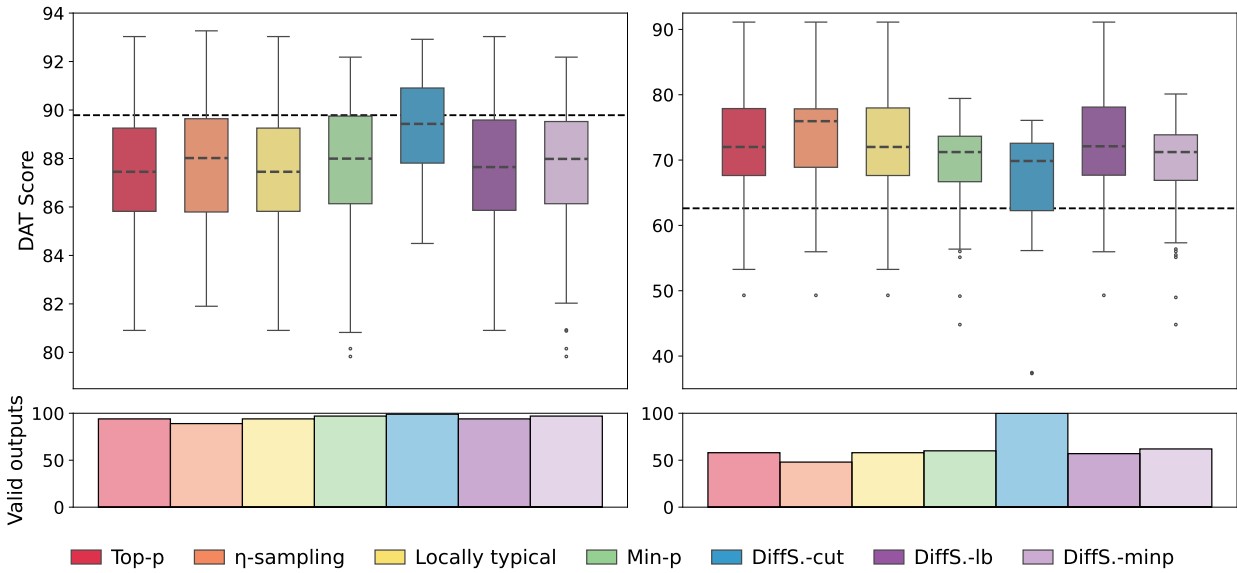

Figure 2: DAT scores for our methods and the baselines over the instructed (left) and pre-trained (right) model. Below, the number of valid outputs produced by each sampling strategy. The dashed line reports the greedy score.

**Experimental Results.** Figure 2 summarizes the DAT results, reporting both diversity (DAT score) and quality (count of valid outputs) measures. For `Meta-Llama-3-8B-Instruct`, the greedy strategy produces a strong set of nouns, achieving a DAT score higher than the average across all sampling methods. However, *DiffSampling-cut* generates a better set in almost half of the generations, and always produces a valid set of nouns. Instead, the other sampling schemes can produce better scores only occasionally, while sometimes failing at providing a valid set of nouns, and both *DiffSampling-lb* and *DiffSampling-minp* perform slightly better or almost identically to the top-$p$ and min-$p$, respectively. The results for the pre-trained version `Meta-Llama-3-8B` are quite different, and the quality-diversity trade-off is more pronounced. *DiffSampling-cut* is substantially better than the greedy strategy, and it produces only valid outputs. However, all other methods achieve higher DAT scores while producing significantly fewer valid outputs. Although very similar, both *DiffSampling-minp* and *DiffSampling-lb* outperform their min-$p$ and top-$p$ counterparts, yielding either slightly higher scores or a greater number of valid outputs.

## 5.5 WritingPrompts

**Experimental Setup.** The previous case studies focus on the generation of short or very short outputs. However, certain issues emerge only in longer-form tasks—for example, the tendency of greedy decoding to repeat tokens, thereby degrading text quality (Fu et al., 2021). To address this limitation, the final case study involves generating stories of up to 1024 tokens using inputs from the WritingPrompts dataset (Fan et al., 2018), which comprises a large collection of prompts sourced from Reddit's WritingPrompts forum. In particular, we sample 500 test prompts among those labeled as *standard* prompts (i.e., that start with `[WP]`), and we generate 5 outputs for each sampling scheme. Then, we evaluate their quality through their coherence (COH) with the prompt as the cosine similarity between their SimCSE embeddings (Gao et al., 2021); instead, diversity is computed through the per-input EAD and SBERT metrics, i.e., calculated among the outputs sampled given the same prompt (Kirk et al., 2024).

| Model: | RLHF-instructed | | | Pre-trained | | |
|---|---|---|---|---|---|---|
| Method | Quality | Per-Input Diversity | | Quality | Per-Input Diversity | |
| | COH | EAD | SBERT | COH | EAD | SBERT |
| Greedy | $0.44_{\pm.01}$ | $0.17_{\pm.01}$ | - | $0.59_{\pm.01}$ | $0.07_{\pm.00}$ | - |
| Top-$p$ | $0.42_{\pm.01}$ | $0.73_{\pm.00}$ | $0.25_{\pm.00}$ | $0.42_{\pm.01}$ | $0.64_{\pm.00}$ | $0.58_{\pm.00}$ |
| $\eta$-sampling | $0.42_{\pm.01}$ | $0.80_{\pm.00}$ | $0.28_{\pm.00}$ | $0.40_{\pm.01}$ | $0.77_{\pm.00}$ | $0.60_{\pm.00}$ |
| Locally typical | $0.42_{\pm.01}$ | $0.73_{\pm.00}$ | $0.25_{\pm.00}$ | $0.42_{\pm.01}$ | $0.64_{\pm.00}$ | $0.58_{\pm.00}$ |
| Min-$p$ | $0.43_{\pm.01}$ | $0.71_{\pm.00}$ | $0.23_{\pm.00}$ | $0.51_{\pm.01}$ | $0.35_{\pm.01}$ | $0.46_{\pm.00}$ |
| DiffS.-cut | $0.43_{\pm.01}$ | $0.63_{\pm.00}$ | $0.19_{\pm.00}$ | $0.60_{\pm.01}$ | $0.15_{\pm.00}$ | $0.31_{\pm.01}$ |
| DiffS.-lb | $0.42_{\pm.01}$ | $0.73_{\pm.00}$ | $0.25_{\pm.00}$ | $0.41_{\pm.01}$ | $0.67_{\pm.00}$ | $0.58_{\pm.00}$ |
| DiffS.-minp | $0.43_{\pm.01}$ | $0.71_{\pm.00}$ | $0.23_{\pm.00}$ | $0.51_{\pm.01}$ | $0.36_{\pm.00}$ | $0.47_{\pm.00}$ |

Table 4: Aggregate results for the WritingPrompts dataset for the instructed model (left) and the pre-trained model (right). The mean and the 95% confidence interval for the full set of answers are reported for all the metrics.

**Experimental Results.** Table 4 reports the results for both instructed (left) and pre-trained (right) models. For the former, coherence remains largely consistent across all methods, while diversity metrics vary depending on the greediness of the approach: *DiffSampling-cut* produces outputs that are significantly different from each other, though still less diverse than those generated by the other sampling-based baselines, among which $\eta$-sampling achieves the best performance. In contrast, coherence varies more noticeably for the pre-trained model, where *DiffSampling-cut* achieves the highest score alongside the greedy strategy, but with substantial improvements in diversity—highlighted by near-zero scores for the greedy strategy, which suggest it tends to repeat the same tokens indefinitely. Instead, *DiffSampling-minp* and *DiffSampling-lb* match the coherence levels of min-$p$ and top-$p$, respectively, while offering notable gains in EAD diversity for the non-instructed model, likely due to certain tokens being correctly preserved from truncation.

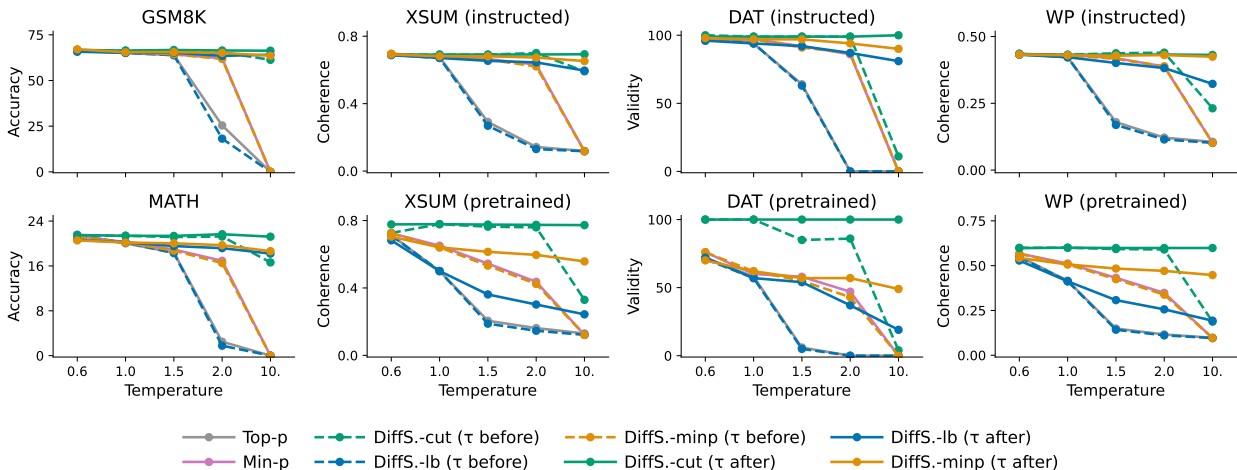

Figure 3: Average quality scores across different temperature values for top-$p$, min-$p$, and our methods when applying temperature before (dashed line) and after (full line) the truncation.

## 5.6 Temperature Scaling

Finally, we experiment with different temperature values $\tau$, i.e., 0.6, 1, 1.5, 2, and 10. As detailed above, to preserve the mathematical guarantees of our approach, we apply temperature *after* the *DiffSampling* truncation, while our baselines apply this before. Due to the different nature of temperature scaling, this comparison is intended only to highlight the impact of temperature position, rather than to imply that our method is superior to the baseline. To allow for a fairer analysis, we also report quality scores for *DiffSampling* when applying temperature before truncation (see Appendix F.5 for a full comparison between temperature before and after truncation). As shown by Figure 3, *DiffSampling+temperature* preserves the output quality, and relevant differences only occur with our two relaxations and pre-trained models. Instead, the output quality rapidly drops with higher temperatures for the min-$p$ (by far the best of our baselines at $\tau > 1$) and top-$p$ baselines. In particular, the non-significant loss in quality for *DiffSampling-cut* confirms that our truncation strategy only preserves correct tokens. At the same time, temperature scaling has an (overall positive) impact on diversity; we refer to Appendix F for a detailed analysis of how all our quality and diversity metrics change at different $\tau$.

## 6 Discussion

Overall, *DiffSampling-cut* has demonstrated performance better than or equal to the greedy strategy. Additionally, it offers the potential for greater diversity. By introducing a lower bound on the preserved total probability or an upper bound on the probability of truncated ones, the method can further relax selection constraints, enabling greater output diversity at the expense of a marginal reduction in prediction accuracy. Once truncation is applied, sampling at higher temperatures becomes viable, promoting greater variability without significantly compromising output quality.

However, selecting the most appropriate method and hyperparameters is not straightforward and requires a case-by-case analysis. If a small validation set is available, the choice of which strategy and parameters can be made empirically. Otherwise, our experiments show that *DiffSampling-cut* works best when the task requires precision: whenever a user might otherwise rely on a greedy decoding strategy or a very low temperature, it enhances diversity without compromising accuracy. *DiffSampling-lb* fosters output diversity by trading off some accuracy, especially at higher values of $p_{lb}$ and, thus, appears most appropriate for divergent solutions. *DiffSampling-minp* is more well-balanced. Both can be used in place of top-$p$ and min-$p$ to "correct" them and potentially improve their diversity with no additional overhead. Increasing the temperature has proven highly effective for fine-tuned models across all methods, whenever it is not strictly necessary to preserve the original distribution.

Our experiments were limited to relatively small LLMs, although preliminary results suggest that the same findings hold for larger models as well (see Appendix E for a more detailed analysis), and based on quantitative, automatic evaluation. Several of the adopted metrics exhibit significant limitations (e.g., Schluter, 2017), often failing to align with human judgments (Tevet & Berant, 2021). Moreover, abstract concepts such as originality and creativity remain inherently difficult to define with precision (Franceschelli & Musolesi, 2025). We plan to experiment with human evaluators to verify whether the quality and diversity that *DiffSampling* aims to provide are also perceived by potential users.

# 7 Conclusion

In this paper, we have presented *DiffSampling*, a novel family of decoding strategies based on the analysis of the next-token distribution. In particular, given the distribution sorted in descending order, we compute the forward difference approximation of its discrete derivative and use it to remove tokens after its minimum value (possibly together with relaxations to allow for more diversity). In this way, we avoid incorrect tokens under the learned distribution. We have experimented with four different tasks, and our method has consistently performed at least as well as similar strategies in terms of accuracy, despite sampling from a larger set of tokens, which has a positive impact on diversity.

Our research agenda includes investigating whether combining *DiffSampling* with complementary techniques, such as re-ranking or controllable generation, can lead to further improvements in output quality. We also plan to leverage additional properties of the underlying probability distribution (e.g., its entropy (Potraghloo et al., 2025)), beyond token likelihoods, to guide generation toward desired characteristics such as coherence, novelty, or user-specific preferences. These directions open up promising opportunities for enhancing the adaptability of text generation systems in general-purpose and task-specific settings.

**Broader Impact Statement**

While our decoding scheme should, in theory, not increase the risk of generating tokens outside the true support, it may still produce unsafe content in certain contexts if the learned distribution itself is unsafe (e.g., containing learned biases, inappropriate language, or misleading information). Thus, it is important to continue using safety filters and domain constraints. Finally, we also perform a small check to ensure that *DiffSampling* does not increase unsafe content rates in the WritingPrompts use case, as it is the most open-ended generation task. Using `Llama-Guard-3-8B` (Inan et al., 2023), we found that the probability of unsafe outputs generated by our methods is identical to that of the corresponding methods at a temperature of 1.0 or lower. However, the application of temperature after the truncation dramatically reduces the rate of unsafe generated text, especially for *DiffSampling-cut* and *DiffSampling-minp*. We report our results in Table 5.

| Method | Pre-Trained | | | | | | Instructed | | | | | |
|---|---|---|---|---|---|---|---|---|---|---|---|---|
| | $\tau = 0.0$ | $\tau = 0.6$ | $\tau = 1.0$ | $\tau = 1.5$ | $\tau = 2.0$ | $\tau = 10.$ | $\tau = 0.0$ | $\tau = 0.6$ | $\tau = 1.0$ | $\tau = 1.5$ | $\tau = 2.0$ | $\tau = 10.$ |
| **Baselines** | | | | | | | | | | | | |
| Greedy | $0.12_{\pm.02}$ | - | - | - | - | - | $0.04_{\pm.01}$ | - | - | - | - | - |
| Top-$p$ | - | $0.12_{\pm.01}$ | $0.09_{\pm.01}$ | $0.62_{\pm.01}$ | $0.70_{\pm.01}$ | $0.69_{\pm.01}$ | - | $0.04_{\pm.01}$ | $0.05_{\pm.01}$ | $0.57_{\pm.01}$ | $0.69_{\pm.01}$ | $0.71_{\pm.01}$ |
| $\eta$-sampling | - | $0.11_{\pm.01}$ | $0.11_{\pm.01}$ | $0.72_{\pm.01}$ | $0.70_{\pm.01}$ | $0.68_{\pm.01}$ | - | $0.04_{\pm.01}$ | $0.11_{\pm.01}$ | $0.62_{\pm.01}$ | $0.69_{\pm.01}$ | $0.69_{\pm.01}$ |
| Locally typical | - | $0.12_{\pm.01}$ | $0.09_{\pm.01}$ | $0.66_{\pm.01}$ | $0.69_{\pm.01}$ | $0.68_{\pm.01}$ | - | $0.04_{\pm.01}$ | $0.05_{\pm.01}$ | $0.58_{\pm.01}$ | $0.69_{\pm.01}$ | $0.70_{\pm.01}$ |
| Min-$p$ | - | $0.11_{\pm.01}$ | $0.10_{\pm.01}$ | $0.09_{\pm.01}$ | $0.19_{\pm.01}$ | $0.68_{\pm.01}$ | - | $0.05_{\pm.01}$ | $0.04_{\pm.01}$ | $0.04_{\pm.01}$ | $0.30_{\pm.01}$ | $0.69_{\pm.01}$ |
| **Ours ($\tau$ before)** | | | | | | | | | | | | |
| DiffS.-cut | - | $0.13_{\pm.01}$ | $0.12_{\pm.01}$ | $0.12_{\pm.01}$ | $0.12_{\pm.01}$ | $0.58_{\pm.01}$ | - | $0.04_{\pm.01}$ | $0.04_{\pm.01}$ | $0.04_{\pm.01}$ | $0.04_{\pm.01}$ | $0.47_{\pm.01}$ |
| DiffS.-lb | - | $0.12_{\pm.01}$ | $0.10_{\pm.01}$ | $0.71_{\pm.01}$ | $0.70_{\pm.01}$ | $0.68_{\pm.01}$ | - | $0.04_{\pm.01}$ | $0.05_{\pm.01}$ | $0.64_{\pm.01}$ | $0.69_{\pm.01}$ | $0.69_{\pm.01}$ |
| DiffS.-minp | - | $0.11_{\pm.01}$ | $0.10_{\pm.01}$ | $0.09_{\pm.01}$ | $0.22_{\pm.01}$ | $0.68_{\pm.01}$ | - | $0.04_{\pm.01}$ | $0.04_{\pm.01}$ | $0.05_{\pm.01}$ | $0.30_{\pm.01}$ | $0.69_{\pm.01}$ |
| **Ours ($\tau$ after)** | | | | | | | | | | | | |
| DiffS.-cut | - | $0.12_{\pm.01}$ | $0.12_{\pm.01}$ | $0.12_{\pm.01}$ | $0.12_{\pm.01}$ | $0.12_{\pm.01}$ | - | $0.04_{\pm.01}$ | $0.04_{\pm.01}$ | $0.04_{\pm.01}$ | $0.04_{\pm.01}$ | $0.05_{\pm.01}$ |
| DiffS.-lb | - | $0.11_{\pm.01}$ | $0.10_{\pm.01}$ | $0.44_{\pm.01}$ | $0.52_{\pm.01}$ | $0.51_{\pm.01}$ | - | $0.04_{\pm.01}$ | $0.05_{\pm.01}$ | $0.32_{\pm.01}$ | $0.37_{\pm.01}$ | $0.40_{\pm.01}$ |
| DiffS.-minp | - | $0.11_{\pm.01}$ | $0.10_{\pm.01}$ | $0.09_{\pm.01}$ | $0.09_{\pm.01}$ | $0.08_{\pm.01}$ | - | $0.04_{\pm.01}$ | $0.04_{\pm.01}$ | $0.04_{\pm.01}$ | $0.05_{\pm.01}$ | $0.04_{\pm.01}$ |

Table 5: Unsafe probability of WritingPrompts outputs for baselines and our methods at different temperatures according to `Llama-Guard-3-8B` The mean and the 95% confidence interval for the full set of answers are reported.

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

## A  Formal Analysis

In the following, we aim to formally define the conditions under which our truncation strategy is *safe*, i.e., the conditions under which all tokens up to $\arg\min \Delta p(x_t^{[n]})$ have a positive probability under the true distribution.

According to Hewitt et al. (2022), we can define the true probability distribution as $P_\star(\cdot|x_{<i}) = \{p_\star^{[1]}, \ldots, p_\star^{[|\mathcal{V}|]}\}$ with $\sum_{i=1}^{|\mathcal{V}|} p_\star^{[i]} = 1$ and $p_\star^{[i]} \geq p_\star^{[i+1]} \ \forall \ i \in [1, |\mathcal{V}| - 1]$, and where exists a $D < |\mathcal{V}|$ such

that $p_\star^{[i]} = 0 \; \forall \; i > D$ (with $D = |\mathcal{V}|$, any truncation strategy is *safe*). Let us define a smoothing distribution $Q(\cdot|x_{<i}) = \{q^{[1]}, \ldots, q^{[|\mathcal{V}|]}\}$ with $q^{[i]} \in (\frac{1-\delta}{|\mathcal{V}|}, \frac{1+\delta}{|\mathcal{V}|}) \; \forall \; i \in [1, |\mathcal{V}|]$ and $\delta$ constant between 0 and 1. Then, the learned distribution $P_\theta(\cdot|x_{<i}) = \{p^{[1]}, \ldots, p^{[|\mathcal{V}|]}\}$ with $\sum_{i=1}^{|\mathcal{V}|} p^{[i]} = 1$ and $p^{[i]} \geq p^{[i+1]} \; \forall \; i \in [1, |\mathcal{V}| - 1]$ can be defined as the weighted sum between the true probability distribution and the smoothing distribution:

$$P_\theta(\cdot|x_{<i}) = \lambda_{<i} P_\star(\cdot|x_{<i}) + (1 - \lambda_{<i}) Q(\cdot|x_{<i}) \tag{2}$$

where $\lambda_{<i} \in (0, 1]$. However, according to Hewitt et al. (2022), we can assume that $\lambda_{<i} \geq \max(\overline{\lambda}_{<i}, \overline{\lambda})$, where $\overline{\lambda}_{<i} = 1 - \frac{|\mathcal{V}| \alpha \exp^{-H_{<i}}}{1+\delta}$ with $H_{<i}$ entropy of $P_\star(\cdot|x_{<i})$, and $\overline{\lambda}$ constant close to 1; for simplicity, we will follow Hewitt et al. (2022) and assume to have $\overline{\lambda} = 0.8$. This has two implications. First, the contribution provided by the smoothing distribution is bounded by $\alpha \exp^{-H_{<i}}$ with $\alpha \in [0, 1]$ and generally very small, so the actual contribution depends on the entropy of the true distribution. Second, the weighting factor $\lambda_{<i}$ has a lower bound equal to $\overline{\lambda} = 0.8$.

In this article, we propose to truncate the learned probability distribution at an index $K$ such that $K = \arg\min_i(p^{[i+1]} - p^{[i]}) = \arg\max_i(p^{[i]} - p^{[i+1]})$. The truncation is *safe* when $K \leq D$, i.e., if the truncation only preserves tokens with a non-zero probability under the true distribution.

**Proposition 1.** *Given a learned probability distribution $P_\theta(\cdot|x_{<i}) = \{p^{[1]}, \ldots, p^{[|\mathcal{V}|]}\}$ sorted in descending order, the truncation performed by means of $\arg\max_i(p^{[i]} - p^{[i+1]})$ only preserves tokens from the support of the true distribution $P_\star(\cdot|x_{<i})$ if $\max_i(p_\star^{[i]} - p_\star^{[i+1]}) > \frac{1}{|\mathcal{V}|}$.*

*Proof.* The truncation is *safe* whenever $K \leq D$, i.e., whenever the maximum difference between a token from the true support and its next token is greater than the maximum difference between tokens from outside the true support. The maximum difference between tokens from outside the true support, i.e., with $p_\star^{[i]} = p_\star^{[i+1]} = 0$, is:

$$
\begin{aligned}
\max_{i>D}(p^{[i]} - p^{[i+1]}) &= \max_{i>D}((1 - \lambda_{<i})q^{[i]} - (1 - \lambda_{<i})q^{[i+1]}) \\
&= (1 - \lambda_{<i})(\frac{1+\delta}{|\mathcal{V}|} - \frac{1-\delta}{|\mathcal{V}|}) = (1 - \lambda_{<i})\frac{2\delta}{|\mathcal{V}|}.
\end{aligned}
\tag{3}
$$

Instead, the maximum difference between a token from the true support and its next token is given by:

$$
\begin{aligned}
\max_{i \leq D}(p^{[i]} - p^{[i+1]}) &= \max_{i \leq D}(\lambda_{<i} p_\star^{[i]} + (1 - \lambda_{<i})q^{[i]} - \lambda_{<i} p_\star^{[i+1]} - (1 - \lambda_{<i})q^{[i+1]}) \\
&= \max_{i \leq D}(\lambda_{<i}(p_\star^{[i]} - p_\star^{[i+1]}) + (1 - \lambda_{<i})(q^{[i]} - q^{[i+1]})).
\end{aligned}
\tag{4}
$$

This value is lower-bounded by $\max_{i \leq D}(\lambda_{<i}(p_\star^{[i]} - p_\star^{[i+1]})) + \min_{i \leq D}((1 - \lambda_{<i})(q^{[i]} - q^{[i+1]})) = \lambda_{<i} \max_{i \leq D}(p_\star^{[i]} - p_\star^{[i+1]}) + (1 - \lambda_{<i}) \min_{i \leq D}(q^{[i]} - q^{[i+1]})$. The second term is exactly the opposite of the maximum value computed above: $\min_i(q^{[i]} - q^{[i+1]}) = \frac{1-\delta}{|\mathcal{V}|} - \frac{1+\delta}{|\mathcal{V}|} = -\frac{2\delta}{|\mathcal{V}|}$. If we define $\Delta_\star^{[i]} = \max_{i \leq D}(p_\star^{[i]} - p_\star^{[i+1]})$, we obtain a lower-bounded maximum given by $\lambda_{<i} \Delta_\star^{[i]} - (1 - \lambda_{<i})\frac{2\delta}{|\mathcal{V}|}$.

To have $K \leq D$, we impose:

$$
\begin{aligned}
\lambda_{<i} \Delta_\star^{[i]} - (1 - \lambda_{<i})\frac{2\delta}{|\mathcal{V}|} &> (1 - \lambda_{<i})\frac{2\delta}{|\mathcal{V}|} \\
\lambda_{<i} \Delta_\star^{[i]} &> (1 - \lambda_{<i})\frac{4\delta}{|\mathcal{V}|} \\
\Delta_\star^{[i]} &> \frac{4\delta}{|\mathcal{V}|}\frac{(1 - \lambda_{<i})}{\lambda_{<i}}
\end{aligned}
\tag{5}
$$

Since $\lambda_{<i}$ is lower-bounded by $\overline{\lambda}$, the second term can be reduced to $\frac{(1-\overline{\lambda})}{\overline{\lambda}}$. As suggested by Hewitt et al. (2022), we assumed $\overline{\lambda} = 0.8$; we obtain that our truncation strategy is *safe* if $\Delta_\star^{[i]} > \frac{4\delta}{|\mathcal{V}|}\frac{1}{4} = \frac{\delta}{|\mathcal{V}|}$. Since $\delta$ is upper-bounded to 1, we get a lower bound of $\Delta_\star^{[i]} > \frac{1}{|\mathcal{V}|}$ that proves our proposition. $\qquad\square$

**Proposition 2.** *Given a learned probability distribution $P_\theta(\cdot|x_{<i}) = \{p^{[1]}, \ldots, p^{[|\mathcal{V}|]}\}$ sorted in descending order, the truncation performed by means of $\arg\max_i(p^{[i]} - p^{[i+1]})$ preserves tokens only from the support with size $D$ of the true distribution $P_\star(\cdot|x_{<i})$ if $D < \sqrt{2|\mathcal{V}|}$.*

*Proof.* The maximum difference $\max_i(p_\star^{[i]} - p_\star^{[i+1]})$ is lower-bounded by $\frac{2}{D(D+1)}$. This bound holds when the differences between the first $D+1$ tokens (with the first $D$ tokens having a positive probability and the $D+1$-th zero probability) are equal, i.e., when the first $D + 1$ tokens are equidistant. According to Proposition 1, $\arg\max_i(p^{[i]} - p^{[i+1]}) \le D$ if $\max_i(p_\star^{[i]} - p_\star^{[i+1]}) > \frac{1}{|\mathcal{V}|}$. Thus, this is also true when $\frac{2}{D(D+1)} > \frac{1}{|\mathcal{V}|}$, i.e., if $D(D + 1) < 2|\mathcal{V}|$, providing an upper bound for the true support size $D$ of $\approx \sqrt{2|\mathcal{V}|}$, which proves our proposition. $\qquad\square$

In summary, our truncation strategy is safe whenever we have $\max(p_\star^{[i]} - p_\star^{[i+1]}) > \frac{1}{|\mathcal{V}|}$ or $D < \sqrt{2|\mathcal{V}|}$. To provide a practical intuition of the meaning of these conditions, consider the case of an LLM, whose vocabulary size $|\mathcal{V}|$ is usually in the order of 50000. This sets the lower bound for the maximum difference at 0.00002 and the upper bound for the true support size at 316. If either of these conditions is satisfied, our truncation strategy is considered safe.

However, it is important to note that these two bounds represent only worst-case scenarios. First of all, we assumed that $\delta$ can be equal to 1 and $\lambda_{<i} = \overline{\lambda} = 0.8$. In practice, $\delta$ will be much smaller than 1, leading to a proportionally smaller lower bound for $\max(p_\star^{[i]} - p_\star^{[i+1]})$. Moreover, $\lambda_{<i}$ will also assume the value of $\overline{\lambda}_{<i}$, which is likely to be higher than $\overline{\lambda}$ when $H_{<i}$ is higher, i.e., for very homogeneous distributions (unbalanced distributions most likely have a $\max(p_\star^{[i]} - p_\star^{[i+1]}) > \frac{1}{|\mathcal{V}|}$). These two elements would make a realistic bound for $\max_i(p_\star^{[i]} - p_\star^{[i+1]})$ much looser. In addition, the proof of Proposition 2 considers an extremely conservative (and impossible to find in practice) lower bound for $\max_{i \le D}(p^{[i]} - p^{[i+1]})$. Indeed, if all probabilities are equidistant, $\max_{i \le D}(p^{[i]} - p^{[i+1]})$ is not lower-bounded by $\lambda_{<i}\frac{2}{D(D+1)} - (1 - \lambda_{<i})\frac{2\delta}{|\mathcal{V}|}$ but by $\lambda_{<i}\frac{2}{D(D+1)}$. While the latter does not provide a valid lower bound for $\max_{i \le D}(p^{[i]} - p^{[i+1]})$, the fact that the used lower bound cannot actually be attained, combined with the considerations regarding $\delta$ and $\lambda_{<i}$ discussed above, makes the upper bound for $D$ much looser in practice.

# B    Computational Infrastructure and Implementation Details

All experiments were carried out on a Linux-based local server equipped with 2 80GB NVIDIA H100 GPUs running Python 3.11.9. All the trainings were repeated, varying the random seed among 1, 42, and 121 (set through the `set_seed` method from the HuggingFace `transformers` library). The hyperparameters governing the sampling strategies adopted as baselines were selected according to the best results reported by their original paper for similar tasks and model sizes.

# C    Evaluation Metrics

In this section, we formally define the quantitative metrics used in our experiments.

- **Cross-Input Diversity:** The diversity of outputs across inputs, i.e., the diversity of $\pi(y)$: $D(\bigcup_{i=1}^{N} \pi(y_i|x_i))$, with $D$ diversity metric (Kirk et al., 2024). In the case of cross-input EAD diversity, this is practically done by computing the EAD score for the entire set of outputs at the same time (i.e., the expected average distinct $N$-grams among all the sequences) for $N \in [1, 5]$ and returning their average. We refer to the original paper by Liu et al. (2022) for the exact computation of EAD. On the contrary, in the case of cross-input SBERT diversity, this is practically implemented

by first encoding all outputs into latent vectors through a given text encoder, then computing all the possible pair-wise cosine similarity between different vectors, and finally returning 1 minus the average cosine similarity.

- **Against-Greedy Diversity:** The diversity between a given output $y_i$ sampled from $\pi(x_i)$ and the greedy output $y_i^{grd}$: $D(y_i, y_i^{grd})$. The against-greedy SBERT diversity requires computing the latent vectors of both outputs and returning 1 minus their cosine similarity. In the case of against-greedy EAD diversity, however, we replace the classic ratio of distinct $N$-gram computation with the ratio of *new*, distinct $N$-gram from $y_i$ with respect to $y_i^{grd}$, and then compute EAD for $N \in [1, 5]$ starting from it.

- **Per-Input Diversity:** The diversity of the output set over a specific input, i.e., the diversity of $\pi(y|x)$: $D(\{y_{i,1} \ldots y_{i,M}\}), y_{i,j} \sim \pi(\cdot|x_i)$ (Kirk et al., 2024). In the case of per-input EAD diversity, this is implemented by computing the EAD score for the set of outputs from the same input (i.e., the expected average distinct $N$-grams among all the sequences) for $N \in [1, 5]$ and returning their average. In the case of the per-input SBERT diversity, this is obtained by first encoding the outputs into latent vectors through a given text encoder, then computing all the possible pair-wise cosine similarity between different vectors, and finally returning 1 minus the average cosine similarity.

- **Accuracy:** The percentage of correctly solved problems, i.e., given the count of solved problems $C_{correct}$ and the count of total problems $C_{total}$, accuracy is defined as $100 \cdot \frac{C_{correct}}{C_{total}}$.

- **Rouge-1 (R1):** The ratio of 1-grams present in both the target $y_i^*$ and the generated output $y_i$ (Lin, 2004).

- **Sentence Embedding Cosine Similarity (SIM):** The cosine similarity between the latent vector $v_{y_i}$ and the latent vector $v_{y_i^*}$, where $y_i^*$ is the target output and $y_i$ is the generated output. The latent vectors are obtained from a pre-trained text encoder.

- **Coherence (COH):** The cosine similarity between the latent vector $v_{s_i}$ and the latent vector $v_{y_i}$, where $s_i$ represents a target text (e.g., a text to summarize) or the input passed to the model, and $y_i$ is the output from the model (e.g., the continuation of $s_i$ or its summary). The latent vectors are obtained from the SimCSE embedder (Gao et al., 2021).

- **DAT Score:** The average cosine distance (i.e., 1 minus cosine similarity) multiplied by 100 between each pair of word embeddings from a given list of $n$ distinct nouns. The word embeddings are obtained from GloVe (Pennington et al., 2014). $n$ can be either 7 or 10. If there are not enough distinct nouns, the score is not computed at all.

- **DAT Valid Outputs:** The percentage of experiments in which the count of generated, distinct nouns is greater than or equal to $n$, i.e., the percentage of experiments where it is possible to compute the DAT score.

# D  Prompts

For the mathematical problem-solving tasks, we adopted the same prompt from Yu et al. (2024), i.e.:

> Below is an instruction that describes a task. Write a response that appropriately completes the request.
>
> ### Instruction:
> {question}
>
> ### Response: Let's think step by step.

For the extreme summarization task, the prompt adopted for `Llama-3.2-3B-Instruct` is the same as in Chhabra et al. (2024):

> user
> For the following article: {`article`}
>
> Return a summary comprising of 1 sentence. Write the sentence in a numbered list format.
>
> For example:
>
> 1. First sentence
> assistant

where `user` and `assistant` are special tokens used by the model to identify different roles in the chat.

Vice versa, for the non-instructed version, we used:

> Generate a 1 sentence summary for the given article.
> Article: {`article`}
> Summary:

For the divergent association task, we considered the following prompt for `Meta-Llama-3-8B-Instruct`:

> user
>
> Please write 10 nouns in English that are as irrelevant from each other as possible, in all meanings and uses of the words. Please note that the words you write should have only single word, only nouns (e.g., things, objects, concepts), and no proper nouns (e.g., no specific people or places).
> assistant
>
> Here are the 10 nouns in English that are as irrelevant from each other as possible:

where `user` and `assistant` are keywords used by the model to identify different roles in the chat, while for `Meta-Llama-3-8B` we used the following:

> Write 10 nouns in English that are as irrelevant from each other as possible, in all meanings and uses of the words. Please note that the words you write should have only single word, only nouns (e.g., things, objects, concepts), and no proper nouns (e.g., no specific people or places).
>
> Solution:
> Here are the 10 nouns in English that are as irrelevant from each other as possible:

Finally, for story generation, we used the same prompt adopted by Chung et al. (2025) in the case of `Llama-3.2-3B-Instruct`:

> system
> You write a creative writing based on the user-given writing prompt.
> user
> {`prompt`}
> assistant

where `system`, `user`, and `assistant` are keywords used by the model to identify different roles in the chat. Instead, for `Llama-3.2-3B` we used the following:

> Write a creative story based on the user-given prompt.
> Prompt: {prompt}
> Story:

# E  Scaling Model Size

Our main experiments focus only on models of limited size (i.e., 3B, 7B, and 8B models). To demonstrate that our sampling methods can scale up with model size, we also conduct some preliminary tests with a larger model, i.e., with `Meta-Llama-3-70B` (quantized to 4-bit precision), considering both its pre-trained and its instructed versions. In particular, we repeat all the experiments conducted in Section 5 at a fixed temperature $\tau = 1.0$.

Tables 6 and 7 report the results for the math problem-solving case study. As for the smaller model, *DiffSampling-cut* performs on par or better than greedy in terms of accuracy, while achieving a higher diversity than greedy but lower than the other sampling methods. *DiffSampling-lb* achieves slightly worse accuracy than top-$p$, but with increases in cross-input EAD. Finally, there are no substantial differences between *DiffSampling-minp* and min-$p$ apart from cross-input EAD, thus confirming that our relaxation allows for a little more diversity with no cost in terms of quality.

| Dataset: | RLHF-instructed | | | | | Pre-trained | | | | |
|---|---|---|---|---|---|---|---|---|---|---|
| Method | Accuracy | Cross-Input | | Against-Greedy | | Accuracy | Cross-Input | | Against-Greedy | |
| | | EAD | SBERT | EAD | SBERT | | EAD | SBERT | EAD | SBERT |
| Greedy | $57.62_{\pm.00}$ | $0.75_{\pm.00}$ | $0.65_{\pm.00}$ | - | - | $10.16_{\pm.00}$ | $0.97_{\pm.00}$ | $0.56_{\pm.00}$ | - | - |
| Top-$p$ | $53.85_{\pm.20}$ | $0.88_{\pm.00}$ | $0.65_{\pm.00}$ | $0.56_{\pm.00}$ | $0.11_{\pm.00}$ | $7.83_{\pm.14}$ | $2.00_{\pm.01}$ | $0.59_{\pm.00}$ | $0.63_{\pm.00}$ | $0.31_{\pm.00}$ |
| $\eta$-sampling | $48.37_{\pm.28}$ | $0.92_{\pm.00}$ | $0.65_{\pm.00}$ | $0.60_{\pm.00}$ | $0.12_{\pm.00}$ | $4.88_{\pm.11}$ | $2.22_{\pm.01}$ | $0.60_{\pm.00}$ | $0.70_{\pm.00}$ | $0.33_{\pm.00}$ |
| Locally typical | $53.68_{\pm.16}$ | $0.88_{\pm.00}$ | $0.65_{\pm.00}$ | $0.56_{\pm.00}$ | $0.11_{\pm.00}$ | $7.43_{\pm.21}$ | $2.00_{\pm.00}$ | $0.59_{\pm.00}$ | $0.63_{\pm.00}$ | $0.32_{\pm.00}$ |
| Min-$p$ | $56.20_{\pm.58}$ | $0.85_{\pm.00}$ | $0.65_{\pm.00}$ | $0.54_{\pm.00}$ | $0.10_{\pm.00}$ | $9.12_{\pm.93}$ | $1.52_{\pm.00}$ | $0.56_{\pm.00}$ | $0.51_{\pm.01}$ | $0.28_{\pm.00}$ |
| DiffS.-cut | $59.29_{\pm.63}$ | $0.77_{\pm.00}$ | $0.65_{\pm.00}$ | $0.39_{\pm.01}$ | $0.08_{\pm.00}$ | $12.69_{\pm.20}$ | $0.97_{\pm.00}$ | $0.57_{\pm.00}$ | $0.26_{\pm.01}$ | $0.18_{\pm.00}$ |
| DiffS.-lb | $52.14_{\pm.60}$ | $0.89_{\pm.00}$ | $0.65_{\pm.00}$ | $0.56_{\pm.00}$ | $0.11_{\pm.00}$ | $7.25_{\pm.42}$ | $2.03_{\pm.01}$ | $0.59_{\pm.00}$ | $0.64_{\pm.00}$ | $0.32_{\pm.00}$ |
| DiffS.-minp | $56.00_{\pm.18}$ | $0.85_{\pm.00}$ | $0.65_{\pm.00}$ | $0.54_{\pm.00}$ | $0.10_{\pm.00}$ | $9.93_{\pm.14}$ | $1.56_{\pm.01}$ | $0.56_{\pm.00}$ | $0.51_{\pm.01}$ | $0.28_{\pm.00}$ |

Table 6: Accuracy and diversity of results for the GSM8K test set over 3 seeds for the instructed (left) and pre-trained (right) `Meta-Llama-3-70B` model. The mean and standard error of the final score for each run are reported for accuracy and cross-input diversity, whereas the mean and the 95% confidence interval for the full set of answers are reported for against-greedy diversity.

| Model: | RLHF-instructed | | | | | Pre-trained | | | | |
|---|---|---|---|---|---|---|---|---|---|---|
| Method | Accuracy | Cross-Input | | Against-Greedy | | Accuracy | Cross-Input | | Against-Greedy | |
| | | EAD | SBERT | EAD | SBERT | | EAD | SBERT | EAD | SBERT |
| Greedy | $22.56_{\pm.00}$ | $2.01_{\pm.00}$ | $0.80_{\pm.00}$ | - | - | $6.70_{\pm.00}$ | $2.18_{\pm.00}$ | $0.78_{\pm.00}$ | - | - |
| Top-$p$ | $20.39_{\pm.15}$ | $2.56_{\pm.01}$ | $0.79_{\pm.00}$ | $0.55_{\pm.00}$ | $0.17_{\pm.00}$ | $4.04_{\pm.15}$ | $5.12_{\pm.02}$ | $0.79_{\pm.00}$ | $0.57_{\pm.00}$ | $0.34_{\pm.00}$ |
| $\eta$-sampling | $19.02_{\pm.13}$ | $2.80_{\pm.01}$ | $0.79_{\pm.00}$ | $0.58_{\pm.00}$ | $0.18_{\pm.00}$ | $3.09_{\pm.09}$ | $6.06_{\pm.02}$ | $0.79_{\pm.00}$ | $0.64_{\pm.00}$ | $0.37_{\pm.00}$ |
| Locally typical | $20.19_{\pm.20}$ | $2.56_{\pm.00}$ | $0.79_{\pm.00}$ | $0.55_{\pm.00}$ | $0.17_{\pm.00}$ | $4.12_{\pm.18}$ | $5.15_{\pm.02}$ | $0.79_{\pm.00}$ | $0.57_{\pm.00}$ | $0.34_{\pm.00}$ |
| Min-$p$ | $20.58_{\pm.04}$ | $2.41_{\pm.00}$ | $0.79_{\pm.00}$ | $0.53_{\pm.00}$ | $0.17_{\pm.00}$ | $5.46_{\pm.13}$ | $3.73_{\pm.01}$ | $0.78_{\pm.00}$ | $0.45_{\pm.00}$ | $0.30_{\pm.00}$ |
| DiffS.-cut | $21.27_{\pm.03}$ | $2.08_{\pm.00}$ | $0.80_{\pm.00}$ | $0.40_{\pm.00}$ | $0.14_{\pm.00}$ | $7.50_{\pm.18}$ | $2.29_{\pm.01}$ | $0.78_{\pm.00}$ | $0.25_{\pm.00}$ | $0.22_{\pm.00}$ |
| DiffS.-lb | $19.79_{\pm.40}$ | $2.58_{\pm.01}$ | $0.79_{\pm.00}$ | $0.55_{\pm.00}$ | $0.18_{\pm.00}$ | $4.15_{\pm.03}$ | $5.22_{\pm.00}$ | $0.79_{\pm.00}$ | $0.57_{\pm.00}$ | $0.34_{\pm.00}$ |
| DiffS.-minp | $20.75_{\pm.56}$ | $2.45_{\pm.00}$ | $0.79_{\pm.00}$ | $0.53_{\pm.00}$ | $0.17_{\pm.00}$ | $5.26_{\pm.21}$ | $3.82_{\pm.01}$ | $0.78_{\pm.00}$ | $0.45_{\pm.00}$ | $0.30_{\pm.00}$ |

Table 7: Accuracy and diversity of results for the MATH test set over 3 seeds for the instructed (left) and pre-trained (right) `Meta-Llama-3-70B` model. The mean and standard error of the final score for each run are reported for accuracy and cross-input diversity, whereas the mean and the 95% confidence interval for the full set of answers are reported for against-greedy diversity.

The findings from the XSum case study are similar. As reported in Table 8, *DiffSampling-cut* behaves on par with greedy in terms of quality, but increases scores in terms of per-input diversity. Instead, our two relaxations do not substantially differ from their corresponding baselines top-$p$ and min-$p$. However, it is remarkable that, contrary to what we found for smaller models, greediness does not provide qualitative advantages for the instructed version of `Meta-Llama-3-70B`.

| Model: | RLHF-instructed | | | | | | | Pre-trained | | | | | | |
|---|---|---|---|---|---|---|---|---|---|---|---|---|---|---|
| Method | Quality | | | Per-Input | | Against-Greedy | | Quality | | | Per-Input | | Against-Greedy | |
| | R-1 | SIM | COH | EAD | SBERT | EAD | SBERT | R-1 | SIM | COH | EAD | SBERT | EAD | SBERT |
| Greedy | $0.29_{\pm.00}$ | $0.63_{\pm.00}$ | $0.75_{\pm.00}$ | $0.20_{\pm.00}$ | - | - | - | $0.24_{\pm.00}$ | $0.50_{\pm.01}$ | $0.70_{\pm.00}$ | $0.20_{\pm.00}$ | - | - | - |
| Top-$p$ | $0.29_{\pm.00}$ | $0.62_{\pm.00}$ | $0.75_{\pm.00}$ | $0.50_{\pm.00}$ | $0.10_{\pm.00}$ | $0.44_{\pm.01}$ | $0.08_{\pm.00}$ | $0.22_{\pm.00}$ | $0.48_{\pm.01}$ | $0.63_{\pm.01}$ | $0.72_{\pm.01}$ | $0.49_{\pm.00}$ | $0.71_{\pm.01}$ | $0.41_{\pm.01}$ |
| $\eta$-sampling | $0.29_{\pm.00}$ | $0.62_{\pm.00}$ | $0.75_{\pm.00}$ | $0.55_{\pm.00}$ | $0.11_{\pm.00}$ | $0.49_{\pm.01}$ | $0.09_{\pm.00}$ | $0.21_{\pm.00}$ | $0.47_{\pm.01}$ | $0.61_{\pm.01}$ | $0.76_{\pm.01}$ | $0.52_{\pm.00}$ | $0.75_{\pm.01}$ | $0.44_{\pm.01}$ |
| Locally typical | $0.29_{\pm.00}$ | $0.62_{\pm.00}$ | $0.75_{\pm.00}$ | $0.50_{\pm.00}$ | $0.10_{\pm.00}$ | $0.44_{\pm.01}$ | $0.08_{\pm.00}$ | $0.22_{\pm.00}$ | $0.48_{\pm.01}$ | $0.63_{\pm.01}$ | $0.72_{\pm.01}$ | $0.49_{\pm.00}$ | $0.71_{\pm.01}$ | $0.42_{\pm.01}$ |
| Min-$p$ | $0.29_{\pm.00}$ | $0.62_{\pm.00}$ | $0.75_{\pm.00}$ | $0.50_{\pm.00}$ | $0.10_{\pm.00}$ | $0.44_{\pm.01}$ | $0.08_{\pm.00}$ | $0.23_{\pm.00}$ | $0.50_{\pm.01}$ | $0.67_{\pm.01}$ | $0.65_{\pm.00}$ | $0.42_{\pm.00}$ | $0.63_{\pm.01}$ | $0.35_{\pm.01}$ |
| DiffS.-cut | $0.29_{\pm.00}$ | $0.63_{\pm.00}$ | $0.75_{\pm.00}$ | $0.33_{\pm.00}$ | $0.05_{\pm.00}$ | $0.25_{\pm.01}$ | $0.05_{\pm.00}$ | $0.24_{\pm.00}$ | $0.50_{\pm.01}$ | $0.70_{\pm.00}$ | $0.38_{\pm.00}$ | $0.18_{\pm.00}$ | $0.32_{\pm.01}$ | $0.16_{\pm.01}$ |
| DiffS.-lb | $0.29_{\pm.00}$ | $0.63_{\pm.00}$ | $0.75_{\pm.00}$ | $0.51_{\pm.00}$ | $0.10_{\pm.00}$ | $0.45_{\pm.01}$ | $0.08_{\pm.00}$ | $0.22_{\pm.00}$ | $0.48_{\pm.01}$ | $0.63_{\pm.01}$ | $0.73_{\pm.01}$ | $0.50_{\pm.00}$ | $0.72_{\pm.01}$ | $0.42_{\pm.01}$ |
| DiffS.-minp | $0.29_{\pm.00}$ | $0.63_{\pm.00}$ | $0.75_{\pm.00}$ | $0.50_{\pm.00}$ | $0.10_{\pm.00}$ | $0.44_{\pm.01}$ | $0.08_{\pm.00}$ | $0.23_{\pm.00}$ | $0.50_{\pm.01}$ | $0.66_{\pm.01}$ | $0.65_{\pm.00}$ | $0.42_{\pm.01}$ | $0.63_{\pm.01}$ | $0.35_{\pm.01}$ |

Table 8: Aggregate results over 5 outputs sampled for each of the 1000 prompts from the XSum dataset for the instructed (left) and the pre-trained (right) `Meta-Llama-3-70B` model. The mean and 95% confidence interval are reported for all the metrics.

The results are similar for the WritingPrompts dataset. As shown in Table 9, the greedy strategy is not optimal for the instructed model; this impacts not only *DiffSampling-cut*, but also min-*p* and *DiffSampling-minp*. However, the other considerations still hold: *DiffSampling-cut* produces more diverse outputs than the greedy strategy, while the two relaxations perform on par with their corresponding baselines. While greediness appears to guarantee higher quality for the pre-trained model, it should be noted that the per-input EAD becomes extremely low. This is due to extensive repetitions within the same output (but not shared across outputs, which causes the SBERT metric to increase) and may indicate that more greedy strategies perform worse with larger models, which may have learned more unbalanced probability distributions. In such scenarios, looser sampling strategies, such as $\eta$-sampling or our *DiffSampling-lb*, are likely to be better choices.

| Model: | RLHF-instructed | | | Pre-trained | | |
|---|---|---|---|---|---|---|
| Method | Quality | Per-Input Diversity | | Quality | Per-Input Diversity | |
| | COH | EAD | SBERT | COH | EAD | SBERT |
| Greedy | $0.27_{\pm.01}$ | $0.06_{\pm.00}$ | - | $0.42_{\pm.01}$ | $0.01_{\pm.00}$ | - |
| Top-$p$ | $0.29_{\pm.01}$ | $0.53_{\pm.00}$ | $0.58_{\pm.01}$ | $0.26_{\pm.01}$ | $0.18_{\pm.01}$ | $0.72_{\pm.00}$ |
| $\eta$-sampling | $0.29_{\pm.01}$ | $0.58_{\pm.00}$ | $0.57_{\pm.01}$ | $0.26_{\pm.01}$ | $0.35_{\pm.01}$ | $0.71_{\pm.00}$ |
| Locally typical | $0.29_{\pm.01}$ | $0.59_{\pm.00}$ | $0.46_{\pm.01}$ | $0.26_{\pm.01}$ | $0.18_{\pm.00}$ | $0.72_{\pm.00}$ |
| Min-$p$ | $0.27_{\pm.01}$ | $0.45_{\pm.01}$ | $0.55_{\pm.01}$ | $0.34_{\pm.01}$ | $0.03_{\pm.00}$ | $0.60_{\pm.00}$ |
| DiffS.-cut | $0.27_{\pm.01}$ | $0.23_{\pm.01}$ | $0.24_{\pm.01}$ | $0.41_{\pm.01}$ | $0.01_{\pm.00}$ | $0.32_{\pm.01}$ |
| DiffS.-lb | $0.29_{\pm.01}$ | $0.53_{\pm.00}$ | $0.57_{\pm.01}$ | $0.25_{\pm.01}$ | $0.19_{\pm.01}$ | $0.72_{\pm.00}$ |
| DiffS.-minp | $0.27_{\pm.01}$ | $0.44_{\pm.01}$ | $0.55_{\pm.01}$ | $0.34_{\pm.01}$ | $0.03_{\pm.00}$ | $0.60_{\pm.00}$ |

Table 9: Aggregate results for the WritingPrompts dataset for the instructed (left) and the pre-trained (right) `Meta-Llama-3-70B` model. The mean and the 95% confidence interval for the full set of answers are reported for all the metrics.

This consideration finds confirmation from the experiments on the divergent association task. As shown in Fig. 4, *DiffSampling-cut* always produces the greedy solution for the instructed model, while showing a little more variation for the pre-trained model. Again, both min-*p* and *DiffSampling-minp* are influenced by the poor performances of greedy for the pre-trained model, resulting in lower DAT score but higher valid output ratio compared with the other sampling strategies. On the contrary, top-*p* sampling, locally typical sampling, and *DiffSampling-lb* seem to better trade off quality and diversity, having a comparably high valid output ratio for the instructed model and especially a substantially higher DAT score for the pre-trained model.

In summary, our preliminary experiments on a larger model confirm the positive relation between our three methods and the corresponding baselines, while also demonstrating that models with different sizes may necessitate different degrees of greediness and thus suggesting a looser sampling strategy (e.g., *DiffSampling-lb*) when dealing with sufficiently large language models.

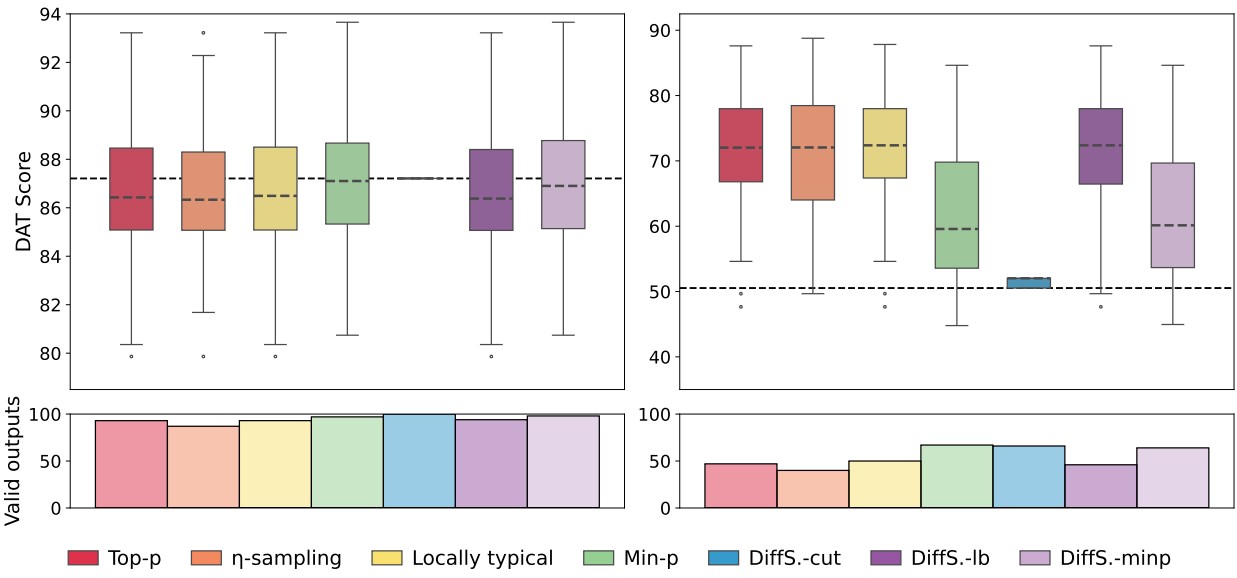

Figure 4: DAT scores for our methods and the baselines over the instructed (left) and pre-trained (right) `Meta-Llama-3-70B` model. Below, the number of valid outputs produced by each sampling strategy. The dashed line reports the greedy score.

## F    Experiments on Temperature Scaling

In addition to investigating performance at a temperature of $\tau = 1.0$, we conduct experiments at a lower temperature (0.6) and three higher temperatures (1.5, 2.0, and 10.0) to verify whether our truncation strategy preserves only appropriate tokens, i.e., whether the quality of generated outputs remains competitive and diversity improves across different temperatures. Overall, we found that *DiffSampling-cut* maintains the same level of quality even at very high temperature, and, thus, can be safely used even with $\tau \gg 1.0$. On the other hand, *DiffSampling-lb* tends to produce lower-quality outputs at higher temperatures. This effect is particularly pronounced for non-instructed models, where performance for $\tau > 1.0$ decreases rapidly. However, fine-tuned models maintain competitive quality scores at higher temperatures while increasing diversity; the choice of $\tau$ is a matter of trading off quality and diversity. When accuracy is not a *sine qua non* requirement, we suggest adopting a $\tau \geq 2.0$. Finally, *DiffSampling-minp* exhibits the same general trend as *DiffSampling-lb* for pre-trained models, although the effect is less pronounced. For tuned models, however, it shows virtually no performance degradation at higher temperatures, enabling the use of larger values of $\tau$ in most scenarios.

### F.1    Math Problem Solving

Table 10 reports all the results with different temperatures for the GSM8K (left side) and MATH (right side) test sets. For the former, a lower temperature makes all the models (including the baselines) more in line with the greedy strategy, thus diminishing the diversity scores while usually increasing the accuracy. On the contrary, all the baselines tend to perform poorer at increasing temperatures in terms of output correctness, while diversity improves accordingly (especially for a syntactic-based metric such as EAD; the qualitative examples reported below demonstrate why). Instead, our methods maintain the highest possible accuracy, with a slight improvement in diversity at higher $\tau$.

For the latter, a lower temperature makes all the baselines closer to our methods in terms of accuracy, while diminishing their diversity scores. At increasing temperature, the baselines rapidly start failing to solve the problems, possibly due to a more random selection of tokens that also causes syntactic diversity to increase. By applying temperature after the truncation, our methods preserve their output quality regardless of the

| Dataset: | GSM8K | | | | | MATH | | | | |
|---|---|---|---|---|---|---|---|---|---|---|
| Method | Accuracy | Cross-Input | | Against-Greedy | | Accuracy | Cross-Input | | Against-Greedy | |
| | | EAD | SBERT | EAD | SBERT | | EAD | SBERT | EAD | SBERT |
| *Temperature = 0.0* | | | | | | | | | | |
| Greedy | $66.44_{\pm.09}$ | $2.03_{\pm.00}$ | $0.64_{\pm.00}$ | - | - | $20.62_{\pm.01}$ | $5.65_{\pm.00}$ | $0.80_{\pm.00}$ | - | - |
| *Temperature = 1.0* | | | | | | | | | | |
| Top-$p$ | $65.00_{\pm.18}$ | $2.08_{\pm.01}$ | $0.64_{\pm.00}$ | $0.23_{\pm.00}$ | $0.03_{\pm.00}$ | $20.02_{\pm.12}$ | $6.08_{\pm.02}$ | $0.80_{\pm.00}$ | $0.36_{\pm.00}$ | $0.10_{\pm.00}$ |
| $\eta$-sampling | $65.05_{\pm.19}$ | $2.12_{\pm.00}$ | $0.64_{\pm.00}$ | $0.25_{\pm.00}$ | $0.04_{\pm.00}$ | $19.67_{\pm.20}$ | $6.36_{\pm.01}$ | $0.80_{\pm.00}$ | $0.39_{\pm.00}$ | $0.11_{\pm.00}$ |
| Locally typical | $66.29_{\pm.55}$ | $2.09_{\pm.00}$ | $0.64_{\pm.00}$ | $0.23_{\pm.00}$ | $0.03_{\pm.00}$ | $19.95_{\pm.26}$ | $6.06_{\pm.01}$ | $0.80_{\pm.00}$ | $0.36_{\pm.00}$ | $0.10_{\pm.00}$ |
| Min-$p$ | $65.76_{\pm.44}$ | $2.09_{\pm.00}$ | $0.64_{\pm.00}$ | $0.23_{\pm.00}$ | $0.03_{\pm.00}$ | $20.25_{\pm.09}$ | $6.09_{\pm.01}$ | $0.80_{\pm.00}$ | $0.36_{\pm.00}$ | $0.10_{\pm.00}$ |
| DiffS.-cut | $66.36_{\pm.23}$ | $2.04_{\pm.00}$ | $0.64_{\pm.00}$ | $0.14_{\pm.00}$ | $0.02_{\pm.00}$ | $21.38_{\pm.20}$ | $5.71_{\pm.01}$ | $0.80_{\pm.00}$ | $0.27_{\pm.00}$ | $0.07_{\pm.00}$ |
| DiffS.-lb | $65.18_{\pm.65}$ | $2.09_{\pm.01}$ | $0.64_{\pm.00}$ | $0.23_{\pm.00}$ | $0.03_{\pm.00}$ | $20.20_{\pm.08}$ | $6.11_{\pm.02}$ | $0.80_{\pm.00}$ | $0.37_{\pm.00}$ | $0.10_{\pm.00}$ |
| DiffS.-minp | $65.48_{\pm.60}$ | $2.09_{\pm.01}$ | $0.64_{\pm.00}$ | $0.23_{\pm.00}$ | $0.03_{\pm.00}$ | $20.18_{\pm.08}$ | $6.06_{\pm.00}$ | $0.80_{\pm.00}$ | $0.36_{\pm.00}$ | $0.10_{\pm.00}$ |
| *Temperature = 0.6* | | | | | | | | | | |
| Top-$p$ | $66.34_{\pm.67}$ | $2.05_{\pm.01}$ | $0.64_{\pm.00}$ | $0.17_{\pm.00}$ | $0.02_{\pm.00}$ | $21.58_{\pm.32}$ | $5.81_{\pm.02}$ | $0.80_{\pm.00}$ | $0.31_{\pm.00}$ | $0.09_{\pm.00}$ |
| $\eta$-sampling | $66.26_{\pm.22}$ | $2.07_{\pm.01}$ | $0.64_{\pm.00}$ | $0.19_{\pm.00}$ | $0.03_{\pm.00}$ | $20.36_{\pm.15}$ | $5.94_{\pm.01}$ | $0.80_{\pm.00}$ | $0.33_{\pm.00}$ | $0.09_{\pm.00}$ |
| Locally typical | $66.34_{\pm.67}$ | $2.05_{\pm.01}$ | $0.64_{\pm.00}$ | $0.17_{\pm.00}$ | $0.02_{\pm.00}$ | $21.58_{\pm.32}$ | $5.81_{\pm.02}$ | $0.80_{\pm.00}$ | $0.31_{\pm.00}$ | $0.09_{\pm.00}$ |
| Min-$p$ | $66.52_{\pm.30}$ | $2.06_{\pm.01}$ | $0.64_{\pm.00}$ | $0.17_{\pm.00}$ | $0.02_{\pm.00}$ | $21.31_{\pm.08}$ | $5.81_{\pm.01}$ | $0.80_{\pm.00}$ | $0.31_{\pm.00}$ | $0.09_{\pm.00}$ |
| DiffS.-cut | $66.74_{\pm.04}$ | $2.05_{\pm.00}$ | $0.64_{\pm.00}$ | $0.13_{\pm.00}$ | $0.02_{\pm.00}$ | $21.52_{\pm.13}$ | $5.72_{\pm.00}$ | $0.80_{\pm.00}$ | $0.25_{\pm.00}$ | $0.07_{\pm.00}$ |
| DiffS.-lb | $65.73_{\pm.23}$ | $2.06_{\pm.00}$ | $0.64_{\pm.00}$ | $0.19_{\pm.00}$ | $0.03_{\pm.00}$ | $20.65_{\pm.20}$ | $5.89_{\pm.01}$ | $0.80_{\pm.00}$ | $0.32_{\pm.00}$ | $0.09_{\pm.00}$ |
| DiffS.-minp | $67.05_{\pm.14}$ | $2.06_{\pm.01}$ | $0.64_{\pm.00}$ | $0.19_{\pm.00}$ | $0.03_{\pm.00}$ | $20.56_{\pm.21}$ | $5.88_{\pm.00}$ | $0.80_{\pm.00}$ | $0.32_{\pm.00}$ | $0.09_{\pm.00}$ |
| *Temperature = 1.5* | | | | | | | | | | |
| Top-$p$ | $63.91_{\pm.57}$ | $2.17_{\pm.01}$ | $0.64_{\pm.00}$ | $0.28_{\pm.00}$ | $0.04_{\pm.00}$ | $18.38_{\pm.22}$ | $6.92_{\pm.02}$ | $0.80_{\pm.00}$ | $0.42_{\pm.00}$ | $0.12_{\pm.00}$ |
| $\eta$-sampling | $60.35_{\pm.55}$ | $2.28_{\pm.00}$ | $0.64_{\pm.00}$ | $0.32_{\pm.00}$ | $0.05_{\pm.00}$ | $15.63_{\pm.17}$ | $7.77_{\pm.01}$ | $0.80_{\pm.00}$ | $0.45_{\pm.00}$ | $0.14_{\pm.00}$ |
| Locally typical | $64.39_{\pm.41}$ | $2.17_{\pm.01}$ | $0.64_{\pm.00}$ | $0.28_{\pm.00}$ | $0.04_{\pm.00}$ | $18.73_{\pm.01}$ | $7.04_{\pm.01}$ | $0.80_{\pm.00}$ | $0.42_{\pm.00}$ | $0.12_{\pm.00}$ |
| Min-$p$ | $64.29_{\pm.38}$ | $2.15_{\pm.00}$ | $0.64_{\pm.00}$ | $0.28_{\pm.00}$ | $0.04_{\pm.00}$ | $18.94_{\pm.23}$ | $6.54_{\pm.02}$ | $0.80_{\pm.00}$ | $0.40_{\pm.00}$ | $0.12_{\pm.00}$ |
| DiffS.-cut | $66.72_{\pm.36}$ | $2.05_{\pm.00}$ | $0.64_{\pm.00}$ | $0.15_{\pm.00}$ | $0.02_{\pm.00}$ | $21.36_{\pm.15}$ | $5.73_{\pm.00}$ | $0.80_{\pm.00}$ | $0.27_{\pm.00}$ | $0.07_{\pm.00}$ |
| DiffS.-lb | $65.20_{\pm.25}$ | $2.11_{\pm.01}$ | $0.64_{\pm.00}$ | $0.25_{\pm.00}$ | $0.04_{\pm.00}$ | $19.55_{\pm.03}$ | $6.31_{\pm.02}$ | $0.80_{\pm.00}$ | $0.39_{\pm.00}$ | $0.11_{\pm.00}$ |
| DiffS.-minp | $65.55_{\pm.61}$ | $2.11_{\pm.00}$ | $0.64_{\pm.00}$ | $0.25_{\pm.00}$ | $0.04_{\pm.00}$ | $20.04_{\pm.13}$ | $6.19_{\pm.01}$ | $0.80_{\pm.00}$ | $0.38_{\pm.00}$ | $0.11_{\pm.00}$ |
| *Temperature = 2.0* | | | | | | | | | | |
| Top-$p$ | $25.40_{\pm.07}$ | $10.13_{\pm.10}$ | $0.66_{\pm.00}$ | $0.70_{\pm.01}$ | $0.36_{\pm.01}$ | $2.49_{\pm.01}$ | $48.71_{\pm.08}$ | $0.52_{\pm.00}$ | $0.92_{\pm.00}$ | $0.68_{\pm.00}$ |
| $\eta$-sampling | $35.51_{\pm.30}$ | $7.35_{\pm.05}$ | $0.69_{\pm.00}$ | $0.58_{\pm.01}$ | $0.22_{\pm.01}$ | $4.26_{\pm.06}$ | $43.39_{\pm.10}$ | $0.64_{\pm.00}$ | $0.86_{\pm.00}$ | $0.53_{\pm.00}$ |
| Locally typical | $24.61_{\pm.60}$ | $10.65_{\pm.05}$ | $0.65_{\pm.00}$ | $0.71_{\pm.01}$ | $0.37_{\pm.01}$ | $2.46_{\pm.03}$ | $51.04_{\pm.07}$ | $0.50_{\pm.00}$ | $0.93_{\pm.00}$ | $0.69_{\pm.00}$ |
| Min-$p$ | $62.19_{\pm.37}$ | $2.24_{\pm.01}$ | $0.64_{\pm.00}$ | $0.32_{\pm.00}$ | $0.05_{\pm.00}$ | $16.92_{\pm.21}$ | $7.21_{\pm.01}$ | $0.80_{\pm.00}$ | $0.44_{\pm.00}$ | $0.13_{\pm.00}$ |
| DiffS.-cut | $66.44_{\pm.18}$ | $2.05_{\pm.00}$ | $0.64_{\pm.00}$ | $0.15_{\pm.00}$ | $0.02_{\pm.00}$ | $21.66_{\pm.20}$ | $5.71_{\pm.01}$ | $0.80_{\pm.00}$ | $0.27_{\pm.00}$ | $0.08_{\pm.00}$ |
| DiffS.-lb | $63.48_{\pm.43}$ | $2.12_{\pm.00}$ | $0.64_{\pm.00}$ | $0.26_{\pm.00}$ | $0.04_{\pm.00}$ | $19.17_{\pm.10}$ | $6.40_{\pm.02}$ | $0.80_{\pm.00}$ | $0.40_{\pm.00}$ | $0.12_{\pm.00}$ |
| DiffS.-minp | $65.13_{\pm.28}$ | $2.12_{\pm.01}$ | $0.64_{\pm.00}$ | $0.26_{\pm.00}$ | $0.04_{\pm.00}$ | $19.70_{\pm.09}$ | $6.32_{\pm.02}$ | $0.80_{\pm.00}$ | $0.39_{\pm.00}$ | $0.11_{\pm.00}$ |
| *Temperature = 10.0* | | | | | | | | | | |
| Top-$p$ | $0.00_{\pm.00}$ | $17.26_{\pm.03}$ | $0.12_{\pm.00}$ | $1.00_{\pm.00}$ | $0.96_{\pm.00}$ | $0.00_{\pm.00}$ | $58.65_{\pm.03}$ | $0.12_{\pm.00}$ | $1.00_{\pm.00}$ | $1.00_{\pm.00}$ |
| $\eta$-sampling | $0.00_{\pm.00}$ | $17.43_{\pm.04}$ | $0.12_{\pm.00}$ | $1.00_{\pm.00}$ | $0.96_{\pm.00}$ | $0.00_{\pm.00}$ | $59.18_{\pm.02}$ | $0.12_{\pm.00}$ | $1.00_{\pm.00}$ | $1.00_{\pm.00}$ |
| Locally typical | $0.00_{\pm.00}$ | $17.52_{\pm.01}$ | $0.11_{\pm.00}$ | $1.01_{\pm.00}$ | $0.96_{\pm.00}$ | $0.00_{\pm.00}$ | $59.69_{\pm.01}$ | $0.11_{\pm.00}$ | $1.01_{\pm.00}$ | $1.00_{\pm.00}$ |
| Min-$p$ | $0.00_{\pm.00}$ | $17.39_{\pm.04}$ | $0.13_{\pm.00}$ | $1.00_{\pm.00}$ | $0.95_{\pm.00}$ | $0.00_{\pm.00}$ | $59.16_{\pm.02}$ | $0.13_{\pm.00}$ | $1.00_{\pm.00}$ | $1.00_{\pm.00}$ |
| DiffS.-cut | $66.31_{\pm.26}$ | $2.04_{\pm.00}$ | $0.64_{\pm.00}$ | $0.15_{\pm.00}$ | $0.02_{\pm.00}$ | $21.22_{\pm.11}$ | $5.74_{\pm.01}$ | $0.80_{\pm.00}$ | $0.28_{\pm.00}$ | $0.08_{\pm.00}$ |
| DiffS.-lb | $64.11_{\pm.13}$ | $2.15_{\pm.00}$ | $0.64_{\pm.00}$ | $0.29_{\pm.00}$ | $0.04_{\pm.00}$ | $18.20_{\pm.07}$ | $6.72_{\pm.00}$ | $0.80_{\pm.00}$ | $0.42_{\pm.00}$ | $0.12_{\pm.00}$ |
| DiffS.-minp | $63.58_{\pm.43}$ | $2.17_{\pm.01}$ | $0.64_{\pm.00}$ | $0.29_{\pm.00}$ | $0.04_{\pm.00}$ | $18.64_{\pm.20}$ | $6.54_{\pm.01}$ | $0.80_{\pm.00}$ | $0.41_{\pm.00}$ | $0.12_{\pm.00}$ |

Table 10: Accuracy and diversity of results for the GSM8K and MATH test sets over 3 seeds with different temperature values. The mean and standard error of the final score for each run are reported for accuracy and cross-input diversity, whereas the mean and 95% confidence interval for the full set of answers are reported for against-greedy diversity.

temperature used, with small but relevant gains in diversity (for example, *DiffSampling-minp* at $\tau = 2.0$ has an accuracy comparable with min-$p$ at $\tau = 1.0$, but with higher diversity scores).

## F.2 Extreme Summarization

Similar considerations can be traced for XSum, as reported in Table 11. For both RLHF-instructed and pre-trained models, the quality of output produced by the baselines tends to dramatically decrease at higher temperatures (only min-$p$ achieves good results at $\tau > 1.0$), with the consequence of an increasing against-greedy diversity due to the choice of random and meaningless tokens. Instead, the quality of the output generated by *DiffSampling* remains more stable, with small but consistent increases in diversity.

| Model: | RLHF-instructed | | | | | | | Pre-trained | | | | | | |
|---|---|---|---|---|---|---|---|---|---|---|---|---|---|---|
| Method | Quality | | | Per-Input | | Against-Greedy | | Quality | | | Per-Input | | Against-Greedy | |
| | R-1 | SIM | COH | EAD | SBERT | EAD | SBERT | R-1 | SIM | COH | EAD | SBERT | EAD | SBERT |
| *Temperature = 0.0* | | | | | | | | | | | | | | |
| Greedy | $0.23_{\pm.00}$ | $0.49_{\pm.01}$ | $0.63_{\pm.01}$ | $0.18_{\pm.00}$ | $0.00_{\pm.00}$ | - | - | $0.22_{\pm.00}$ | $0.51_{\pm.00}$ | $0.74_{\pm.00}$ | $0.19_{\pm.00}$ | $0.00_{\pm.00}$ | - | - |
| *Temperature = 1.0* | | | | | | | | | | | | | | |
| Top-$p$ | $0.21_{\pm.00}$ | $0.45_{\pm.01}$ | $0.59_{\pm.01}$ | $0.36_{\pm.01}$ | $0.47_{\pm.01}$ | $0.66_{\pm.01}$ | $0.41_{\pm.01}$ | $0.16_{\pm.00}$ | $0.34_{\pm.01}$ | $0.48_{\pm.01}$ | $0.72_{\pm.01}$ | $0.66_{\pm.01}$ | $0.77_{\pm.01}$ | $0.55_{\pm.01}$ |
| $\eta$-sampling | $0.20_{\pm.00}$ | $0.45_{\pm.01}$ | $0.58_{\pm.01}$ | $0.38_{\pm.01}$ | $0.49_{\pm.01}$ | $0.69_{\pm.01}$ | $0.43_{\pm.01}$ | $0.16_{\pm.00}$ | $0.34_{\pm.01}$ | $0.48_{\pm.01}$ | $0.75_{\pm.01}$ | $0.67_{\pm.00}$ | $0.80_{\pm.01}$ | $0.56_{\pm.01}$ |
| Locally typical | $0.21_{\pm.00}$ | $0.45_{\pm.01}$ | $0.59_{\pm.01}$ | $0.36_{\pm.01}$ | $0.47_{\pm.01}$ | $0.66_{\pm.01}$ | $0.41_{\pm.01}$ | $0.16_{\pm.00}$ | $0.34_{\pm.01}$ | $0.48_{\pm.01}$ | $0.72_{\pm.01}$ | $0.66_{\pm.01}$ | $0.77_{\pm.01}$ | $0.55_{\pm.01}$ |
| Min-$p$ | $0.22_{\pm.00}$ | $0.46_{\pm.01}$ | $0.61_{\pm.01}$ | $0.36_{\pm.01}$ | $0.43_{\pm.01}$ | $0.64_{\pm.01}$ | $0.38_{\pm.01}$ | $0.20_{\pm.00}$ | $0.44_{\pm.01}$ | $0.63_{\pm.01}$ | $0.65_{\pm.01}$ | $0.47_{\pm.01}$ | $0.62_{\pm.01}$ | $0.39_{\pm.01}$ |
| DiffS.-cut | $0.23_{\pm.00}$ | $0.48_{\pm.01}$ | $0.63_{\pm.01}$ | $0.35_{\pm.01}$ | $0.25_{\pm.01}$ | $0.45_{\pm.01}$ | $0.23_{\pm.01}$ | $0.21_{\pm.00}$ | $0.49_{\pm.00}$ | $0.73_{\pm.00}$ | $0.38_{\pm.01}$ | $0.19_{\pm.00}$ | $0.32_{\pm.01}$ | $0.17_{\pm.01}$ |
| DiffS.-lb | $0.21_{\pm.00}$ | $0.45_{\pm.01}$ | $0.59_{\pm.01}$ | $0.37_{\pm.01}$ | $0.47_{\pm.01}$ | $0.67_{\pm.01}$ | $0.41_{\pm.01}$ | $0.16_{\pm.00}$ | $0.34_{\pm.01}$ | $0.48_{\pm.01}$ | $0.72_{\pm.01}$ | $0.66_{\pm.01}$ | $0.77_{\pm.01}$ | $0.55_{\pm.01}$ |
| DiffS.-minp | $0.22_{\pm.00}$ | $0.46_{\pm.01}$ | $0.60_{\pm.01}$ | $0.35_{\pm.01}$ | $0.43_{\pm.01}$ | $0.64_{\pm.01}$ | $0.38_{\pm.01}$ | $0.20_{\pm.00}$ | $0.44_{\pm.01}$ | $0.62_{\pm.01}$ | $0.65_{\pm.01}$ | $0.47_{\pm.01}$ | $0.63_{\pm.01}$ | $0.39_{\pm.01}$ |
| *Temperature = 0.6* | | | | | | | | | | | | | | |
| Top-$p$ | $0.22_{\pm.00}$ | $0.48_{\pm.01}$ | $0.62_{\pm.01}$ | $0.38_{\pm.01}$ | $0.34_{\pm.01}$ | $0.56_{\pm.01}$ | $0.30_{\pm.01}$ | $0.20_{\pm.00}$ | $0.47_{\pm.01}$ | $0.68_{\pm.01}$ | $0.53_{\pm.01}$ | $0.37_{\pm.01}$ | $0.50_{\pm.01}$ | $0.30_{\pm.01}$ |
| $\eta$-sampling | $0.22_{\pm.00}$ | $0.47_{\pm.01}$ | $0.62_{\pm.01}$ | $0.38_{\pm.01}$ | $0.38_{\pm.01}$ | $0.59_{\pm.01}$ | $0.33_{\pm.01}$ | $0.20_{\pm.00}$ | $0.46_{\pm.01}$ | $0.66_{\pm.01}$ | $0.59_{\pm.01}$ | $0.42_{\pm.01}$ | $0.55_{\pm.01}$ | $0.33_{\pm.01}$ |
| Locally typical | $0.22_{\pm.00}$ | $0.48_{\pm.01}$ | $0.62_{\pm.01}$ | $0.38_{\pm.01}$ | $0.34_{\pm.01}$ | $0.56_{\pm.01}$ | $0.30_{\pm.01}$ | $0.20_{\pm.00}$ | $0.47_{\pm.01}$ | $0.68_{\pm.01}$ | $0.53_{\pm.01}$ | $0.37_{\pm.01}$ | $0.50_{\pm.01}$ | $0.30_{\pm.01}$ |
| Min-$p$ | $0.22_{\pm.00}$ | $0.48_{\pm.01}$ | $0.63_{\pm.01}$ | $0.39_{\pm.01}$ | $0.34_{\pm.01}$ | $0.55_{\pm.01}$ | $0.30_{\pm.01}$ | $0.21_{\pm.00}$ | $0.47_{\pm.01}$ | $0.69_{\pm.01}$ | $0.50_{\pm.01}$ | $0.33_{\pm.01}$ | $0.46_{\pm.01}$ | $0.28_{\pm.01}$ |
| DiffS.-cut | $0.23_{\pm.00}$ | $0.49_{\pm.01}$ | $0.63_{\pm.01}$ | $0.35_{\pm.01}$ | $0.24_{\pm.01}$ | $0.43_{\pm.01}$ | $0.22_{\pm.01}$ | $0.21_{\pm.00}$ | $0.49_{\pm.00}$ | $0.73_{\pm.00}$ | $0.37_{\pm.01}$ | $0.18_{\pm.00}$ | $0.30_{\pm.01}$ | $0.16_{\pm.01}$ |
| DiffS.-lb | $0.22_{\pm.00}$ | $0.47_{\pm.01}$ | $0.62_{\pm.01}$ | $0.39_{\pm.01}$ | $0.38_{\pm.01}$ | $0.59_{\pm.01}$ | $0.32_{\pm.01}$ | $0.20_{\pm.00}$ | $0.45_{\pm.01}$ | $0.65_{\pm.01}$ | $0.59_{\pm.01}$ | $0.42_{\pm.01}$ | $0.55_{\pm.01}$ | $0.34_{\pm.01}$ |
| DiffS.-minp | $0.22_{\pm.00}$ | $0.48_{\pm.01}$ | $0.62_{\pm.01}$ | $0.39_{\pm.01}$ | $0.37_{\pm.01}$ | $0.58_{\pm.01}$ | $0.32_{\pm.01}$ | $0.20_{\pm.00}$ | $0.47_{\pm.01}$ | $0.67_{\pm.01}$ | $0.55_{\pm.01}$ | $0.38_{\pm.01}$ | $0.52_{\pm.01}$ | $0.31_{\pm.01}$ |
| *Temperature = 1.5* | | | | | | | | | | | | | | |
| Top-$p$ | $0.07_{\pm.00}$ | $0.18_{\pm.01}$ | $0.30_{\pm.01}$ | $0.64_{\pm.01}$ | $0.72_{\pm.00}$ | $0.88_{\pm.01}$ | $0.77_{\pm.01}$ | $0.03_{\pm.00}$ | $0.08_{\pm.00}$ | $0.19_{\pm.00}$ | $0.79_{\pm.01}$ | $0.77_{\pm.00}$ | $0.97_{\pm.00}$ | $0.88_{\pm.00}$ |
| $\eta$-sampling | $0.10_{\pm.00}$ | $0.25_{\pm.01}$ | $0.38_{\pm.01}$ | $0.63_{\pm.01}$ | $0.70_{\pm.00}$ | $0.86_{\pm.01}$ | $0.69_{\pm.01}$ | $0.03_{\pm.00}$ | $0.08_{\pm.00}$ | $0.19_{\pm.00}$ | $0.79_{\pm.01}$ | $0.77_{\pm.00}$ | $0.96_{\pm.00}$ | $0.88_{\pm.00}$ |
| Locally typical | $0.06_{\pm.00}$ | $0.17_{\pm.01}$ | $0.30_{\pm.01}$ | $0.69_{\pm.01}$ | $0.70_{\pm.00}$ | $0.89_{\pm.01}$ | $0.77_{\pm.01}$ | $0.02_{\pm.00}$ | $0.07_{\pm.00}$ | $0.19_{\pm.00}$ | $0.82_{\pm.01}$ | $0.72_{\pm.00}$ | $0.97_{\pm.00}$ | $0.90_{\pm.00}$ |
| Min-$p$ | $0.20_{\pm.00}$ | $0.44_{\pm.01}$ | $0.58_{\pm.01}$ | $0.37_{\pm.01}$ | $0.50_{\pm.01}$ | $0.71_{\pm.01}$ | $0.45_{\pm.01}$ | $0.17_{\pm.00}$ | $0.38_{\pm.01}$ | $0.53_{\pm.01}$ | $0.78_{\pm.01}$ | $0.60_{\pm.00}$ | $0.79_{\pm.01}$ | $0.51_{\pm.01}$ |
| DiffS.-cut | $0.23_{\pm.00}$ | $0.48_{\pm.01}$ | $0.63_{\pm.01}$ | $0.35_{\pm.01}$ | $0.25_{\pm.01}$ | $0.46_{\pm.01}$ | $0.23_{\pm.01}$ | $0.21_{\pm.00}$ | $0.49_{\pm.00}$ | $0.73_{\pm.01}$ | $0.38_{\pm.01}$ | $0.19_{\pm.00}$ | $0.33_{\pm.01}$ | $0.18_{\pm.01}$ |
| DiffS.-lb | $0.20_{\pm.00}$ | $0.43_{\pm.01}$ | $0.57_{\pm.01}$ | $0.40_{\pm.01}$ | $0.52_{\pm.01}$ | $0.71_{\pm.01}$ | $0.45_{\pm.01}$ | $0.11_{\pm.00}$ | $0.23_{\pm.01}$ | $0.34_{\pm.01}$ | $0.75_{\pm.01}$ | $0.79_{\pm.00}$ | $0.89_{\pm.00}$ | $0.70_{\pm.01}$ |
| DiffS.-minp | $0.21_{\pm.00}$ | $0.46_{\pm.01}$ | $0.60_{\pm.01}$ | $0.35_{\pm.01}$ | $0.45_{\pm.01}$ | $0.67_{\pm.01}$ | $0.41_{\pm.01}$ | $0.19_{\pm.00}$ | $0.43_{\pm.01}$ | $0.60_{\pm.01}$ | $0.70_{\pm.01}$ | $0.52_{\pm.00}$ | $0.70_{\pm.01}$ | $0.44_{\pm.01}$ |
| *Temperature = 2.0* | | | | | | | | | | | | | | |
| Top-$p$ | $0.01_{\pm.00}$ | $0.03_{\pm.00}$ | $0.14_{\pm.00}$ | $0.63_{\pm.01}$ | $0.65_{\pm.00}$ | $0.92_{\pm.01}$ | $0.91_{\pm.00}$ | $0.01_{\pm.00}$ | $0.03_{\pm.00}$ | $0.15_{\pm.00}$ | $0.81_{\pm.01}$ | $0.65_{\pm.00}$ | $0.98_{\pm.00}$ | $0.94_{\pm.00}$ |
| $\eta$-sampling | $0.01_{\pm.00}$ | $0.03_{\pm.00}$ | $0.13_{\pm.00}$ | $0.65_{\pm.01}$ | $0.67_{\pm.00}$ | $0.92_{\pm.01}$ | $0.92_{\pm.00}$ | $0.01_{\pm.00}$ | $0.03_{\pm.00}$ | $0.14_{\pm.00}$ | $0.80_{\pm.01}$ | $0.66_{\pm.00}$ | $0.98_{\pm.00}$ | $0.93_{\pm.00}$ |
| Locally typical | $0.00_{\pm.00}$ | $0.03_{\pm.00}$ | $0.14_{\pm.00}$ | $0.71_{\pm.01}$ | $0.63_{\pm.00}$ | $0.93_{\pm.00}$ | $0.92_{\pm.00}$ | $0.01_{\pm.00}$ | $0.03_{\pm.00}$ | $0.15_{\pm.00}$ | $0.83_{\pm.01}$ | $0.63_{\pm.00}$ | $0.98_{\pm.00}$ | $0.94_{\pm.00}$ |
| Min-$p$ | $0.19_{\pm.00}$ | $0.42_{\pm.01}$ | $0.55_{\pm.01}$ | $0.41_{\pm.01}$ | $0.56_{\pm.01}$ | $0.76_{\pm.01}$ | $0.50_{\pm.01}$ | $0.13_{\pm.00}$ | $0.30_{\pm.01}$ | $0.43_{\pm.01}$ | $0.82_{\pm.01}$ | $0.72_{\pm.00}$ | $0.89_{\pm.00}$ | $0.62_{\pm.01}$ |
| DiffS.-cut | $0.23_{\pm.00}$ | $0.48_{\pm.01}$ | $0.63_{\pm.01}$ | $0.35_{\pm.01}$ | $0.25_{\pm.01}$ | $0.46_{\pm.01}$ | $0.24_{\pm.01}$ | $0.21_{\pm.00}$ | $0.49_{\pm.00}$ | $0.73_{\pm.01}$ | $0.38_{\pm.01}$ | $0.19_{\pm.00}$ | $0.33_{\pm.01}$ | $0.18_{\pm.01}$ |
| DiffS.-lb | $0.19_{\pm.00}$ | $0.43_{\pm.01}$ | $0.56_{\pm.01}$ | $0.43_{\pm.01}$ | $0.54_{\pm.01}$ | $0.73_{\pm.01}$ | $0.48_{\pm.01}$ | $0.09_{\pm.00}$ | $0.19_{\pm.01}$ | $0.29_{\pm.01}$ | $0.76_{\pm.01}$ | $0.82_{\pm.00}$ | $0.92_{\pm.00}$ | $0.75_{\pm.01}$ |
| DiffS.-minp | $0.21_{\pm.00}$ | $0.46_{\pm.01}$ | $0.59_{\pm.01}$ | $0.37_{\pm.01}$ | $0.47_{\pm.01}$ | $0.69_{\pm.01}$ | $0.42_{\pm.01}$ | $0.19_{\pm.00}$ | $0.42_{\pm.01}$ | $0.58_{\pm.01}$ | $0.72_{\pm.01}$ | $0.53_{\pm.00}$ | $0.72_{\pm.01}$ | $0.45_{\pm.01}$ |
| *Temperature = 10.0* | | | | | | | | | | | | | | |
| Top-$p$ | $0.00_{\pm.00}$ | $0.02_{\pm.00}$ | $0.12_{\pm.00}$ | $0.74_{\pm.01}$ | $0.64_{\pm.00}$ | $0.93_{\pm.00}$ | $0.92_{\pm.00}$ | $0.00_{\pm.00}$ | $0.03_{\pm.00}$ | $0.13_{\pm.00}$ | $0.78_{\pm.01}$ | $0.61_{\pm.00}$ | $0.97_{\pm.00}$ | $0.95_{\pm.00}$ |
| $\eta$-sampling | $0.00_{\pm.00}$ | $0.02_{\pm.00}$ | $0.11_{\pm.00}$ | $0.74_{\pm.01}$ | $0.64_{\pm.00}$ | $0.93_{\pm.00}$ | $0.93_{\pm.00}$ | $0.00_{\pm.00}$ | $0.02_{\pm.00}$ | $0.12_{\pm.00}$ | $0.75_{\pm.01}$ | $0.63_{\pm.00}$ | $0.97_{\pm.00}$ | $0.95_{\pm.00}$ |
| Locally typical | $0.00_{\pm.00}$ | $0.02_{\pm.00}$ | $0.12_{\pm.00}$ | $0.75_{\pm.01}$ | $0.63_{\pm.00}$ | $0.93_{\pm.00}$ | $0.93_{\pm.00}$ | $0.00_{\pm.00}$ | $0.02_{\pm.00}$ | $0.12_{\pm.00}$ | $0.77_{\pm.01}$ | $0.61_{\pm.00}$ | $0.97_{\pm.00}$ | $0.95_{\pm.00}$ |
| Min-$p$ | $0.00_{\pm.00}$ | $0.02_{\pm.00}$ | $0.11_{\pm.00}$ | $0.74_{\pm.01}$ | $0.64_{\pm.00}$ | $0.93_{\pm.00}$ | $0.93_{\pm.00}$ | $0.00_{\pm.00}$ | $0.02_{\pm.00}$ | $0.12_{\pm.00}$ | $0.75_{\pm.01}$ | $0.63_{\pm.00}$ | $0.97_{\pm.00}$ | $0.95_{\pm.00}$ |
| DiffS.-cut | $0.23_{\pm.00}$ | $0.48_{\pm.01}$ | $0.63_{\pm.01}$ | $0.35_{\pm.01}$ | $0.25_{\pm.01}$ | $0.47_{\pm.01}$ | $0.25_{\pm.01}$ | $0.21_{\pm.00}$ | $0.49_{\pm.00}$ | $0.73_{\pm.01}$ | $0.39_{\pm.01}$ | $0.19_{\pm.00}$ | $0.34_{\pm.01}$ | $0.19_{\pm.01}$ |
| DiffS.-lb | $0.17_{\pm.00}$ | $0.40_{\pm.01}$ | $0.52_{\pm.01}$ | $0.48_{\pm.01}$ | $0.59_{\pm.01}$ | $0.78_{\pm.01}$ | $0.54_{\pm.01}$ | $0.05_{\pm.00}$ | $0.13_{\pm.00}$ | $0.23_{\pm.01}$ | $0.77_{\pm.01}$ | $0.83_{\pm.00}$ | $0.95_{\pm.00}$ | $0.83_{\pm.01}$ |
| DiffS.-minp | $0.21_{\pm.00}$ | $0.45_{\pm.01}$ | $0.58_{\pm.01}$ | $0.38_{\pm.01}$ | $0.49_{\pm.01}$ | $0.72_{\pm.01}$ | $0.45_{\pm.01}$ | $0.18_{\pm.00}$ | $0.40_{\pm.01}$ | $0.55_{\pm.01}$ | $0.76_{\pm.01}$ | $0.57_{\pm.00}$ | $0.79_{\pm.01}$ | $0.50_{\pm.01}$ |

Table 11: Aggregate results over 5 outputs sampled for each of the 1000 prompts from the XSum dataset for the instructed model (left) and the pre-trained model (right) when adopting different temperature values. The mean and 95% confidence interval are reported for all the metrics.

### F.3 Divergence Association Task

Figure 5 reports the DAT score and the percentage of output validity of the DPO-instructed and pre-trained models with different temperature values. Top-$p$ sampling, locally typical sampling, and $\eta$-sampling rapidly stop outputting valid lists of nouns when the temperature raises, even if the DAT score tends to be higher at $\tau = 1.5$; min-$p$ returns a high percentage of valid outputs even at $\tau = 2.0$, but increase the DAT score only for the pre-trained model and cannot produce anything valid at $\tau = 10.0$. Instead, the performance of our methods remains very similar across different temperatures in terms of both the DAT score and the percentage of valid outputs, except for *DiffSampling-lb*, whose scores increase at the cost of some validity.

On the other hand, the greedy decoding strategy is less effective for the pre-trained model, which results in higher temperatures yielding better DAT scores across both the baselines and our methods. However, the number of valid outputs decreases faster, and top-$p$ sampling, locally typical sampling, and $\eta$-sampling produce very few correct lists at a temperature of 1.5 (but with a higher DAT score). Again, min-$p$ better manages temperatures around 1.5 and 2.0, with higher scores and still at least half of the outputs as valid, but cannot produce any correct output at a temperature of 10.0.

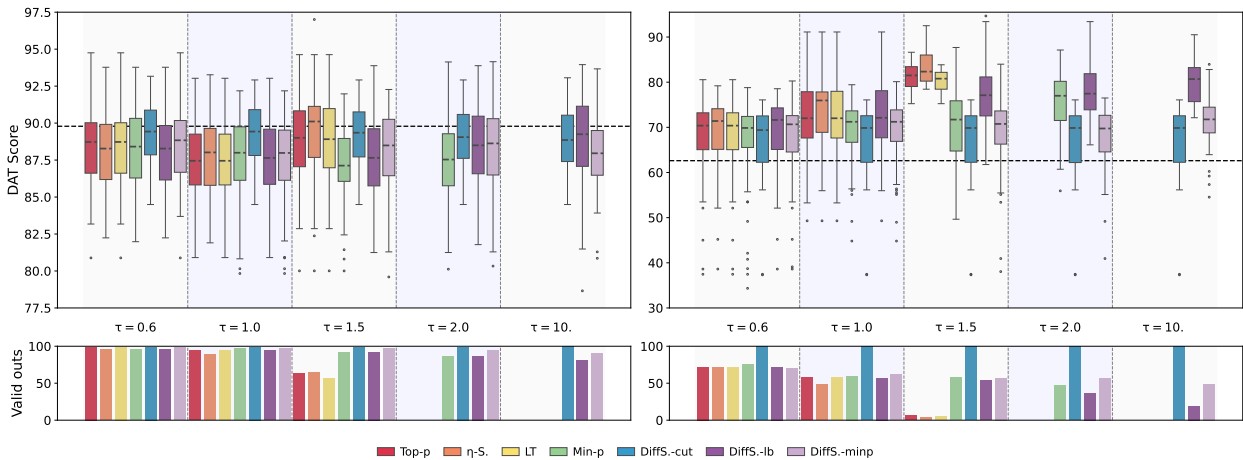

Figure 5: DAT scores for our methods and the baselines for the instructed (left) and pre-trained (right) model with different temperature values, together with the number of valid outputs produced by each sampling strategy. The dashed line represents the score of the greedy strategy.

## F.4 WritingPrompts

Finally, Table 12 reports the full results for the story generation task at different temperatures. As above, the quality of output produced by the baselines drops quickly at higher temperatures, and only min-$p$ achieves competitive results at $\tau = 2.0$. On the contrary, the quality of the output generated by *DiffSampling-cut* and *DiffSampling-minp* remains more stable, with small but consistent increases in diversity; *DiffSampling-lb* behaves worse, but still better than its competitor top-$p$.

## F.5 Temperature Before or After Truncating

As thoroughly described in the article, we apply temperature after truncating based on the minimum discrete derivative to preserve the guarantees of correctness of selected tokens. However, the de facto standard is to apply temperature before any other truncation or modification. In this section, we examine the implications of the temperature position in terms of quality and diversity.

Table 13 reports the results of our methods with temperature before (left side) and after (right side) the truncation for the GSM8K test set. As we can see, applying the temperature before causes the accuracy to degrade at higher temperatures, while ensuring a slightly higher diversity. Interestingly, at $\tau = 0.6$, applying the temperature after leads to better results in terms of both accuracy and diversity. This confirms that our choice preserves the quality as much as possible, at the cost of some additional diversity.

Table 14 reports the results of our methods with temperature before (left side) and after (right side) the truncation for the MATH test set. Again, applying a higher temperature before causes the accuracy to drop quickly for the two relaxations, and smoothly for *DiffSampling-cut*, with benefits only in terms of syntactic diversity. Instead, applying the temperature after has a negligible impact on quality while fostering diversity.

The same considerations hold for XSum as well. For both the instructed (Table 15) and pre-trained (Table 16) models, the quality is not preserved with the temperature before, while it is with the temperature after, although diversity does not increase in the same way. Again, for *DiffSampling-lb* and *DiffSampling-minp*, the diversity at $\tau = 0.6$ is instead greater with the temperature after, even if the quality is, more or less, the same. However, in the case of *DiffSampling-cut*, we can observe the opposite behavior: moving the temperature before cutting leads to a higher diversity at the cost of some quality.

Applying the temperature before does not seem to give benefits for the divergence association task as well. As shown by Figure 6, for both the instructed and pre-trained models, the DAT scores are very similar regardless of the temperature position, but almost no valid solutions are generated when a temperature of

| Model: | RLHF-instructed | | | Pre-trained | | |
|---|---|---|---|---|---|---|
| Method | Quality | Per-Input Diversity | | Quality | Per-Input Diversity | |
| | COH | EAD | SBERT | COH | EAD | SBERT |
| *Temperature = 0.0* | | | | | | |
| Greedy | $0.44_{\pm.01}$ | $0.06_{\pm.01}$ | - | $0.59_{\pm.01}$ | $0.07_{\pm.00}$ | - |
| *Temperature = 1.0* | | | | | | |
| Top-$p$ | $0.42_{\pm.01}$ | $0.73_{\pm.00}$ | $0.25_{\pm.00}$ | $0.42_{\pm.01}$ | $0.64_{\pm.00}$ | $0.58_{\pm.00}$ |
| $\eta$-sampling | $0.42_{\pm.01}$ | $0.80_{\pm.00}$ | $0.28_{\pm.00}$ | $0.40_{\pm.01}$ | $0.77_{\pm.00}$ | $0.60_{\pm.00}$ |
| Locally typical | $0.42_{\pm.01}$ | $0.73_{\pm.00}$ | $0.25_{\pm.00}$ | $0.42_{\pm.01}$ | $0.64_{\pm.00}$ | $0.58_{\pm.00}$ |
| Min-$p$ | $0.43_{\pm.01}$ | $0.71_{\pm.00}$ | $0.23_{\pm.00}$ | $0.51_{\pm.01}$ | $0.35_{\pm.01}$ | $0.46_{\pm.00}$ |
| DiffS.-cut | $0.43_{\pm.01}$ | $0.63_{\pm.00}$ | $0.19_{\pm.00}$ | $0.60_{\pm.01}$ | $0.15_{\pm.00}$ | $0.31_{\pm.01}$ |
| DiffS.-lb | $0.42_{\pm.01}$ | $0.73_{\pm.00}$ | $0.25_{\pm.00}$ | $0.41_{\pm.01}$ | $0.67_{\pm.00}$ | $0.58_{\pm.00}$ |
| DiffS.-minp | $0.43_{\pm.01}$ | $0.71_{\pm.00}$ | $0.23_{\pm.00}$ | $0.51_{\pm.01}$ | $0.36_{\pm.00}$ | $0.47_{\pm.00}$ |
| *Temperature = 0.6* | | | | | | |
| Top-$p$ | $0.43_{\pm.01}$ | $0.67_{\pm.00}$ | $0.22_{\pm.00}$ | $0.55_{\pm.01}$ | $0.21_{\pm.00}$ | $0.42_{\pm.00}$ |
| $\eta$-sampling | $0.43_{\pm.01}$ | $0.68_{\pm.00}$ | $0.22_{\pm.00}$ | $0.53_{\pm.01}$ | $0.28_{\pm.00}$ | $0.45_{\pm.00}$ |
| Locally typical | $0.43_{\pm.01}$ | $0.67_{\pm.00}$ | $0.22_{\pm.00}$ | $0.55_{\pm.01}$ | $0.21_{\pm.00}$ | $0.42_{\pm.00}$ |
| Min-$p$ | $0.43_{\pm.01}$ | $0.66_{\pm.00}$ | $0.22_{\pm.00}$ | $0.57_{\pm.01}$ | $0.20_{\pm.00}$ | $0.40_{\pm.00}$ |
| DiffS.-cut | $0.43_{\pm.01}$ | $0.63_{\pm.00}$ | $0.19_{\pm.00}$ | $0.60_{\pm.01}$ | $0.16_{\pm.00}$ | $0.31_{\pm.01}$ |
| DiffS.-lb | $0.43_{\pm.01}$ | $0.68_{\pm.00}$ | $0.23_{\pm.00}$ | $0.53_{\pm.01}$ | $0.27_{\pm.00}$ | $0.45_{\pm.00}$ |
| DiffS.-minp | $0.43_{\pm.01}$ | $0.68_{\pm.00}$ | $0.22_{\pm.00}$ | $0.54_{\pm.01}$ | $0.24_{\pm.00}$ | $0.43_{\pm.00}$ |
| *Temperature = 1.5* | | | | | | |
| Top-$p$ | $0.18_{\pm.00}$ | $1.00_{\pm.00}$ | $0.14_{\pm.00}$ | $0.15_{\pm.00}$ | $1.00_{\pm.00}$ | $0.24_{\pm.00}$ |
| $\eta$-sampling | $0.21_{\pm.00}$ | $1.01_{\pm.00}$ | $0.17_{\pm.00}$ | $0.15_{\pm.00}$ | $1.00_{\pm.00}$ | $0.29_{\pm.00}$ |
| Locally typical | $0.17_{\pm.00}$ | $1.01_{\pm.00}$ | $0.14_{\pm.00}$ | $0.14_{\pm.00}$ | $1.00_{\pm.00}$ | $0.25_{\pm.00}$ |
| Min-$p$ | $0.42_{\pm.01}$ | $0.78_{\pm.00}$ | $0.26_{\pm.00}$ | $0.43_{\pm.01}$ | $0.73_{\pm.00}$ | $0.55_{\pm.00}$ |
| DiffS.-cut | $0.43_{\pm.01}$ | $0.63_{\pm.00}$ | $0.19_{\pm.00}$ | $0.60_{\pm.01}$ | $0.16_{\pm.00}$ | $0.31_{\pm.01}$ |
| DiffS.-lb | $0.40_{\pm.01}$ | $0.89_{\pm.00}$ | $0.33_{\pm.00}$ | $0.31_{\pm.01}$ | $0.92_{\pm.00}$ | $0.63_{\pm.00}$ |
| DiffS.-minp | $0.43_{\pm.01}$ | $0.73_{\pm.00}$ | $0.24_{\pm.00}$ | $0.48_{\pm.01}$ | $0.46_{\pm.00}$ | $0.49_{\pm.00}$ |
| *Temperature = 2.0* | | | | | | |
| Top-$p$ | $0.12_{\pm.00}$ | $1.01_{\pm.00}$ | $0.10_{\pm.00}$ | $0.12_{\pm.00}$ | $1.01_{\pm.00}$ | $0.27_{\pm.00}$ |
| $\eta$-sampling | $0.11_{\pm.00}$ | $1.02_{\pm.00}$ | $0.12_{\pm.00}$ | $0.11_{\pm.00}$ | $1.00_{\pm.00}$ | $0.32_{\pm.00}$ |
| Locally typical | $0.12_{\pm.00}$ | $1.01_{\pm.00}$ | $0.10_{\pm.00}$ | $0.11_{\pm.00}$ | $1.01_{\pm.00}$ | $0.29_{\pm.00}$ |
| Min-$p$ | $0.39_{\pm.01}$ | $0.92_{\pm.00}$ | $0.33_{\pm.00}$ | $0.35_{\pm.01}$ | $0.85_{\pm.00}$ | $0.62_{\pm.00}$ |
| DiffS.-cut | $0.43_{\pm.01}$ | $0.64_{\pm.00}$ | $0.19_{\pm.00}$ | $0.60_{\pm.01}$ | $0.16_{\pm.00}$ | $0.31_{\pm.00}$ |
| DiffS.-lb | $0.38_{\pm.01}$ | $0.93_{\pm.00}$ | $0.33_{\pm.00}$ | $0.26_{\pm.01}$ | $0.97_{\pm.00}$ | $0.52_{\pm.00}$ |
| DiffS.-minp | $0.43_{\pm.01}$ | $0.74_{\pm.00}$ | $0.24_{\pm.00}$ | $0.47_{\pm.01}$ | $0.53_{\pm.00}$ | $0.50_{\pm.00}$ |
| *Temperature = 10.0* | | | | | | |
| Top-$p$ | $0.11_{\pm.00}$ | $1.02_{\pm.00}$ | $0.13_{\pm.00}$ | $0.10_{\pm.00}$ | $1.00_{\pm.00}$ | $0.33_{\pm.00}$ |
| $\eta$-sampling | $0.10_{\pm.00}$ | $1.02_{\pm.00}$ | $0.14_{\pm.00}$ | $0.10_{\pm.00}$ | $1.00_{\pm.00}$ | $0.36_{\pm.00}$ |
| Locally typical | $0.10_{\pm.00}$ | $1.02_{\pm.00}$ | $0.14_{\pm.00}$ | $0.10_{\pm.00}$ | $1.00_{\pm.00}$ | $0.35_{\pm.00}$ |
| Min-$p$ | $0.10_{\pm.00}$ | $1.02_{\pm.00}$ | $0.14_{\pm.00}$ | $0.10_{\pm.00}$ | $1.00_{\pm.00}$ | $0.36_{\pm.00}$ |
| DiffS.-cut | $0.43_{\pm.01}$ | $0.64_{\pm.00}$ | $0.20_{\pm.00}$ | $0.60_{\pm.01}$ | $0.17_{\pm.00}$ | $0.31_{\pm.01}$ |
| DiffS.-lb | $0.32_{\pm.01}$ | $0.98_{\pm.00}$ | $0.29_{\pm.00}$ | $0.19_{\pm.00}$ | $1.00_{\pm.00}$ | $0.37_{\pm.00}$ |
| DiffS.-minp | $0.42_{\pm.01}$ | $0.77_{\pm.00}$ | $0.25_{\pm.00}$ | $0.45_{\pm.01}$ | $0.68_{\pm.00}$ | $0.52_{\pm.00}$ |

Table 12: Aggregate results over 3 seeds for the WritingPrompts dataset for the instructed model (left) and the pre-trained model (right) with different temperature values. The mean and standard error of the final score for each run are reported for cross-input diversity, whereas the mean and 95% confidence interval for the full set of answers are reported for the other metrics.

10.0 is applied before truncating (and the same happens for a temperature of 2.0 in the case of *DiffSampling-lb*).

Finally, as above, for both the instructed (Table 17) and pre-trained (Table 18) models, the quality of generated stories is not preserved with the temperature before, while it largely is with the temperature after.

| Method | BEFORE | | | | | AFTER | | | | |
|---|---|---|---|---|---|---|---|---|---|---|
| | Accuracy | Cross-Input | | Against-Greedy | | Accuracy | Cross-Input | | Against-Greedy | |
| | | EAD | SBERT | EAD | SBERT | | EAD | SBERT | EAD | SBERT |
| *Temperature = 0.6* | | | | | | | | | | |
| DiffS.-cut | $66.19_{\pm.12}$ | $2.04_{\pm.00}$ | $0.64_{\pm.00}$ | $0.10_{\pm.00}$ | $0.01_{\pm.00}$ | $66.74_{\pm.04}$ | $2.05_{\pm.00}$ | $0.64_{\pm.00}$ | $0.13_{\pm.00}$ | $0.02_{\pm.00}$ |
| DiffS.-lb | $66.59_{\pm.28}$ | $2.06_{\pm.00}$ | $0.64_{\pm.00}$ | $0.18_{\pm.00}$ | $0.02_{\pm.00}$ | $65.73_{\pm.23}$ | $2.06_{\pm.00}$ | $0.64_{\pm.00}$ | $0.19_{\pm.00}$ | $0.03_{\pm.00}$ |
| DiffS.-minp | $66.14_{\pm.15}$ | $2.05_{\pm.00}$ | $0.64_{\pm.00}$ | $0.17_{\pm.00}$ | $0.02_{\pm.00}$ | $67.05_{\pm.14}$ | $2.06_{\pm.01}$ | $0.64_{\pm.00}$ | $0.19_{\pm.00}$ | $0.03_{\pm.00}$ |
| *Temperature = 1.5* | | | | | | | | | | |
| DiffS.-cut | $66.16_{\pm.57}$ | $2.05_{\pm.00}$ | $0.64_{\pm.00}$ | $0.17_{\pm.00}$ | $0.02_{\pm.00}$ | $66.72_{\pm.36}$ | $2.05_{\pm.00}$ | $0.64_{\pm.00}$ | $0.15_{\pm.00}$ | $0.02_{\pm.00}$ |
| DiffS.-lb | $63.84_{\pm.53}$ | $2.18_{\pm.01}$ | $0.64_{\pm.00}$ | $0.28_{\pm.00}$ | $0.04_{\pm.00}$ | $65.20_{\pm.25}$ | $2.11_{\pm.01}$ | $0.64_{\pm.00}$ | $0.25_{\pm.00}$ | $0.04_{\pm.00}$ |
| DiffS.-minp | $64.34_{\pm.36}$ | $2.15_{\pm.01}$ | $0.64_{\pm.00}$ | $0.28_{\pm.00}$ | $0.04_{\pm.00}$ | $65.55_{\pm.61}$ | $2.11_{\pm.00}$ | $0.64_{\pm.00}$ | $0.25_{\pm.00}$ | $0.04_{\pm.00}$ |
| *Temperature = 2.0* | | | | | | | | | | |
| DiffS.-cut | $65.50_{\pm.09}$ | $2.06_{\pm.01}$ | $0.64_{\pm.00}$ | $0.19_{\pm.00}$ | $0.03_{\pm.00}$ | $66.44_{\pm.18}$ | $2.05_{\pm.00}$ | $0.64_{\pm.00}$ | $0.15_{\pm.00}$ | $0.02_{\pm.00}$ |
| DiffS.-lb | $18.17_{\pm.55}$ | $11.92_{\pm.04}$ | $0.61_{\pm.00}$ | $0.77_{\pm.01}$ | $0.43_{\pm.01}$ | $63.48_{\pm.43}$ | $2.12_{\pm.00}$ | $0.64_{\pm.00}$ | $0.26_{\pm.00}$ | $0.04_{\pm.00}$ |
| DiffS.-minp | $61.81_{\pm.14}$ | $2.25_{\pm.01}$ | $0.64_{\pm.00}$ | $0.32_{\pm.00}$ | $0.05_{\pm.00}$ | $65.13_{\pm.28}$ | $2.12_{\pm.01}$ | $0.64_{\pm.00}$ | $0.26_{\pm.00}$ | $0.04_{\pm.00}$ |
| *Temperature = 10.0* | | | | | | | | | | |
| DiffS.-cut | $61.31_{\pm.21}$ | $2.22_{\pm.01}$ | $0.64_{\pm.00}$ | $0.31_{\pm.00}$ | $0.04_{\pm.00}$ | $66.31_{\pm.26}$ | $2.04_{\pm.00}$ | $0.64_{\pm.00}$ | $0.15_{\pm.00}$ | $0.02_{\pm.00}$ |
| DiffS.-lb | $0.00_{\pm.00}$ | $17.38_{\pm.02}$ | $0.13_{\pm.00}$ | $1.00_{\pm.00}$ | $0.96_{\pm.00}$ | $64.11_{\pm.13}$ | $2.15_{\pm.00}$ | $0.64_{\pm.00}$ | $0.29_{\pm.00}$ | $0.04_{\pm.00}$ |
| DiffS.-minp | $0.00_{\pm.00}$ | $17.38_{\pm.02}$ | $0.13_{\pm.00}$ | $1.00_{\pm.00}$ | $0.96_{\pm.00}$ | $63.58_{\pm.43}$ | $2.17_{\pm.01}$ | $0.64_{\pm.00}$ | $0.29_{\pm.00}$ | $0.04_{\pm.00}$ |

Table 13: Accuracy and diversity of results for the GSM8K test set over 3 seeds. The mean and standard error of the final score for each run are reported for accuracy and cross-input diversity, whereas the mean and 95% confidence interval for the full set of answers are reported for against-greedy diversity.

| Method | BEFORE | | | | | AFTER | | | | |
|---|---|---|---|---|---|---|---|---|---|---|
| | Accuracy | Cross-Input | | Against-Greedy | | Accuracy | Cross-Input | | Against-Greedy | |
| | | EAD | SBERT | EAD | SBERT | | EAD | SBERT | EAD | SBERT |
| *Temperature = 0.6* | | | | | | | | | | |
| DiffS.-cut | $21.44_{\pm.12}$ | $5.69_{\pm.01}$ | $0.80_{\pm.00}$ | $0.22_{\pm.00}$ | $0.06_{\pm.00}$ | $21.52_{\pm.13}$ | $5.72_{\pm.00}$ | $0.80_{\pm.00}$ | $0.25_{\pm.00}$ | $0.07_{\pm.00}$ |
| DiffS.-lb | $21.22_{\pm.14}$ | $5.83_{\pm.01}$ | $0.80_{\pm.00}$ | $0.31_{\pm.00}$ | $0.09_{\pm.00}$ | $20.65_{\pm.20}$ | $5.89_{\pm.01}$ | $0.80_{\pm.00}$ | $0.32_{\pm.00}$ | $0.09_{\pm.00}$ |
| DiffS.-minp | $21.20_{\pm.06}$ | $5.83_{\pm.00}$ | $0.80_{\pm.00}$ | $0.31_{\pm.00}$ | $0.09_{\pm.00}$ | $20.56_{\pm.21}$ | $5.88_{\pm.00}$ | $0.80_{\pm.00}$ | $0.32_{\pm.00}$ | $0.09_{\pm.00}$ |
| *Temperature = 1.5* | | | | | | | | | | |
| DiffS.-cut | $21.15_{\pm.09}$ | $5.78_{\pm.01}$ | $0.80_{\pm.00}$ | $0.30_{\pm.00}$ | $0.08_{\pm.00}$ | $21.36_{\pm.15}$ | $5.73_{\pm.00}$ | $0.80_{\pm.00}$ | $0.27_{\pm.00}$ | $0.07_{\pm.00}$ |
| DiffS.-lb | $18.28_{\pm.05}$ | $7.05_{\pm.05}$ | $0.80_{\pm.00}$ | $0.42_{\pm.00}$ | $0.12_{\pm.00}$ | $19.55_{\pm.03}$ | $6.31_{\pm.02}$ | $0.80_{\pm.00}$ | $0.39_{\pm.00}$ | $0.11_{\pm.00}$ |
| DiffS.-minp | $18.72_{\pm.19}$ | $6.54_{\pm.01}$ | $0.80_{\pm.00}$ | $0.41_{\pm.00}$ | $0.12_{\pm.00}$ | $20.04_{\pm.13}$ | $6.19_{\pm.01}$ | $0.80_{\pm.00}$ | $0.38_{\pm.00}$ | $0.11_{\pm.00}$ |
| *Temperature = 2.0* | | | | | | | | | | |
| DiffS.-cut | $21.25_{\pm.10}$ | $5.85_{\pm.00}$ | $0.80_{\pm.00}$ | $0.32_{\pm.00}$ | $0.09_{\pm.00}$ | $21.66_{\pm.20}$ | $5.71_{\pm.01}$ | $0.80_{\pm.00}$ | $0.27_{\pm.00}$ | $0.08_{\pm.00}$ |
| DiffS.-lb | $1.77_{\pm.06}$ | $51.00_{\pm.09}$ | $0.48_{\pm.00}$ | $0.94_{\pm.00}$ | $0.72_{\pm.00}$ | $19.17_{\pm.10}$ | $6.40_{\pm.02}$ | $0.80_{\pm.00}$ | $0.40_{\pm.00}$ | $0.12_{\pm.00}$ |
| DiffS.-minp | $16.51_{\pm.06}$ | $7.25_{\pm.02}$ | $0.80_{\pm.00}$ | $0.45_{\pm.00}$ | $0.13_{\pm.00}$ | $19.70_{\pm.09}$ | $6.32_{\pm.02}$ | $0.80_{\pm.00}$ | $0.39_{\pm.00}$ | $0.11_{\pm.00}$ |
| *Temperature = 10.0* | | | | | | | | | | |
| DiffS.-cut | $16.63_{\pm.12}$ | $6.78_{\pm.01}$ | $0.80_{\pm.00}$ | $0.43_{\pm.00}$ | $0.12_{\pm.00}$ | $21.22_{\pm.11}$ | $5.74_{\pm.01}$ | $0.80_{\pm.00}$ | $0.28_{\pm.00}$ | $0.08_{\pm.00}$ |
| DiffS.-lb | $0.00_{\pm.00}$ | $59.15_{\pm.04}$ | $0.13_{\pm.00}$ | $1.00_{\pm.00}$ | $1.00_{\pm.00}$ | $18.20_{\pm.07}$ | $6.72_{\pm.00}$ | $0.80_{\pm.00}$ | $0.42_{\pm.00}$ | $0.12_{\pm.00}$ |
| DiffS.-minp | $0.00_{\pm.00}$ | $59.15_{\pm.04}$ | $0.13_{\pm.00}$ | $1.00_{\pm.00}$ | $1.00_{\pm.00}$ | $18.64_{\pm.20}$ | $6.54_{\pm.01}$ | $0.80_{\pm.00}$ | $0.41_{\pm.00}$ | $0.12_{\pm.00}$ |

Table 14: Accuracy and diversity of results for the MATH test set over 3 seeds. The mean and standard error of the final score for each run are reported for accuracy and cross-input diversity, whereas the mean and 95% confidence interval for the full set of answers are reported for against-greedy diversity.

While EAD does not increase in the same way, it is important to notice how SBERT is instead higher with the temperature after, highlighting how randomness does not correlate with semantic diversity.

### F.6 Quality-Diversity Trade-Off Visualization

To simplify the comprehension of how temperature impacts the quality-diversity trade-off and how the temperature position changes the performance of different sampling methods, we present here the Pareto fronts for top-$p$, min-$p$, and our three *DiffSampling* methods (with temperature applied either before or after truncation) at five different temperatures $\tau$. We omit the DAT plots for conciseness, as they convey the same message as Figures 5 and 6.

Figure 7 reports the plots for the math problem-solving datasets by considering both against-greedy EAD diversity and against-greedy SBERT diversity with respect to accuracy. As already discussed, applying the

| Method | BEFORE | | | | | | | AFTER | | | | | | |
|---|---|---|---|---|---|---|---|---|---|---|---|---|---|---|
| | Quality | | | Per-Input | | Against-Greedy | | Quality | | | Per-Input | | Against-Greedy | |
| | R-1 | SIM | COH | EAD | SBERT | EAD | SBERT | R-1 | SIM | COH | EAD | SBERT | EAD | SBERT |
| *Temperature = 0.6* | | | | | | | | | | | | | | |
| DiffS.-cut | $0.22_{\pm.00}$ | $0.48_{\pm.01}$ | $0.62_{\pm.01}$ | $0.41_{\pm.01}$ | $0.37_{\pm.01}$ | $0.64_{\pm.01}$ | $0.36_{\pm.01}$ | $0.23_{\pm.00}$ | $0.49_{\pm.01}$ | $0.63_{\pm.01}$ | $0.35_{\pm.01}$ | $0.24_{\pm.01}$ | $0.43_{\pm.01}$ | $0.22_{\pm.01}$ |
| DiffS.-lb | $0.22_{\pm.00}$ | $0.48_{\pm.01}$ | $0.62_{\pm.01}$ | $0.38_{\pm.01}$ | $0.35_{\pm.01}$ | $0.56_{\pm.01}$ | $0.30_{\pm.01}$ | $0.22_{\pm.00}$ | $0.47_{\pm.01}$ | $0.62_{\pm.01}$ | $0.39_{\pm.01}$ | $0.38_{\pm.01}$ | $0.59_{\pm.01}$ | $0.32_{\pm.01}$ |
| DiffS.-minp | $0.22_{\pm.00}$ | $0.48_{\pm.01}$ | $0.63_{\pm.01}$ | $0.39_{\pm.01}$ | $0.34_{\pm.01}$ | $0.56_{\pm.01}$ | $0.30_{\pm.01}$ | $0.22_{\pm.00}$ | $0.48_{\pm.01}$ | $0.62_{\pm.01}$ | $0.39_{\pm.01}$ | $0.37_{\pm.01}$ | $0.58_{\pm.01}$ | $0.32_{\pm.01}$ |
| *Temperature = 1.5* | | | | | | | | | | | | | | |
| DiffS.-cut | $0.23_{\pm.00}$ | $0.48_{\pm.01}$ | $0.63_{\pm.01}$ | $0.37_{\pm.01}$ | $0.28_{\pm.01}$ | $0.50_{\pm.01}$ | $0.26_{\pm.01}$ | $0.23_{\pm.00}$ | $0.48_{\pm.01}$ | $0.63_{\pm.01}$ | $0.35_{\pm.01}$ | $0.25_{\pm.01}$ | $0.46_{\pm.01}$ | $0.23_{\pm.01}$ |
| DiffS.-lb | $0.06_{\pm.00}$ | $0.16_{\pm.01}$ | $0.27_{\pm.01}$ | $0.64_{\pm.01}$ | $0.73_{\pm.00}$ | $0.89_{\pm.01}$ | $0.79_{\pm.01}$ | $0.20_{\pm.00}$ | $0.43_{\pm.01}$ | $0.57_{\pm.01}$ | $0.40_{\pm.01}$ | $0.52_{\pm.01}$ | $0.71_{\pm.01}$ | $0.45_{\pm.01}$ |
| DiffS.-minp | $0.20_{\pm.00}$ | $0.44_{\pm.01}$ | $0.58_{\pm.01}$ | $0.38_{\pm.01}$ | $0.51_{\pm.01}$ | $0.71_{\pm.01}$ | $0.45_{\pm.01}$ | $0.21_{\pm.00}$ | $0.46_{\pm.01}$ | $0.60_{\pm.01}$ | $0.35_{\pm.01}$ | $0.45_{\pm.01}$ | $0.67_{\pm.01}$ | $0.41_{\pm.01}$ |
| *Temperature = 2.0* | | | | | | | | | | | | | | |
| DiffS.-cut | $0.23_{\pm.00}$ | $0.48_{\pm.01}$ | $0.63_{\pm.01}$ | $0.38_{\pm.01}$ | $0.31_{\pm.01}$ | $0.54_{\pm.01}$ | $0.29_{\pm.01}$ | $0.23_{\pm.00}$ | $0.48_{\pm.01}$ | $0.63_{\pm.01}$ | $0.35_{\pm.01}$ | $0.25_{\pm.01}$ | $0.46_{\pm.01}$ | $0.24_{\pm.01}$ |
| DiffS.-lb | $0.01_{\pm.00}$ | $0.03_{\pm.00}$ | $0.13_{\pm.00}$ | $0.65_{\pm.01}$ | $0.66_{\pm.00}$ | $0.92_{\pm.01}$ | $0.92_{\pm.00}$ | $0.19_{\pm.00}$ | $0.43_{\pm.01}$ | $0.56_{\pm.01}$ | $0.43_{\pm.01}$ | $0.54_{\pm.01}$ | $0.73_{\pm.01}$ | $0.48_{\pm.01}$ |
| DiffS.-minp | $0.19_{\pm.00}$ | $0.42_{\pm.01}$ | $0.54_{\pm.01}$ | $0.42_{\pm.01}$ | $0.57_{\pm.00}$ | $0.76_{\pm.01}$ | $0.50_{\pm.01}$ | $0.21_{\pm.00}$ | $0.46_{\pm.01}$ | $0.59_{\pm.01}$ | $0.37_{\pm.01}$ | $0.47_{\pm.01}$ | $0.69_{\pm.01}$ | $0.42_{\pm.01}$ |
| *Temperature = 10.0* | | | | | | | | | | | | | | |
| DiffS.-cut | $0.17_{\pm.00}$ | $0.38_{\pm.01}$ | $0.51_{\pm.01}$ | $0.57_{\pm.01}$ | $0.53_{\pm.01}$ | $0.74_{\pm.01}$ | $0.50_{\pm.01}$ | $0.23_{\pm.00}$ | $0.48_{\pm.01}$ | $0.63_{\pm.01}$ | $0.35_{\pm.01}$ | $0.25_{\pm.01}$ | $0.47_{\pm.01}$ | $0.25_{\pm.01}$ |
| DiffS.-lb | $0.00_{\pm.00}$ | $0.02_{\pm.00}$ | $0.11_{\pm.00}$ | $0.74_{\pm.01}$ | $0.64_{\pm.00}$ | $0.93_{\pm.00}$ | $0.93_{\pm.00}$ | $0.17_{\pm.00}$ | $0.40_{\pm.01}$ | $0.52_{\pm.01}$ | $0.48_{\pm.01}$ | $0.59_{\pm.01}$ | $0.78_{\pm.01}$ | $0.54_{\pm.01}$ |
| DiffS.-minp | $0.00_{\pm.00}$ | $0.02_{\pm.00}$ | $0.11_{\pm.00}$ | $0.74_{\pm.01}$ | $0.64_{\pm.00}$ | $0.93_{\pm.00}$ | $0.93_{\pm.00}$ | $0.21_{\pm.00}$ | $0.45_{\pm.01}$ | $0.58_{\pm.01}$ | $0.38_{\pm.01}$ | $0.49_{\pm.01}$ | $0.72_{\pm.01}$ | $0.45_{\pm.01}$ |

Table 15: The mean and 95% confidence interval of quality and diversity metrics for the 5 samples generated by the instructed model with temperature before and after *DiffSampling* for each of the 1000 prompts from the XSum test set.

| Method | BEFORE | | | | | | | AFTER | | | | | | |
|---|---|---|---|---|---|---|---|---|---|---|---|---|---|---|
| | Quality | | | Pre | | Against-Greedy | | Quality | | | Per-Input | | Against-Greedy | |
| | R-1 | SIM | COH | EAD | SBERT | EAD | SBERT | R-1 | SIM | COH | EAD | SBERT | EAD | SBERT |
| *Temperature = 0.6* | | | | | | | | | | | | | | |
| DiffS.-cut | $0.21_{\pm.00}$ | $0.48_{\pm.01}$ | $0.70_{\pm.01}$ | $0.50_{\pm.01}$ | $0.29_{\pm.01}$ | $0.47_{\pm.01}$ | $0.27_{\pm.01}$ | $0.21_{\pm.00}$ | $0.49_{\pm.00}$ | $0.73_{\pm.00}$ | $0.37_{\pm.01}$ | $0.18_{\pm.00}$ | $0.30_{\pm.01}$ | $0.16_{\pm.01}$ |
| DiffS.-lb | $0.20_{\pm.00}$ | $0.46_{\pm.01}$ | $0.68_{\pm.01}$ | $0.54_{\pm.01}$ | $0.38_{\pm.01}$ | $0.50_{\pm.01}$ | $0.31_{\pm.01}$ | $0.20_{\pm.00}$ | $0.45_{\pm.01}$ | $0.65_{\pm.01}$ | $0.59_{\pm.01}$ | $0.42_{\pm.01}$ | $0.55_{\pm.01}$ | $0.34_{\pm.01}$ |
| DiffS.-minp | $0.20_{\pm.00}$ | $0.47_{\pm.01}$ | $0.69_{\pm.01}$ | $0.50_{\pm.01}$ | $0.34_{\pm.01}$ | $0.47_{\pm.01}$ | $0.28_{\pm.01}$ | $0.20_{\pm.00}$ | $0.47_{\pm.01}$ | $0.67_{\pm.01}$ | $0.55_{\pm.01}$ | $0.38_{\pm.01}$ | $0.52_{\pm.01}$ | $0.31_{\pm.01}$ |
| *Temperature = 1.5* | | | | | | | | | | | | | | |
| DiffS.-cut | $0.21_{\pm.00}$ | $0.49_{\pm.00}$ | $0.73_{\pm.01}$ | $0.41_{\pm.01}$ | $0.22_{\pm.00}$ | $0.36_{\pm.01}$ | $0.20_{\pm.01}$ | $0.21_{\pm.00}$ | $0.49_{\pm.00}$ | $0.73_{\pm.01}$ | $0.38_{\pm.01}$ | $0.19_{\pm.00}$ | $0.33_{\pm.01}$ | $0.18_{\pm.01}$ |
| DiffS.-lb | $0.03_{\pm.00}$ | $0.07_{\pm.00}$ | $0.18_{\pm.00}$ | $0.79_{\pm.01}$ | $0.75_{\pm.00}$ | $0.97_{\pm.00}$ | $0.89_{\pm.00}$ | $0.11_{\pm.00}$ | $0.23_{\pm.01}$ | $0.34_{\pm.01}$ | $0.75_{\pm.01}$ | $0.79_{\pm.00}$ | $0.89_{\pm.00}$ | $0.70_{\pm.01}$ |
| DiffS.-minp | $0.17_{\pm.00}$ | $0.38_{\pm.01}$ | $0.52_{\pm.01}$ | $0.77_{\pm.01}$ | $0.61_{\pm.00}$ | $0.80_{\pm.01}$ | $0.52_{\pm.01}$ | $0.19_{\pm.00}$ | $0.43_{\pm.01}$ | $0.60_{\pm.01}$ | $0.70_{\pm.01}$ | $0.52_{\pm.00}$ | $0.70_{\pm.01}$ | $0.44_{\pm.01}$ |
| *Temperature = 2.0* | | | | | | | | | | | | | | |
| DiffS.-cut | $0.21_{\pm.00}$ | $0.49_{\pm.00}$ | $0.72_{\pm.01}$ | $0.44_{\pm.01}$ | $0.24_{\pm.00}$ | $0.39_{\pm.01}$ | $0.22_{\pm.01}$ | $0.21_{\pm.00}$ | $0.49_{\pm.00}$ | $0.73_{\pm.01}$ | $0.38_{\pm.01}$ | $0.19_{\pm.00}$ | $0.33_{\pm.01}$ | $0.18_{\pm.01}$ |
| DiffS.-lb | $0.01_{\pm.00}$ | $0.03_{\pm.00}$ | $0.14_{\pm.00}$ | $0.80_{\pm.01}$ | $0.66_{\pm.00}$ | $0.98_{\pm.00}$ | $0.94_{\pm.00}$ | $0.09_{\pm.00}$ | $0.19_{\pm.01}$ | $0.29_{\pm.01}$ | $0.76_{\pm.01}$ | $0.82_{\pm.00}$ | $0.92_{\pm.00}$ | $0.75_{\pm.01}$ |
| DiffS.-minp | $0.13_{\pm.00}$ | $0.29_{\pm.01}$ | $0.41_{\pm.01}$ | $0.81_{\pm.01}$ | $0.73_{\pm.00}$ | $0.90_{\pm.00}$ | $0.64_{\pm.01}$ | $0.19_{\pm.00}$ | $0.42_{\pm.01}$ | $0.58_{\pm.01}$ | $0.72_{\pm.01}$ | $0.53_{\pm.00}$ | $0.72_{\pm.01}$ | $0.45_{\pm.01}$ |
| *Temperature = 10.0* | | | | | | | | | | | | | | |
| DiffS.-cut | $0.07_{\pm.00}$ | $0.17_{\pm.01}$ | $0.32_{\pm.01}$ | $0.53_{\pm.01}$ | $0.58_{\pm.01}$ | $0.73_{\pm.01}$ | $0.71_{\pm.01}$ | $0.21_{\pm.00}$ | $0.49_{\pm.00}$ | $0.73_{\pm.01}$ | $0.39_{\pm.01}$ | $0.19_{\pm.00}$ | $0.34_{\pm.01}$ | $0.19_{\pm.01}$ |
| DiffS.-lb | $0.00_{\pm.00}$ | $0.02_{\pm.00}$ | $0.12_{\pm.00}$ | $0.75_{\pm.01}$ | $0.63_{\pm.00}$ | $0.97_{\pm.00}$ | $0.95_{\pm.00}$ | $0.05_{\pm.00}$ | $0.13_{\pm.00}$ | $0.23_{\pm.01}$ | $0.77_{\pm.01}$ | $0.83_{\pm.00}$ | $0.95_{\pm.00}$ | $0.83_{\pm.01}$ |
| DiffS.-minp | $0.00_{\pm.00}$ | $0.02_{\pm.00}$ | $0.12_{\pm.00}$ | $0.75_{\pm.01}$ | $0.63_{\pm.00}$ | $0.97_{\pm.00}$ | $0.95_{\pm.00}$ | $0.18_{\pm.00}$ | $0.40_{\pm.01}$ | $0.55_{\pm.01}$ | $0.76_{\pm.01}$ | $0.57_{\pm.00}$ | $0.79_{\pm.01}$ | $0.50_{\pm.01}$ |

Table 16: The mean and 95% confidence interval of quality and diversity metrics for the 5 samples generated by the pre-trained model with temperature before and after *DiffSampling* for each of the 1000 prompts from the XSum test set.

temperature after preserves higher quality while reducing the increments in diversity. Vice versa, applying the temperature before has dramatic effects on accuracy, and only *DiffSampling-cut* achieves competitive results at very high $\tau$. In general, *DiffSampling-minp* with temperature applied after seems to provide the best balance between quality and diversity at different $\tau$.

The results for XSum are similar. As shown in Figure 8, the coherence of generated outputs degrades fast with $\tau > 1$ if temperature is applied before truncation. While min-$p$ and *DiffSampling-minp* have interesting performances at $\tau \leq 2.0$, for very high temperatures only *DiffSampling-cut* achieves comparable coherence. Overall, all sampling methods apart from *DiffSampling-cut* have very similar quality-diversity trade-offs, but *DiffSampling-minp* with temperature applied after seems to provide the best results considering the full range of temperature values.

Finally, Figure 9 shows the quality-diversity trade-offs in the case of the WritingPrompts dataset. Once more, *DiffSampling-cut*, especially with temperature applied after, better preserves quality at different $\tau$, while *DiffSampling-minp* with temperature applied after better balances quality and diversity for higher temperatures.

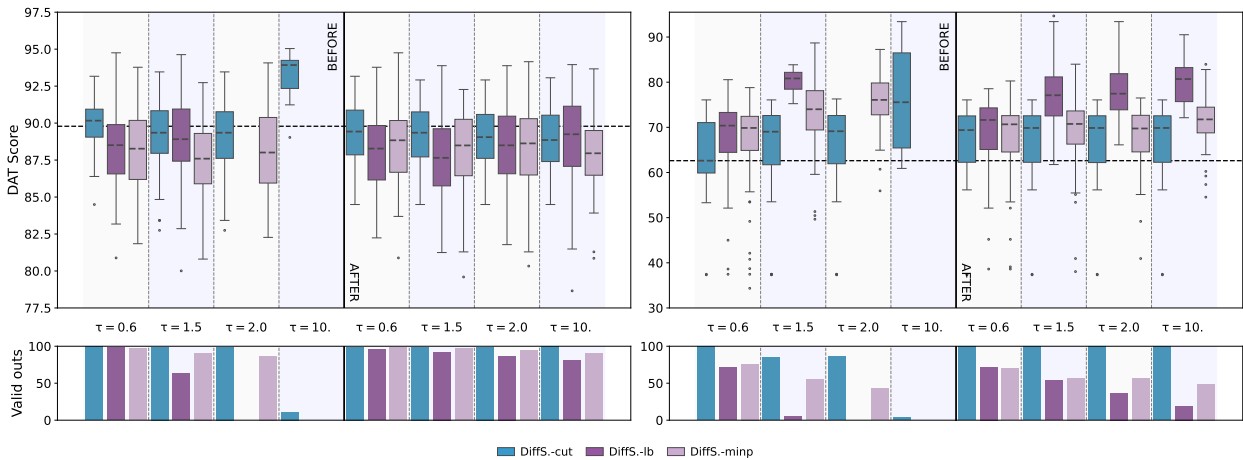

Figure 6: DAT scores and validity percentage of outputs with temperature scaling before and after the truncation for the instructed (left) and pre-trained (right) models. The dashed line represents the score of the greedy strategy.

| Model: | BEFORE | | | AFTER | | |
|---|---|---|---|---|---|---|
| Method | Quality | Per-Input Diversity | | Quality | Per-Input Diversity | |
| | COH | EAD | SBERT | COH | EAD | SBERT |
| *Temperature = 0.6* | | | | | | |
| DiffS.-cut | $0.43_{\pm.01}$ | $0.62_{\pm.00}$ | $0.18_{\pm.00}$ | $0.43_{\pm.01}$ | $0.63_{\pm.00}$ | $0.19_{\pm.00}$ |
| DiffS.-lb | $0.44_{\pm.01}$ | $0.67_{\pm.00}$ | $0.22_{\pm.00}$ | $0.43_{\pm.01}$ | $0.68_{\pm.00}$ | $0.23_{\pm.00}$ |
| DiffS.-minp | $0.43_{\pm.01}$ | $0.67_{\pm.00}$ | $0.21_{\pm.00}$ | $0.43_{\pm.01}$ | $0.68_{\pm.00}$ | $0.22_{\pm.00}$ |
| *Temperature = 1.5* | | | | | | |
| DiffS.-cut | $0.44_{\pm.01}$ | $0.64_{\pm.00}$ | $0.20_{\pm.00}$ | $0.43_{\pm.01}$ | $0.63_{\pm.00}$ | $0.19_{\pm.00}$ |
| DiffS.-lb | $0.17_{\pm.00}$ | $1.01_{\pm.00}$ | $0.15_{\pm.00}$ | $0.40_{\pm.01}$ | $0.89_{\pm.00}$ | $0.33_{\pm.00}$ |
| DiffS.-minp | $0.42_{\pm.01}$ | $0.79_{\pm.00}$ | $0.27_{\pm.00}$ | $0.43_{\pm.01}$ | $0.73_{\pm.00}$ | $0.24_{\pm.00}$ |
| *Temperature = 2.0* | | | | | | |
| DiffS.-cut | $0.44_{\pm.01}$ | $0.65_{\pm.00}$ | $0.21_{\pm.00}$ | $0.43_{\pm.01}$ | $0.64_{\pm.00}$ | $0.19_{\pm.00}$ |
| DiffS.-lb | $0.11_{\pm.00}$ | $1.02_{\pm.00}$ | $0.12_{\pm.00}$ | $0.38_{\pm.01}$ | $0.93_{\pm.00}$ | $0.33_{\pm.00}$ |
| DiffS.-minp | $0.39_{\pm.01}$ | $0.92_{\pm.00}$ | $0.34_{\pm.00}$ | $0.43_{\pm.01}$ | $0.74_{\pm.00}$ | $0.24_{\pm.00}$ |
| *Temperature = 10.0* | | | | | | |
| DiffS.-cut | $0.23_{\pm.01}$ | $0.94_{\pm.00}$ | $0.24_{\pm.00}$ | $0.43_{\pm.01}$ | $0.64_{\pm.00}$ | $0.20_{\pm.00}$ |
| DiffS.-lb | $0.10_{\pm.00}$ | $1.02_{\pm.00}$ | $0.14_{\pm.00}$ | $0.32_{\pm.01}$ | $0.98_{\pm.00}$ | $0.29_{\pm.00}$ |
| DiffS.-minp | $0.10_{\pm.00}$ | $1.02_{\pm.00}$ | $0.14_{\pm.00}$ | $0.42_{\pm.01}$ | $0.77_{\pm.00}$ | $0.25_{\pm.00}$ |

Table 17: Quality and diversity of results for story generation with the instructed model over 3 seeds. The mean and standard error of the final score for each run are reported for cross-input diversity, whereas the mean and 95% confidence interval for the full set of answers are reported for the other metrics.

# G  Additional Experiments

## G.1  Divergent Association Task over All Nouns

While the classic DAT implementation only considers outputs containing at least 7 distinct nouns (out of 10), and computes the score based on the first 7 nouns, it is informative to examine performance when all 10 nouns are distinct and the score is computed over the full set. Indeed, the standard implementation may be overly conservative, allowing methods that fail to meet the full requirements to still receive a final score. Figure 10 presents these scores for all baselines and our methods across the five temperatures considered. Although the DAT scores are largely consistent with those in Figure 5, the number of valid outputs changes.

| Model: | BEFORE | | | AFTER | | |
|---|---|---|---|---|---|---|
| Method | Quality | Per-Input Diversity | | Quality | Per-Input Diversity | |
| | COH | EAD | SBERT | COH | EAD | SBERT |
| *Temperature = 0.6* | | | | | | |
| DiffS.-cut | $0.60_{\pm.01}$ | $0.14_{\pm.00}$ | $0.29_{\pm.01}$ | $0.60_{\pm.01}$ | $0.16_{\pm.00}$ | $0.31_{\pm.01}$ |
| DiffS.-lb | $0.55_{\pm.01}$ | $0.22_{\pm.00}$ | $0.43_{\pm.00}$ | $0.53_{\pm.01}$ | $0.27_{\pm.00}$ | $0.45_{\pm.00}$ |
| DiffS.-minp | $0.57_{\pm.01}$ | $0.20_{\pm.00}$ | $0.41_{\pm.00}$ | $0.54_{\pm.01}$ | $0.24_{\pm.00}$ | $0.43_{\pm.00}$ |
| *Temperature = 1.5* | | | | | | |
| DiffS.-cut | $0.59_{\pm.01}$ | $0.17_{\pm.00}$ | $0.33_{\pm.00}$ | $0.60_{\pm.01}$ | $0.16_{\pm.00}$ | $0.31_{\pm.01}$ |
| DiffS.-lb | $0.14_{\pm.00}$ | $1.00_{\pm.00}$ | $0.32_{\pm.00}$ | $0.31_{\pm.01}$ | $0.92_{\pm.00}$ | $0.63_{\pm.00}$ |
| DiffS.-minp | $0.42_{\pm.01}$ | $0.75_{\pm.00}$ | $0.55_{\pm.00}$ | $0.48_{\pm.01}$ | $0.46_{\pm.00}$ | $0.49_{\pm.00}$ |
| *Temperature = 2.0* | | | | | | |
| DiffS.-cut | $0.59_{\pm.01}$ | $0.18_{\pm.00}$ | $0.34_{\pm.00}$ | $0.60_{\pm.01}$ | $0.16_{\pm.00}$ | $0.31_{\pm.00}$ |
| DiffS.-lb | $0.11_{\pm.00}$ | $1.00_{\pm.00}$ | $0.32_{\pm.00}$ | $0.26_{\pm.01}$ | $0.97_{\pm.00}$ | $0.52_{\pm.00}$ |
| DiffS.-minp | $0.34_{\pm.01}$ | $0.86_{\pm.00}$ | $0.63_{\pm.00}$ | $0.47_{\pm.01}$ | $0.53_{\pm.00}$ | $0.50_{\pm.00}$ |
| *Temperature = 10.0* | | | | | | |
| DiffS.-cut | $0.19_{\pm.01}$ | $0.69_{\pm.01}$ | $0.43_{\pm.00}$ | $0.60_{\pm.01}$ | $0.17_{\pm.00}$ | $0.31_{\pm.01}$ |
| DiffS.-lb | $0.10_{\pm.00}$ | $1.00_{\pm.00}$ | $0.36_{\pm.00}$ | $0.19_{\pm.00}$ | $1.00_{\pm.00}$ | $0.37_{\pm.00}$ |
| DiffS.-minp | $0.10_{\pm.00}$ | $1.00_{\pm.00}$ | $0.36_{\pm.00}$ | $0.45_{\pm.01}$ | $0.68_{\pm.00}$ | $0.52_{\pm.00}$ |

Table 18: Quality and diversity of results for story generation with the pre-trained model over 3 seeds. The mean and standard error of the final score for each run are reported for cross-input diversity, whereas the mean and 95% confidence interval for the full set of answers are reported for the other metrics.

Notably, the difference between *DiffSampling* methods and the baselines becomes even more pronounced, particularly at temperatures 1.0 and 1.5. This further confirms that our strategy is more robust and better adheres to the task requirements.

## G.2   Ablation Study on the Lower Bound

| Dataset: | GSM8K | | | | | MATH | | | | |
|---|---|---|---|---|---|---|---|---|---|---|
| Method | Accuracy | Cross-Input | | Against-Greedy | | Accuracy | Cross-Input | | Against-Greedy | |
| DiffSampling-lb | | EAD | SBERT | EAD | SBERT | | EAD | SBERT | EAD | SBERT |
| $p_{lb} = 0.0$ | $66.36_{\pm.23}$ | $2.04_{\pm.00}$ | $0.64_{\pm.00}$ | $0.14_{\pm.00}$ | $0.02_{\pm.00}$ | $21.38_{\pm.20}$ | $5.71_{\pm.01}$ | $0.80_{\pm.00}$ | $0.27_{\pm.00}$ | $0.07_{\pm.00}$ |
| $p_{lb} = 0.1$ | $66.46_{\pm.34}$ | $2.05_{\pm.00}$ | $0.64_{\pm.00}$ | $0.14_{\pm.00}$ | $0.02_{\pm.00}$ | $20.95_{\pm.20}$ | $5.72_{\pm.01}$ | $0.80_{\pm.00}$ | $0.27_{\pm.00}$ | $0.07_{\pm.00}$ |
| $p_{lb} = 0.2$ | $66.46_{\pm.34}$ | $2.05_{\pm.00}$ | $0.64_{\pm.00}$ | $0.14_{\pm.00}$ | $0.02_{\pm.00}$ | $20.95_{\pm.20}$ | $5.72_{\pm.01}$ | $0.80_{\pm.00}$ | $0.27_{\pm.00}$ | $0.07_{\pm.00}$ |
| $p_{lb} = 0.3$ | $66.79_{\pm.40}$ | $2.04_{\pm.00}$ | $0.64_{\pm.00}$ | $0.14_{\pm.00}$ | $0.02_{\pm.00}$ | $21.30_{\pm.08}$ | $5.73_{\pm.00}$ | $0.80_{\pm.00}$ | $0.27_{\pm.00}$ | $0.07_{\pm.00}$ |
| $p_{lb} = 0.4$ | $66.57_{\pm.39}$ | $2.06_{\pm.00}$ | $0.64_{\pm.00}$ | $0.14_{\pm.00}$ | $0.02_{\pm.00}$ | $21.08_{\pm.11}$ | $5.73_{\pm.02}$ | $0.80_{\pm.00}$ | $0.27_{\pm.00}$ | $0.07_{\pm.00}$ |
| $p_{lb} = 0.5$ | $67.17_{\pm.41}$ | $2.04_{\pm.00}$ | $0.64_{\pm.00}$ | $0.15_{\pm.00}$ | $0.02_{\pm.00}$ | $21.18_{\pm.41}$ | $5.74_{\pm.01}$ | $0.80_{\pm.00}$ | $0.28_{\pm.00}$ | $0.08_{\pm.00}$ |
| $p_{lb} = 0.6$ | $66.67_{\pm.37}$ | $2.05_{\pm.00}$ | $0.64_{\pm.00}$ | $0.16_{\pm.00}$ | $0.02_{\pm.00}$ | $21.18_{\pm.22}$ | $5.79_{\pm.02}$ | $0.80_{\pm.00}$ | $0.30_{\pm.00}$ | $0.09_{\pm.00}$ |
| $p_{lb} = 0.7$ | $65.58_{\pm.19}$ | $2.06_{\pm.00}$ | $0.64_{\pm.00}$ | $0.18_{\pm.00}$ | $0.03_{\pm.00}$ | $21.14_{\pm.15}$ | $5.86_{\pm.01}$ | $0.80_{\pm.00}$ | $0.32_{\pm.00}$ | $0.09_{\pm.00}$ |
| $p_{lb} = 0.8$ | $66.92_{\pm.08}$ | $2.07_{\pm.00}$ | $0.64_{\pm.00}$ | $0.20_{\pm.00}$ | $0.03_{\pm.00}$ | $20.78_{\pm.14}$ | $6.00_{\pm.01}$ | $0.80_{\pm.00}$ | $0.35_{\pm.00}$ | $0.10_{\pm.00}$ |
| $p_{lb} = 0.9$ | $65.18_{\pm.65}$ | $2.09_{\pm.01}$ | $0.64_{\pm.00}$ | $0.23_{\pm.00}$ | $0.03_{\pm.00}$ | $20.20_{\pm.08}$ | $6.11_{\pm.02}$ | $0.80_{\pm.00}$ | $0.37_{\pm.00}$ | $0.10_{\pm.00}$ |
| $p_{lb} = 0.95$ | $64.82_{\pm.31}$ | $2.09_{\pm.01}$ | $0.64_{\pm.00}$ | $0.24_{\pm.00}$ | $0.03_{\pm.00}$ | $20.24_{\pm.19}$ | $6.21_{\pm.01}$ | $0.80_{\pm.00}$ | $0.37_{\pm.00}$ | $0.11_{\pm.00}$ |
| $p_{lb} = 1.0$ | $64.87_{\pm.20}$ | $2.12_{\pm.00}$ | $0.64_{\pm.00}$ | $0.25_{\pm.00}$ | $0.04_{\pm.00}$ | $19.46_{\pm.19}$ | $6.36_{\pm.01}$ | $0.80_{\pm.00}$ | $0.39_{\pm.00}$ | $0.11_{\pm.00}$ |

Table 19: Ablation study on the $p_{lb}$ value over 3 seeds for the GSM8K (left) and MATH (right) test sets. The mean and standard error of the final score for each run are reported for accuracy and cross-input diversity, whereas the mean and 95% confidence interval for the full set of answers are reported for against-greedy diversity.

We also conducted experiments on the four aforementioned case studies, varying the lower bound of the critical mass. Table 19 reports the results for the math problem-solving tasks, considering the GSM8K (left side) and MATH (right side) test sets. As expected, the against-greedy diversity scores and cross-input EAD increase together with $p_{lb}$; instead, while accuracy tends to decrease with higher lower bounds, the differences are not significant, and even a quite high value (e.g., 0.8) achieves competitive results.

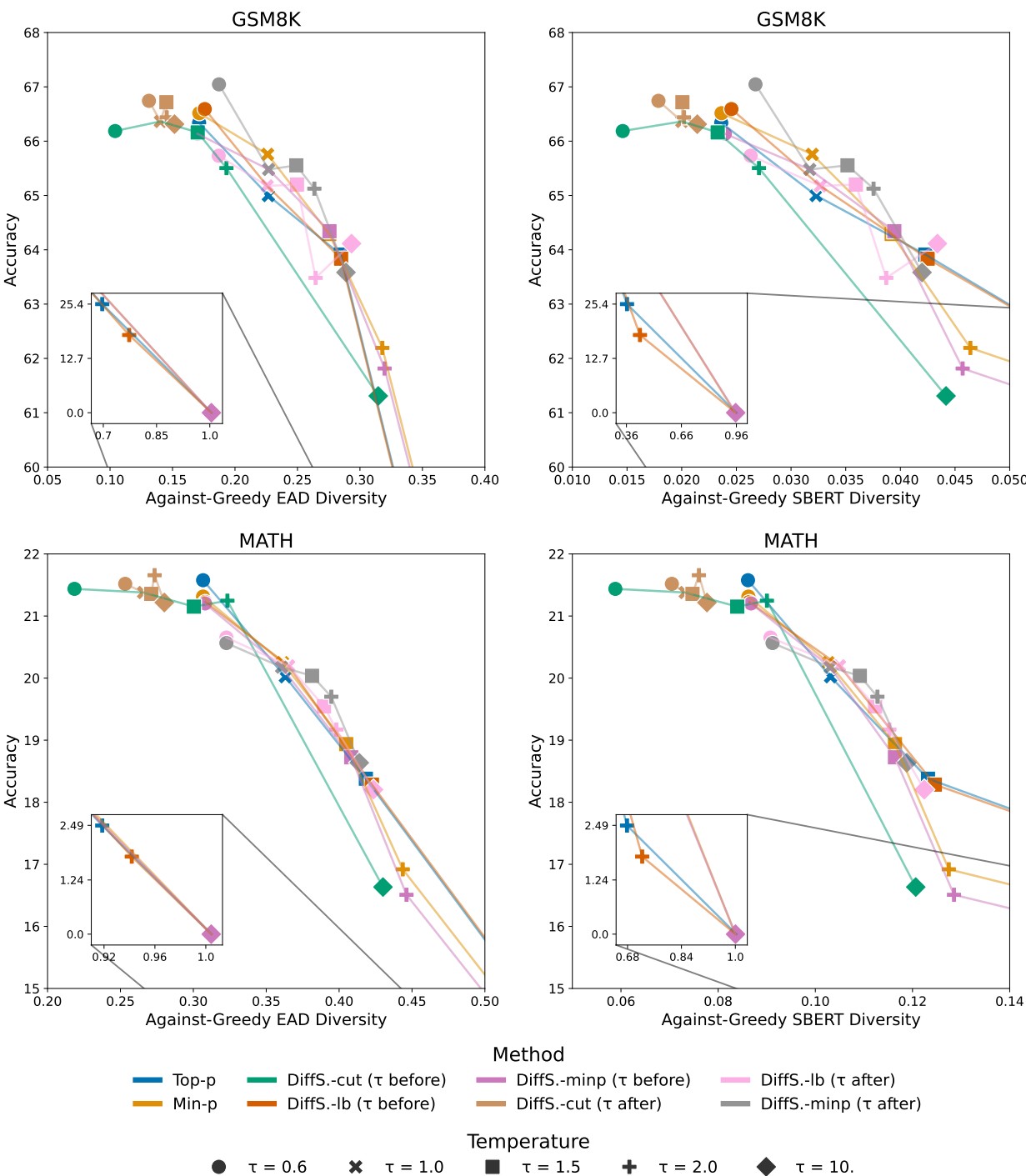

Figure 7: Quality-diversity trade-offs for GSM8K and MATH datasets at five different temperatures for two baselines (top-*p* and min-*p*) and our three *DiffSampling* methods, with temperature applied either before or after the truncation. Each point represents the mean score across the entire dataset and three different seeds.

Table 20 reports the results for the extreme summarization task for both instructed (left side) and pre-trained (right side) models. Again, diversity scores are directly correlated with the lower bound. Instead, qualitative metrics do not vary much for the instructed model, while constantly decreasing for the pre-trained model

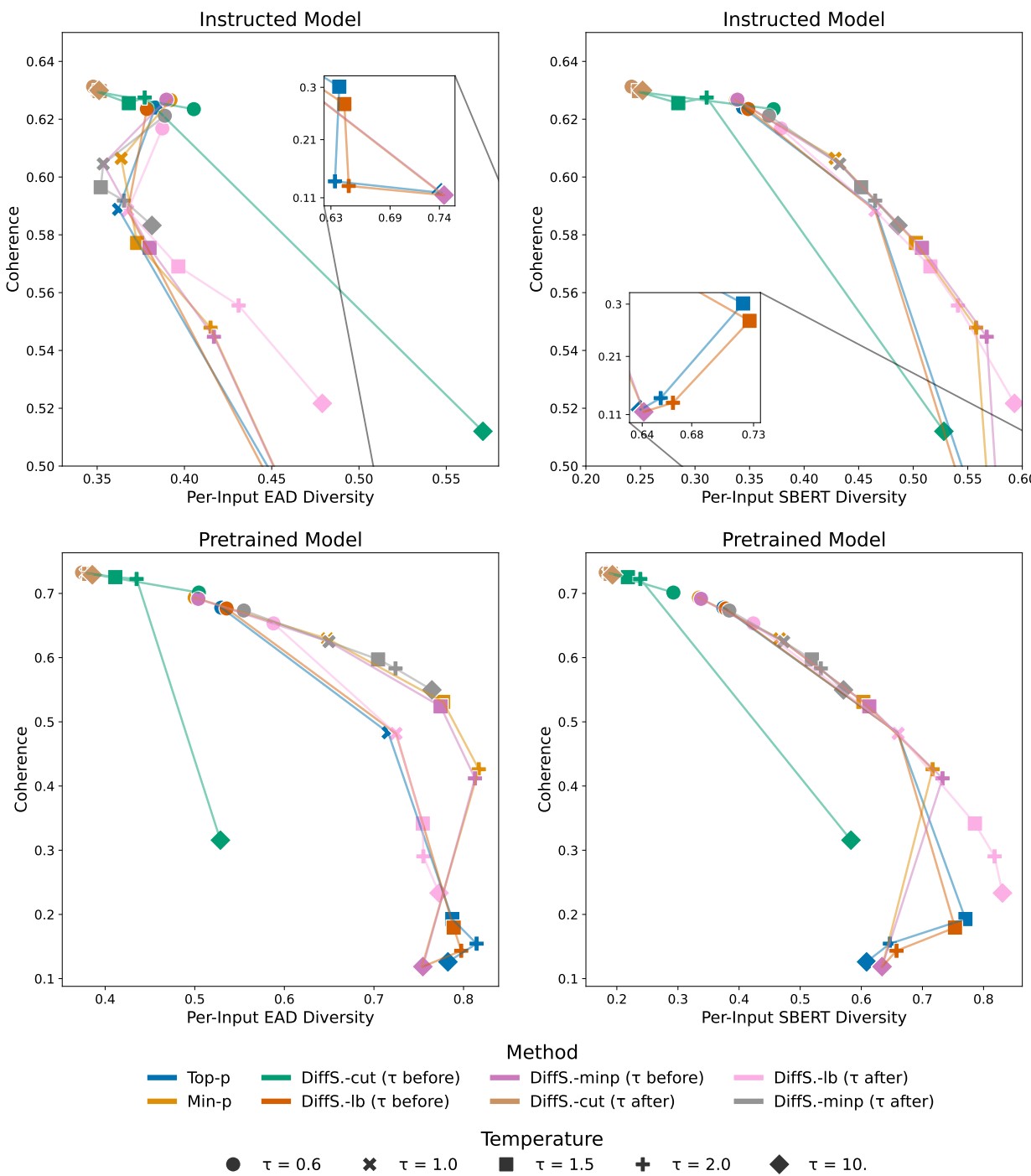

Figure 8: Quality-diversity trade-offs for the XSum dataset with the instructed and pre-trained models at five different temperatures for two baselines (top-$p$ and min-$p$) and our three *DiffSampling* methods, with temperature applied either before or after the truncation. Each point represents the mean score across all the outputs generated from the same 1000 randomly sampled inputs.

with increasing $p_{lb}$. In this situation, the choice of $p_{lb}$ is relevant and requires us to decide whether to trade off quality or diversity.

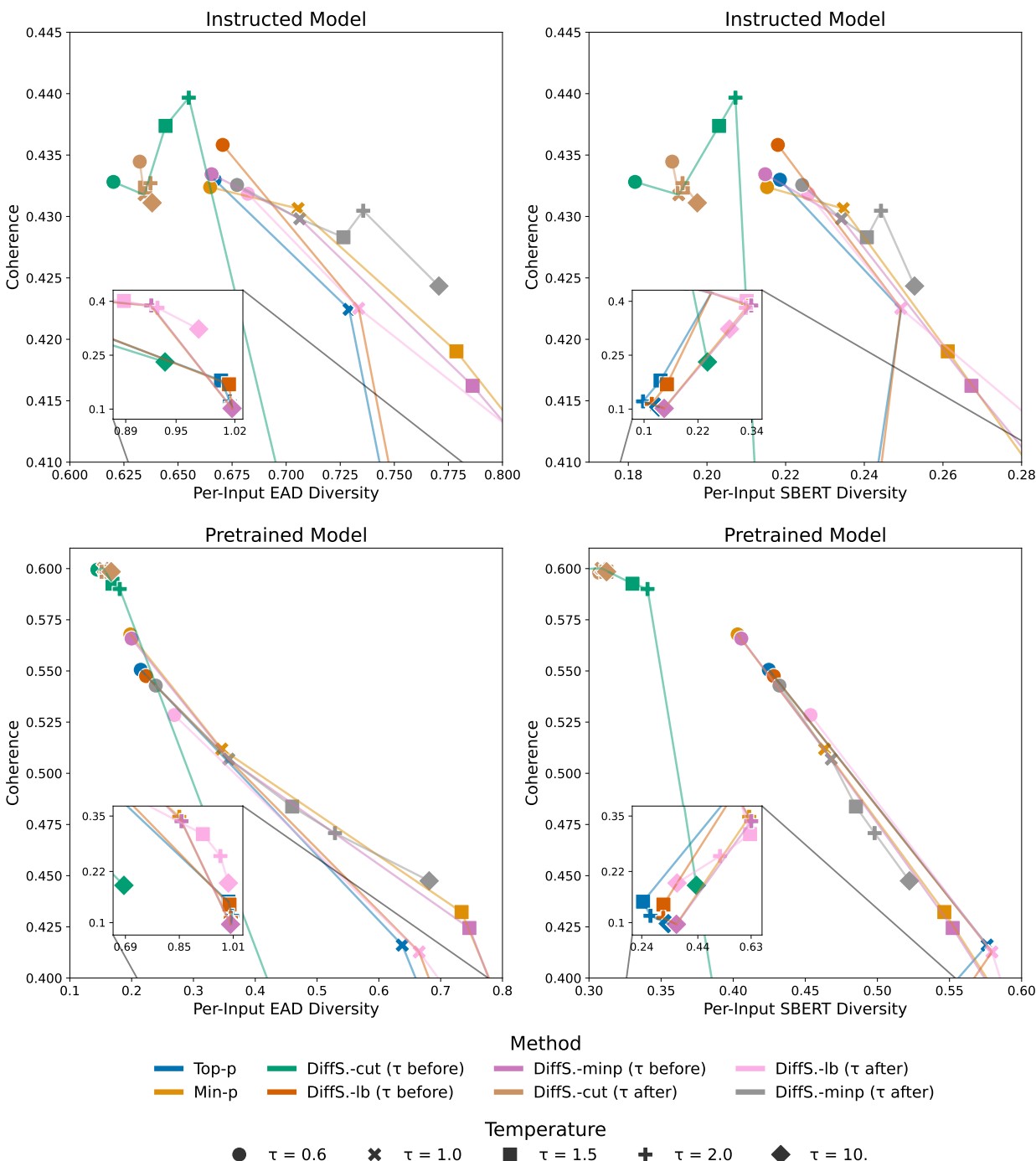

Figure 9: Quality-diversity trade-offs for the WritingPrompts dataset with the instructed and pre-trained models at five different temperatures for two baselines (top-*p* and min-*p*) and our three *DiffSampling* methods, with temperature applied either before or after the truncation. Each point represents the mean score across all the outputs generated from the same 500 randomly sampled inputs.

Table 21 reports the results for the story generation task for both instructed (left side) and pre-trained (right side) models. Coherence decreases at higher $p_{lb}$ values, but this effect is significant only for the pre-trained model. However, both diversity scores are directly correlated with the lower bound, especially at high values.

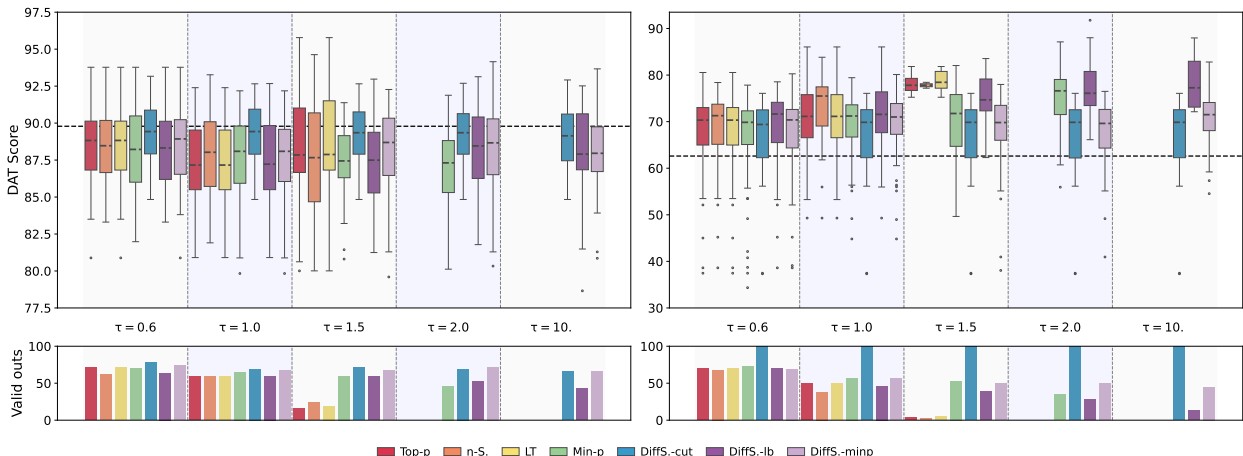

Figure 10: DAT scores over all 10 nouns for our methods and the baselines for the instructed (left) and pre-trained (right) model with different temperature values, together with the number of valid outputs produced by each sampling strategy. The dashed line represents the score of the greedy strategy.

| Model: | RLHF-instructed | | | | | | | Pre-trained | | | | | | |
|---|---|---|---|---|---|---|---|---|---|---|---|---|---|---|
| Method | Quality | | | Per-Input | | Against-Greedy | | Quality | | | Per-Input | | Against-Greedy | |
| DiffSampling-lb | R-1 | SIM | COH | EAD | SBERT | EAD | SBERT | R-1 | SIM | COH | EAD | SBERT | EAD | SBERT |
| $p_{lb} = 0.0$ | $0.23_{\pm.00}$ | $0.48_{\pm.01}$ | $0.63_{\pm.01}$ | $0.35_{\pm.01}$ | $0.25_{\pm.01}$ | $0.45_{\pm.01}$ | $0.23_{\pm.01}$ | $0.21_{\pm.00}$ | $0.49_{\pm.00}$ | $0.73_{\pm.00}$ | $0.38_{\pm.01}$ | $0.19_{\pm.00}$ | $0.32_{\pm.01}$ | $0.17_{\pm.01}$ |
| $p_{lb} = 0.1$ | $0.23_{\pm.00}$ | $0.48_{\pm.01}$ | $0.63_{\pm.01}$ | $0.35_{\pm.01}$ | $0.25_{\pm.01}$ | $0.45_{\pm.01}$ | $0.23_{\pm.01}$ | $0.21_{\pm.00}$ | $0.49_{\pm.00}$ | $0.73_{\pm.01}$ | $0.39_{\pm.01}$ | $0.20_{\pm.01}$ | $0.34_{\pm.01}$ | $0.19_{\pm.01}$ |
| $p_{lb} = 0.2$ | $0.23_{\pm.00}$ | $0.48_{\pm.01}$ | $0.63_{\pm.01}$ | $0.35_{\pm.01}$ | $0.25_{\pm.01}$ | $0.45_{\pm.01}$ | $0.23_{\pm.01}$ | $0.21_{\pm.00}$ | $0.48_{\pm.01}$ | $0.71_{\pm.01}$ | $0.43_{\pm.01}$ | $0.27_{\pm.01}$ | $0.40_{\pm.01}$ | $0.24_{\pm.01}$ |
| $p_{lb} = 0.3$ | $0.23_{\pm.00}$ | $0.48_{\pm.01}$ | $0.63_{\pm.01}$ | $0.37_{\pm.01}$ | $0.26_{\pm.01}$ | $0.47_{\pm.01}$ | $0.24_{\pm.01}$ | $0.21_{\pm.00}$ | $0.46_{\pm.01}$ | $0.68_{\pm.01}$ | $0.48_{\pm.01}$ | $0.35_{\pm.01}$ | $0.48_{\pm.01}$ | $0.30_{\pm.01}$ |
| $p_{lb} = 0.4$ | $0.23_{\pm.00}$ | $0.48_{\pm.01}$ | $0.63_{\pm.01}$ | $0.39_{\pm.01}$ | $0.29_{\pm.01}$ | $0.50_{\pm.01}$ | $0.27_{\pm.01}$ | $0.20_{\pm.00}$ | $0.45_{\pm.01}$ | $0.66_{\pm.01}$ | $0.54_{\pm.01}$ | $0.40_{\pm.01}$ | $0.54_{\pm.01}$ | $0.34_{\pm.01}$ |
| $p_{lb} = 0.5$ | $0.22_{\pm.00}$ | $0.48_{\pm.01}$ | $0.62_{\pm.01}$ | $0.39_{\pm.01}$ | $0.33_{\pm.01}$ | $0.54_{\pm.01}$ | $0.30_{\pm.01}$ | $0.20_{\pm.00}$ | $0.44_{\pm.01}$ | $0.63_{\pm.01}$ | $0.58_{\pm.01}$ | $0.46_{\pm.01}$ | $0.59_{\pm.01}$ | $0.38_{\pm.01}$ |
| $p_{lb} = 0.6$ | $0.22_{\pm.00}$ | $0.47_{\pm.01}$ | $0.62_{\pm.01}$ | $0.40_{\pm.01}$ | $0.37_{\pm.01}$ | $0.58_{\pm.01}$ | $0.33_{\pm.01}$ | $0.19_{\pm.00}$ | $0.42_{\pm.01}$ | $0.60_{\pm.01}$ | $0.63_{\pm.01}$ | $0.51_{\pm.01}$ | $0.64_{\pm.01}$ | $0.42_{\pm.01}$ |
| $p_{lb} = 0.7$ | $0.22_{\pm.00}$ | $0.46_{\pm.01}$ | $0.61_{\pm.01}$ | $0.37_{\pm.01}$ | $0.41_{\pm.01}$ | $0.62_{\pm.01}$ | $0.36_{\pm.01}$ | $0.19_{\pm.00}$ | $0.40_{\pm.01}$ | $0.57_{\pm.01}$ | $0.67_{\pm.01}$ | $0.56_{\pm.01}$ | $0.69_{\pm.01}$ | $0.46_{\pm.01}$ |
| $p_{lb} = 0.8$ | $0.21_{\pm.00}$ | $0.45_{\pm.01}$ | $0.60_{\pm.01}$ | $0.36_{\pm.01}$ | $0.45_{\pm.01}$ | $0.64_{\pm.01}$ | $0.39_{\pm.01}$ | $0.18_{\pm.00}$ | $0.38_{\pm.01}$ | $0.53_{\pm.01}$ | $0.70_{\pm.01}$ | $0.60_{\pm.01}$ | $0.73_{\pm.01}$ | $0.50_{\pm.01}$ |
| $p_{lb} = 0.9$ | $0.21_{\pm.00}$ | $0.45_{\pm.01}$ | $0.59_{\pm.01}$ | $0.37_{\pm.01}$ | $0.47_{\pm.01}$ | $0.67_{\pm.01}$ | $0.41_{\pm.01}$ | $0.16_{\pm.00}$ | $0.34_{\pm.01}$ | $0.48_{\pm.01}$ | $0.72_{\pm.01}$ | $0.66_{\pm.01}$ | $0.77_{\pm.01}$ | $0.55_{\pm.01}$ |
| $p_{lb} = 0.95$ | $0.21_{\pm.00}$ | $0.45_{\pm.01}$ | $0.58_{\pm.01}$ | $0.38_{\pm.01}$ | $0.48_{\pm.01}$ | $0.68_{\pm.01}$ | $0.42_{\pm.01}$ | $0.15_{\pm.00}$ | $0.32_{\pm.01}$ | $0.46_{\pm.01}$ | $0.73_{\pm.01}$ | $0.69_{\pm.00}$ | $0.80_{\pm.01}$ | $0.58_{\pm.01}$ |
| $p_{lb} = 1.0$ | $0.20_{\pm.00}$ | $0.44_{\pm.01}$ | $0.58_{\pm.01}$ | $0.40_{\pm.01}$ | $0.50_{\pm.01}$ | $0.70_{\pm.01}$ | $0.43_{\pm.01}$ | $0.14_{\pm.00}$ | $0.29_{\pm.01}$ | $0.42_{\pm.01}$ | $0.74_{\pm.01}$ | $0.74_{\pm.00}$ | $0.83_{\pm.01}$ | $0.62_{\pm.01}$ |

Table 20: Ablation study on the $p_{lb}$ value over the 5 outputs sampled for each of the 1000 prompts from the XSum dataset for the instructed model (left) and the pre-trained model (right). The mean and 95% confidence interval are reported for all the metrics.

Finally, Figure 11 reports the results for the divergent association task. As we would expect, the DAT score changes almost linearly between that for a lower bound of 0 (that means *DiffSampling-cut*) and 1 (that means *standard* sampling). Interestingly, the number of correct answers by the non-instructed model drops constantly, while it remains consistently higher in the case of the instructed model.

To sum up, when greediness is desirable, a lower value of $p_{lb}$ can lead to high quality and diversity; otherwise, increasing $p_{lb}$ improves diversity, but the cost in terms of validity is not negligible and requires careful consideration. We suggest practitioners select the most appropriate $p_{lb}$ value by running it on a validation set if available, and otherwise lie in the $[0.8, 0.95]$ range, which has shown competitive results on both quality and diversity metrics.

### G.3 Ablation Study on the Dynamic Upper Bound

Finally, we conducted experiments on the four aforementioned case studies, varying the dynamic upper bound of the truncated tokens $p_{min}$.

Table 22 reports the results for the math problem-solving tasks, considering the GSM8K (left side) and MATH (right side) test sets. As expected, the against-greedy diversity scores and cross-input EAD decrease together with $p_{min}$, plateauing at $p_{min} = 0.6$ (from that on, results are comparable with *DiffSampling-cut*);

| Model: | RLHF-instructed | | | Pre-trained | | |
|---|---|---|---|---|---|---|
| Method | Quality | Per-Input Diversity | | Quality | Per-Input Diversity | |
| DiffSampling-lb | COH | EAD | SBERT | COH | EAD | SBERT |
| $p_{lb} = 0.0$ | $0.43_{\pm.01}$ | $0.63_{\pm.00}$ | $0.19_{\pm.00}$ | $0.60_{\pm.01}$ | $0.15_{\pm.00}$ | $0.31_{\pm.01}$ |
| $p_{lb} = 0.1$ | $0.43_{\pm.01}$ | $0.63_{\pm.00}$ | $0.19_{\pm.00}$ | $0.60_{\pm.01}$ | $0.16_{\pm.00}$ | $0.32_{\pm.01}$ |
| $p_{lb} = 0.2$ | $0.43_{\pm.01}$ | $0.64_{\pm.00}$ | $0.20_{\pm.00}$ | $0.58_{\pm.01}$ | $0.16_{\pm.00}$ | $0.37_{\pm.01}$ |
| $p_{lb} = 0.3$ | $0.43_{\pm.01}$ | $0.65_{\pm.00}$ | $0.20_{\pm.00}$ | $0.56_{\pm.01}$ | $0.18_{\pm.00}$ | $0.41_{\pm.00}$ |
| $p_{lb} = 0.4$ | $0.43_{\pm.01}$ | $0.65_{\pm.00}$ | $0.21_{\pm.00}$ | $0.54_{\pm.01}$ | $0.19_{\pm.00}$ | $0.43_{\pm.00}$ |
| $p_{lb} = 0.5$ | $0.43_{\pm.01}$ | $0.66_{\pm.00}$ | $0.22_{\pm.00}$ | $0.52_{\pm.01}$ | $0.23_{\pm.00}$ | $0.46_{\pm.00}$ |
| $p_{lb} = 0.6$ | $0.43_{\pm.01}$ | $0.67_{\pm.00}$ | $0.22_{\pm.00}$ | $0.49_{\pm.01}$ | $0.28_{\pm.00}$ | $0.49_{\pm.00}$ |
| $p_{lb} = 0.7$ | $0.43_{\pm.01}$ | $0.69_{\pm.00}$ | $0.23_{\pm.00}$ | $0.47_{\pm.01}$ | $0.33_{\pm.00}$ | $0.51_{\pm.00}$ |
| $p_{lb} = 0.8$ | $0.43_{\pm.01}$ | $0.70_{\pm.00}$ | $0.24_{\pm.00}$ | $0.44_{\pm.01}$ | $0.47_{\pm.00}$ | $0.54_{\pm.00}$ |
| $p_{lb} = 0.9$ | $0.42_{\pm.01}$ | $0.73_{\pm.00}$ | $0.25_{\pm.00}$ | $0.41_{\pm.01}$ | $0.67_{\pm.00}$ | $0.58_{\pm.00}$ |
| $p_{lb} = 0.95$ | $0.42_{\pm.01}$ | $0.78_{\pm.00}$ | $0.27_{\pm.00}$ | $0.39_{\pm.01}$ | $0.76_{\pm.00}$ | $0.61_{\pm.00}$ |
| $p_{lb} = 1.0$ | $0.41_{\pm.01}$ | $0.88_{\pm.00}$ | $0.33_{\pm.00}$ | $0.35_{\pm.01}$ | $0.84_{\pm.00}$ | $0.64_{\pm.00}$ |

Table 21: Ablation study on the $p_{lb}$ value over 3 seeds for the WritingPrompts dataset for the instructed model (left) and the pre-trained model (right). The mean and standard error of the final score for each run are reported for cross-input diversity, whereas the mean and 95% confidence interval for the full set of answers are reported for the other metrics.

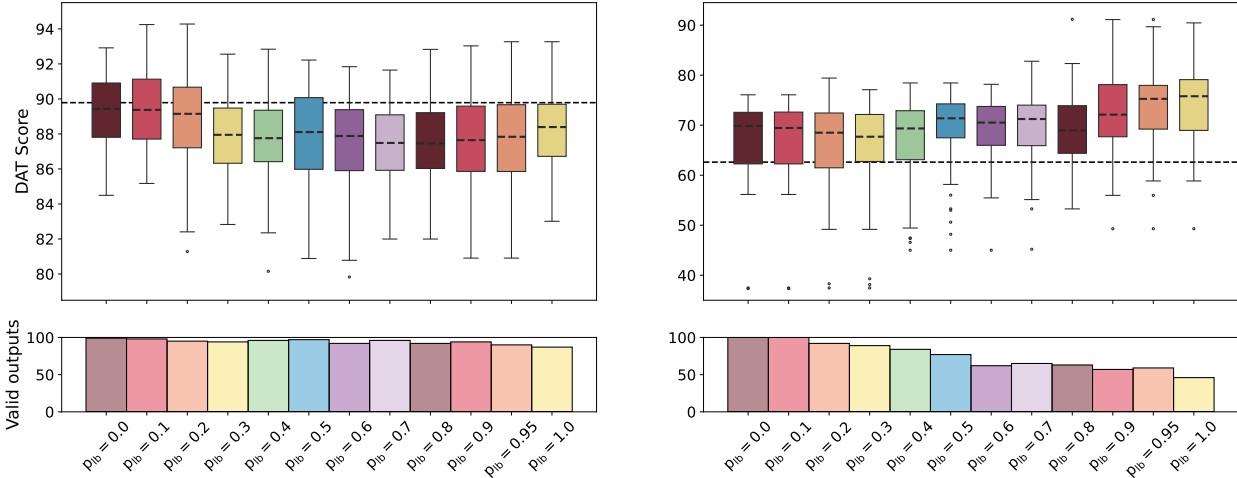

Figure 11: DAT scores and output validity percentage for *DiffSampling-lb* when varying the $p_{lb}$ parameter for the instructed (left) and pre-trained (right) models. The dashed line represents the score of the greedy strategy.

specularly, accuracy is lower at smaller $p_{min}$, but the instructed model reaches a competitive score even at $p_{min} = 0.3$.

The same holds for XSum as well. As shown in Table 23, diversity decreases when increasing $p_{min}$ and plateaus at 0.5, while quality rapidly increases for the pre-trained model and is almost constant for the instructed model.

As reported in Table 24, in the case of story generation, diversity rapidly drops when increasing $p_{min}$ and plateaus around 0.5, while quality increases for the pre-trained model and is almost constant for the instructed model.

The same considerations are even more apparent for the divergent association task with Figure 12. While behaving differently for the instructed and pre-trained models, the DAT score plateaus around $p_{min} = 0.5$. On the other hand, the percentage of valid outputs is close to 100% for all $p_{min}$ values when considering the instructed model, and linearly increases when considering the pre-trained model.

| Dataset: | GSM8K | | | | | MATH | | | | |
|---|---|---|---|---|---|---|---|---|---|---|
| Method | Accuracy | Cross-Input | | Against-Greedy | | Accuracy | Cross-Input | | Against-Greedy | |
| DiffSampling-minp | | EAD | SBERT | EAD | SBERT | | EAD | SBERT | EAD | SBERT |
| $p_{min} = 0.0$ | $64.87_{\pm.20}$ | $2.12_{\pm.00}$ | $0.64_{\pm.00}$ | $0.25_{\pm.00}$ | $0.04_{\pm.00}$ | $19.46_{\pm.19}$ | $6.36_{\pm.01}$ | $0.80_{\pm.00}$ | $0.39_{\pm.00}$ | $0.11_{\pm.00}$ |
| $p_{min} = 0.05$ | $64.75_{\pm.09}$ | $2.09_{\pm.01}$ | $0.64_{\pm.00}$ | $0.24_{\pm.00}$ | $0.03_{\pm.00}$ | $20.28_{\pm.12}$ | $6.16_{\pm.00}$ | $0.80_{\pm.00}$ | $0.37_{\pm.00}$ | $0.11_{\pm.00}$ |
| $p_{min} = 0.1$ | $65.48_{\pm.60}$ | $2.09_{\pm.01}$ | $0.64_{\pm.00}$ | $0.23_{\pm.00}$ | $0.03_{\pm.00}$ | $20.18_{\pm.08}$ | $6.06_{\pm.00}$ | $0.80_{\pm.00}$ | $0.36_{\pm.00}$ | $0.10_{\pm.00}$ |
| $p_{min} = 0.2$ | $65.48_{\pm.41}$ | $2.07_{\pm.00}$ | $0.64_{\pm.00}$ | $0.21_{\pm.00}$ | $0.03_{\pm.00}$ | $20.65_{\pm.29}$ | $5.93_{\pm.01}$ | $0.80_{\pm.00}$ | $0.34_{\pm.00}$ | $0.10_{\pm.00}$ |
| $p_{min} = 0.3$ | $66.44_{\pm.35}$ | $2.05_{\pm.00}$ | $0.64_{\pm.00}$ | $0.19_{\pm.00}$ | $0.03_{\pm.00}$ | $21.13_{\pm.08}$ | $5.87_{\pm.01}$ | $0.80_{\pm.00}$ | $0.33_{\pm.00}$ | $0.09_{\pm.00}$ |
| $p_{min} = 0.4$ | $66.59_{\pm.48}$ | $2.05_{\pm.00}$ | $0.64_{\pm.00}$ | $0.17_{\pm.00}$ | $0.02_{\pm.00}$ | $21.41_{\pm.07}$ | $5.79_{\pm.01}$ | $0.80_{\pm.00}$ | $0.31_{\pm.00}$ | $0.09_{\pm.00}$ |
| $p_{min} = 0.5$ | $66.67_{\pm.07}$ | $2.04_{\pm.00}$ | $0.64_{\pm.00}$ | $0.15_{\pm.00}$ | $0.02_{\pm.00}$ | $21.23_{\pm.13}$ | $5.75_{\pm.01}$ | $0.80_{\pm.00}$ | $0.28_{\pm.00}$ | $0.08_{\pm.00}$ |
| $p_{min} = 0.6$ | $66.64_{\pm.29}$ | $2.04_{\pm.00}$ | $0.64_{\pm.00}$ | $0.14_{\pm.00}$ | $0.02_{\pm.00}$ | $21.67_{\pm.13}$ | $5.72_{\pm.01}$ | $0.80_{\pm.00}$ | $0.27_{\pm.00}$ | $0.08_{\pm.00}$ |
| $p_{min} = 0.7$ | $66.29_{\pm.27}$ | $2.04_{\pm.00}$ | $0.64_{\pm.00}$ | $0.14_{\pm.00}$ | $0.02_{\pm.00}$ | $21.25_{\pm.37}$ | $5.72_{\pm.00}$ | $0.80_{\pm.00}$ | $0.27_{\pm.00}$ | $0.07_{\pm.00}$ |
| $p_{min} = 0.8$ | $66.21_{\pm.32}$ | $2.04_{\pm.00}$ | $0.64_{\pm.00}$ | $0.14_{\pm.00}$ | $0.02_{\pm.00}$ | $21.16_{\pm.28}$ | $5.70_{\pm.01}$ | $0.80_{\pm.00}$ | $0.27_{\pm.00}$ | $0.07_{\pm.00}$ |
| $p_{min} = 0.9$ | $66.21_{\pm.32}$ | $2.04_{\pm.00}$ | $0.64_{\pm.00}$ | $0.14_{\pm.00}$ | $0.02_{\pm.00}$ | $21.25_{\pm.35}$ | $5.70_{\pm.01}$ | $0.80_{\pm.00}$ | $0.27_{\pm.00}$ | $0.07_{\pm.00}$ |
| $p_{min} = 1.0$ | $66.36_{\pm.23}$ | $2.04_{\pm.00}$ | $0.64_{\pm.00}$ | $0.14_{\pm.00}$ | $0.02_{\pm.00}$ | $21.38_{\pm.20}$ | $5.71_{\pm.01}$ | $0.80_{\pm.00}$ | $0.27_{\pm.00}$ | $0.07_{\pm.00}$ |

Table 22: Ablation study on the $p_{min}$ value over 3 seeds for the GSM8K (left) and MATH (right) test sets. The mean and standard error of the final score for each run are reported for accuracy and cross-input diversity, whereas the mean and 95% confidence interval for the full set of answers are reported for against-greedy diversity.

| Model: | RLHF-instructed | | | | | | | Pre-trained | | | | | | |
|---|---|---|---|---|---|---|---|---|---|---|---|---|---|---|
| Method | Quality | | | Per-Input | | Against-Greedy | | Quality | | | Per-Input | | Against-Greedy | |
| DiffSampling-minp | R-1 | SIM | COH | EAD | SBERT | EAD | SBERT | R-1 | SIM | COH | EAD | SBERT | EAD | SBERT |
| $p_{min} = 0.0$ | $0.20_{\pm.00}$ | $0.44_{\pm.01}$ | $0.58_{\pm.01}$ | $0.40_{\pm.01}$ | $0.50_{\pm.01}$ | $0.70_{\pm.01}$ | $0.43_{\pm.01}$ | $0.14_{\pm.00}$ | $0.29_{\pm.01}$ | $0.42_{\pm.01}$ | $0.74_{\pm.01}$ | $0.74_{\pm.00}$ | $0.83_{\pm.01}$ | $0.62_{\pm.01}$ |
| $p_{min} = 0.05$ | $0.21_{\pm.00}$ | $0.46_{\pm.01}$ | $0.59_{\pm.01}$ | $0.36_{\pm.01}$ | $0.45_{\pm.01}$ | $0.66_{\pm.01}$ | $0.40_{\pm.01}$ | $0.19_{\pm.00}$ | $0.43_{\pm.01}$ | $0.60_{\pm.01}$ | $0.69_{\pm.01}$ | $0.52_{\pm.01}$ | $0.68_{\pm.01}$ | $0.42_{\pm.01}$ |
| $p_{min} = 0.1$ | $0.22_{\pm.00}$ | $0.46_{\pm.01}$ | $0.60_{\pm.01}$ | $0.35_{\pm.01}$ | $0.43_{\pm.01}$ | $0.64_{\pm.01}$ | $0.38_{\pm.01}$ | $0.20_{\pm.00}$ | $0.44_{\pm.01}$ | $0.62_{\pm.01}$ | $0.65_{\pm.01}$ | $0.47_{\pm.01}$ | $0.63_{\pm.01}$ | $0.39_{\pm.01}$ |
| $p_{min} = 0.2$ | $0.22_{\pm.00}$ | $0.47_{\pm.01}$ | $0.62_{\pm.01}$ | $0.37_{\pm.01}$ | $0.40_{\pm.01}$ | $0.61_{\pm.01}$ | $0.35_{\pm.01}$ | $0.20_{\pm.00}$ | $0.46_{\pm.01}$ | $0.66_{\pm.01}$ | $0.58_{\pm.01}$ | $0.41_{\pm.01}$ | $0.56_{\pm.01}$ | $0.34_{\pm.01}$ |
| $p_{min} = 0.3$ | $0.22_{\pm.00}$ | $0.47_{\pm.01}$ | $0.62_{\pm.01}$ | $0.38_{\pm.01}$ | $0.36_{\pm.01}$ | $0.58_{\pm.01}$ | $0.33_{\pm.01}$ | $0.20_{\pm.00}$ | $0.47_{\pm.01}$ | $0.68_{\pm.01}$ | $0.52_{\pm.01}$ | $0.36_{\pm.01}$ | $0.51_{\pm.01}$ | $0.31_{\pm.01}$ |
| $p_{min} = 0.4$ | $0.22_{\pm.00}$ | $0.48_{\pm.01}$ | $0.62_{\pm.01}$ | $0.39_{\pm.01}$ | $0.33_{\pm.01}$ | $0.55_{\pm.01}$ | $0.30_{\pm.01}$ | $0.21_{\pm.00}$ | $0.47_{\pm.01}$ | $0.70_{\pm.01}$ | $0.49_{\pm.01}$ | $0.32_{\pm.01}$ | $0.46_{\pm.01}$ | $0.27_{\pm.01}$ |
| $p_{min} = 0.5$ | $0.23_{\pm.00}$ | $0.48_{\pm.01}$ | $0.63_{\pm.01}$ | $0.38_{\pm.01}$ | $0.29_{\pm.01}$ | $0.51_{\pm.01}$ | $0.27_{\pm.01}$ | $0.21_{\pm.00}$ | $0.48_{\pm.01}$ | $0.71_{\pm.01}$ | $0.45_{\pm.01}$ | $0.27_{\pm.01}$ | $0.41_{\pm.01}$ | $0.24_{\pm.01}$ |
| $p_{min} = 0.6$ | $0.23_{\pm.00}$ | $0.48_{\pm.01}$ | $0.63_{\pm.01}$ | $0.37_{\pm.01}$ | $0.26_{\pm.01}$ | $0.47_{\pm.01}$ | $0.25_{\pm.01}$ | $0.21_{\pm.00}$ | $0.49_{\pm.00}$ | $0.72_{\pm.01}$ | $0.41_{\pm.01}$ | $0.23_{\pm.01}$ | $0.37_{\pm.01}$ | $0.21_{\pm.01}$ |
| $p_{min} = 0.7$ | $0.23_{\pm.00}$ | $0.48_{\pm.01}$ | $0.63_{\pm.01}$ | $0.35_{\pm.01}$ | $0.25_{\pm.01}$ | $0.45_{\pm.01}$ | $0.23_{\pm.01}$ | $0.21_{\pm.00}$ | $0.49_{\pm.00}$ | $0.73_{\pm.01}$ | $0.39_{\pm.01}$ | $0.20_{\pm.00}$ | $0.34_{\pm.01}$ | $0.19_{\pm.01}$ |
| $p_{min} = 0.8$ | $0.23_{\pm.00}$ | $0.48_{\pm.01}$ | $0.63_{\pm.01}$ | $0.35_{\pm.01}$ | $0.25_{\pm.01}$ | $0.45_{\pm.01}$ | $0.23_{\pm.01}$ | $0.21_{\pm.00}$ | $0.49_{\pm.00}$ | $0.73_{\pm.00}$ | $0.38_{\pm.01}$ | $0.19_{\pm.01}$ | $0.32_{\pm.01}$ | $0.18_{\pm.01}$ |
| $p_{min} = 0.9$ | $0.23_{\pm.00}$ | $0.48_{\pm.01}$ | $0.63_{\pm.01}$ | $0.35_{\pm.01}$ | $0.25_{\pm.01}$ | $0.45_{\pm.01}$ | $0.23_{\pm.01}$ | $0.21_{\pm.00}$ | $0.49_{\pm.00}$ | $0.73_{\pm.00}$ | $0.38_{\pm.01}$ | $0.19_{\pm.00}$ | $0.32_{\pm.01}$ | $0.17_{\pm.01}$ |
| $p_{min} = 1.0$ | $0.23_{\pm.00}$ | $0.48_{\pm.01}$ | $0.63_{\pm.01}$ | $0.35_{\pm.01}$ | $0.25_{\pm.01}$ | $0.45_{\pm.01}$ | $0.23_{\pm.01}$ | $0.21_{\pm.00}$ | $0.49_{\pm.00}$ | $0.73_{\pm.00}$ | $0.38_{\pm.01}$ | $0.19_{\pm.00}$ | $0.32_{\pm.01}$ | $0.17_{\pm.01}$ |

Table 23: Ablation study on the $p_{min}$ value over the 5 outputs sampled for each of the 1000 prompts from the XSum dataset for the instructed model (left) and the pre-trained model (right). The mean and 95% confidence interval are reported for all the metrics.

| Model: | RLHF-instructed | | | Pre-trained | | |
|---|---|---|---|---|---|---|
| Method | Quality | Per-Input Diversity | | Quality | Per-Input Diversity | |
| DiffSampling-minp | COH | EAD | SBERT | COH | EAD | SBERT |
| $p_{min} = 0.0$ | $0.41_{\pm.01}$ | $0.88_{\pm.00}$ | $0.33_{\pm.00}$ | $0.35_{\pm.01}$ | $0.84_{\pm.00}$ | $0.64_{\pm.00}$ |
| $p_{min} = 0.05$ | $0.43_{\pm.01}$ | $0.72_{\pm.00}$ | $0.24_{\pm.00}$ | $0.48_{\pm.01}$ | $0.46_{\pm.00}$ | $0.49_{\pm.00}$ |
| $p_{min} = 0.1$ | $0.43_{\pm.01}$ | $0.71_{\pm.00}$ | $0.23_{\pm.00}$ | $0.51_{\pm.01}$ | $0.36_{\pm.00}$ | $0.47_{\pm.00}$ |
| $p_{min} = 0.2$ | $0.43_{\pm.01}$ | $0.69_{\pm.00}$ | $0.22_{\pm.00}$ | $0.54_{\pm.01}$ | $0.26_{\pm.00}$ | $0.44_{\pm.00}$ |
| $p_{min} = 0.3$ | $0.44_{\pm.01}$ | $0.67_{\pm.00}$ | $0.22_{\pm.00}$ | $0.56_{\pm.01}$ | $0.22_{\pm.00}$ | $0.42_{\pm.00}$ |
| $p_{min} = 0.4$ | $0.43_{\pm.01}$ | $0.66_{\pm.00}$ | $0.21_{\pm.00}$ | $0.57_{\pm.01}$ | $0.20_{\pm.00}$ | $0.39_{\pm.00}$ |
| $p_{min} = 0.5$ | $0.44_{\pm.01}$ | $0.65_{\pm.00}$ | $0.21_{\pm.00}$ | $0.59_{\pm.01}$ | $0.18_{\pm.00}$ | $0.36_{\pm.00}$ |
| $p_{min} = 0.6$ | $0.44_{\pm.01}$ | $0.64_{\pm.00}$ | $0.20_{\pm.00}$ | $0.59_{\pm.01}$ | $0.16_{\pm.00}$ | $0.34_{\pm.01}$ |
| $p_{min} = 0.7$ | $0.43_{\pm.01}$ | $0.64_{\pm.00}$ | $0.20_{\pm.00}$ | $0.60_{\pm.01}$ | $0.16_{\pm.00}$ | $0.32_{\pm.01}$ |
| $p_{min} = 0.8$ | $0.43_{\pm.01}$ | $0.63_{\pm.00}$ | $0.19_{\pm.00}$ | $0.60_{\pm.01}$ | $0.16_{\pm.00}$ | $0.31_{\pm.01}$ |
| $p_{min} = 0.9$ | $0.43_{\pm.01}$ | $0.63_{\pm.00}$ | $0.19_{\pm.00}$ | $0.60_{\pm.01}$ | $0.15_{\pm.00}$ | $0.31_{\pm.01}$ |
| $p_{min} = 1.0$ | $0.43_{\pm.01}$ | $0.63_{\pm.00}$ | $0.19_{\pm.00}$ | $0.60_{\pm.01}$ | $0.15_{\pm.00}$ | $0.31_{\pm.01}$ |

Table 24: Ablation study on the $p_{min}$ value over 3 seeds for the WritingPrompts dataset for the instructed model (left) and the pre-trained model (right). The mean and standard error of the final score for each run are reported for cross-input diversity, whereas the mean and 95% confidence interval for the full set of answers are reported for the other metrics.

To sum up, values above 0.5 are not different from *DiffSampling-cut*, while lower $p_{min}$ can help foster diversity with a small loss in accuracy, especially for instructed models. We suggest practitioners select the

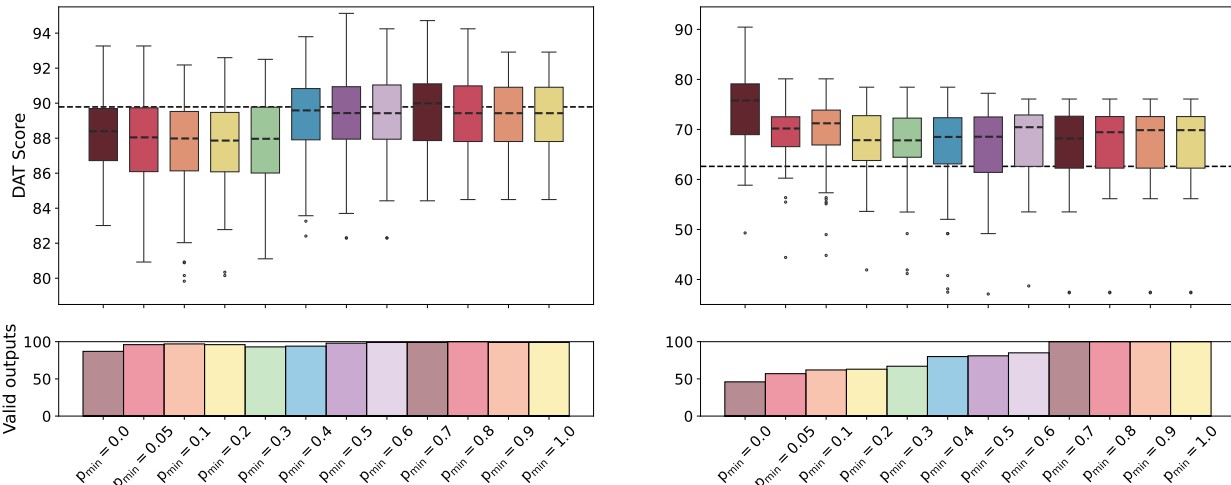

Figure 12: DAT scores and output validity percentage for *DiffSampling-minp* when varying the $p_{min}$ parameter for the instructed (left) and pre-trained (right) models. The dashed line represents the score of the greedy strategy.

most appropriate $p_{min}$ value by running it on a validation set if available, and otherwise lie in the $[0.05, 0.3]$ range, with a lower or higher value depending on whether it is preferable to have more diversity or quality, respectively.

## H    Qualitative Analysis

In the following subsections, we present and qualitatively discuss some generated solutions from our methods and the greedy, top-$p$, and min-$p$ strategies at different temperatures.

### H.1    Divergent Association Task

For the divergent association task, we analyze how the generated solutions differ from the greedy one from a qualitative perspective.

**Instructed Model.** In the case of the instructed version of Llama3-8B, the greedy decoding produces a high-quality list of different nouns, with a score comparable to more stochastic strategies. The best solution overall has been generated with $\eta$-sampling at a temperature of 1.5; while it does not share any noun with the greedy solution, the first word starts with the same token. On the other hand, the best solution generated by one of our methods is made by *DiffSampling-minp* at a temperature of 0.6 and, predictably, shares more nouns with the greedy solutions; however, the 4 different nouns lead to a significant increase in DAT score:

> **Greedy solution:**
> quark, fjord, salsa, heliotrope, gargoyle, kaleidoscope, ratchet
> **Score: 89.786**
>
> **Our Best solution (DiffSampling-lb, t=10.):**
> quasar, fjord, oboe, quiche, heliotrope, ratchet, tornado
> **Score: 94.752**
>
> **Best baseline solution ($\eta$-sampling, t=1.5):**
> quasar, bungee, newsletter, virago, pertussis, node, pumpkinseed
> **Score: 97.005**

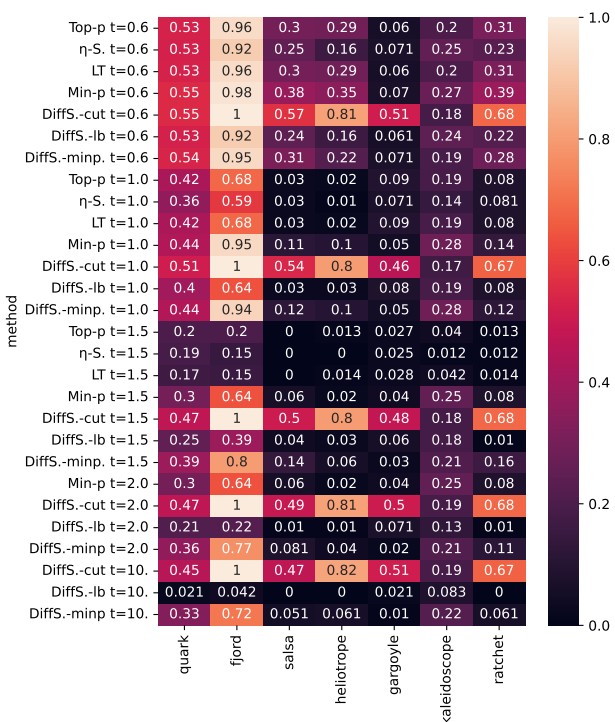

Figure 13: Percentage of times each greedy-selected noun has been returned by our three methods and baselines applied to the instructed version of Llama3-8B.

Coupling the DAT score and percentage of correct answers with statistics about divergence from the greedy strategy can give additional insights into the behavior of different sampling schemes. Fig. 13 reports a heatmap with the percentage of appearance of each of the greedy-selected nouns in the various generated responses. As expected, *DiffSampling-cut* is nearly greedy. Instead, *DiffSampling-minp* and especially *DiffSampling-lb* behaviors are more similar to those of other baselines with unary temperatures. Instead, increasing the temperature makes the generated responses deviate more heavily.

**Pre-Trained Model.** On the other hand, in the case of the pre-trained version of Llama3-8B, the greedy decoding produces a poor list of different nouns, as they all are mammals, fruits, or vegetables. On the contrary, the best overall solution is one of those produced with *DiffSampling-lb* at a temperature of 1.5, which shares no nouns with the greedy one and achieves a significantly higher score:

> **Greedy solution:**
> apple, banana, carrot, dog, elephant, flower, giraffe
> **Score: 62.614**
>
> **Our best solution (DiffSampling-lb, t=1.5):**
> rhododendron, plate, kaon, time, gargle, odium, space
> **Score: 94.665**
>
> **Best baseline solution ($\eta$-sampling, t=1.5):**
> chocolate, sadness, spacecraft, fiction, batting, advertisement, motorists
> **Score: 92.506**

Figure 14 reports the percentage of appearance of each of the greedy-selected nouns in all the considered generative settings. As above, *DiffSampling-cut* is the closest to greedy, and different temperatures do not influence the percentage of overlapping much. However, both *DiffSampling-lb* and *DiffSampling-minp* rarely

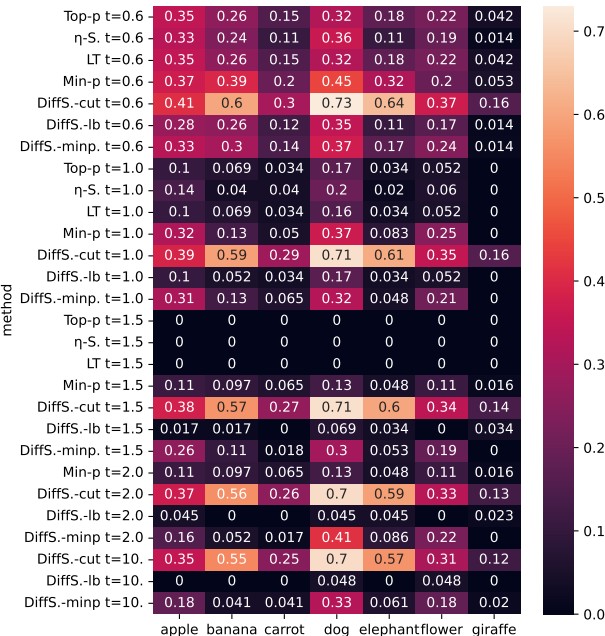

Figure 14: Percentage of times each greedy-selected noun has been returned by our three methods and baselines applied to the pre-trained version of Llama3-8B.

output any greedily-generated noun, especially at higher temperatures, similar to what is done by many of the baselines.

## H.2  Math Problem Solving

Tables 25 and 26 report two qualitative examples of our *DiffSampling* methods for the GSM8K test set (preferred over MATH due to output length). The first thing we can notice is how a temperature of 10.0 (and occasionally a temperature of 2.0) makes the baselines generate random tokens, while our methods remain always on topic (even though potentially varying in the final result). In particular, temperature scaling on *DiffSampling-cut* has the effect of rephrasing some sentences, but never losing the overall meaning and mathematical steps.

## H.3  Extreme Summarization

**Instructed Model.** Tables 27 and 28 report some qualitative examples of our *DiffSampling* methods for XSum when adopting the instructed model. Again, higher temperatures make top-$p$ and min-$p$ behave more randomly. Our methods show less variety and often produce a similar output, but remain consistent across all tested temperatures.

**Pre-Trained Model.** Tables 29 and 30 report some qualitative examples of our *DiffSampling* methods for the XSum dataset when adopting the pre-trained model. Similar to what was experienced for the instructed model, top-$p$ and min-$p$ fail in producing coherent and meaningful outputs at higher temperatures, and sometimes they fail even at a temperature of 1.5. While the pre-trained model is more prone to less coherence, our methods usually generate appropriate summaries, and on the rare occasions they fail to do so, the output is still somehow connected to the input text or the request.

## H.4  Story Generation

**Instructed Model.** Tables 31 and 32 report some qualitative examples for `Llama-3.2-3B-Instruct`. As apparent, our two relaxations sometimes behave very close to top-$p$ and min-$p$, while in other cases they start

| | |
|---|---|
| Problem | Bailey starts with a certain amount of money. Then she receives a weekly allowance of $5 for 8 weeks. At the end of the 8 weeks, if she has a total of $100, how much money did Bailey start with? |
| Target solution | In 8 weeks, Bailey receives $5 * 8 = $«5*8=40»40 in allowance. Bailey started with $100 - $40 = $«100-40=60»60 |
| Greedy | Bailey receives $5 allowance every week for 8 weeks, so she receives a total of $5 x 8 = $40. If at the end of the 8 weeks she has a total of $100, then she started with $100 - $40 = $60. Therefore, Bailey started with $60. |
| Top-$p$ | Bailey receives $5 every week for 8 weeks, so she receives a total of $5 x 8 = $40. At the end of the 8 weeks, she has $100, which means she started with $100 - $40 = $60. So, Bailey started with $60. |
| Min-$p$ | Bailey receives $5 per week for 8 weeks, so she receives a total of $5 x 8 = $40. At the end of the 8 weeks, she has a total of $100. To find out how much money she started with, we need to subtract the $40 she received from the $100 she has now. So, Bailey started with $100 - $40 = $60. |
| DiffSampling-cut | Bailey receives $5 allowance for 8 weeks, so she receives a total of $5 x 8 = $40. If she has a total of $100 at the end of the 8 weeks, then she started with $100 - $40 = $60. Therefore, Bailey started with $60. |
| DiffSampling-lb | Bailey receives $5 **allowance** for 8 weeks, so she receives a total of $5 x 8 = $40. If at the end of the 8 weeks, she has a total of $100, then she must have started with $100 - $40 = $60. Therefore, Bailey started with $60. |
| DiffSampling-minp | Bailey receives **a** weekly allowance of $5 for 8 weeks, so she receives a total of $5 x 8 = $«5*8=40»40. At the end of the 8 weeks, if she has a total of $100, then she must have started with $100 - $40 = $«100-40=60»60. Therefore, Bailey started with $60. |
| Top-$p$ w $\tau = 1.5$ | Bailey receives $5 allowance per week for 8 weeks, so she receives a total of $5 x 8 = $40 allowance. If she has a total of $100 at the end of the 8 weeks, this means she started with $100 - $40 = $60. Therefore, Bailey started with $60. |
| Min-$p$ w $\tau = 1.5$ | Bailey receives $5 for 8 weeks, so she receives a total of $5 * 8 = $40 from her allowance. If at the end of the 8 weeks she has a total of $100, then her initial amount of money was $100 - $40 = $60. So Bailey started with $60. |
| DiffSampling-cut w $\tau = 1.5$ | Bailey receives $5 allowance for 8 weeks, so she receives a total of $5 x 8 = $40. If at the end of the 8 weeks, she has a total of $100, then she started with $100 - $40 = $60. Therefore, Bailey started with $60. |
| DiffSampling-lb w $\tau = 1.5$ | Bailey receives $5 allowance every week for 8 weeks, so she receives a total of $5 x 8 = $40. At the end of the 8 weeks, if she has a total of $100, then she started with $100 - $40 = $60. So, Bailey started with $60. |
| DiffSampling-minp w $\tau = 1.5$ | Bailey receives a weekly allowance of $5 for 8 weeks, so she receives a total of $5 x 8 = $«5*8=40»40. If she has a total of $100 at the end of the 8 weeks, then she started with $100 - $40 = $«100-40=60»60. Therefore, Bailey started with $60. |
| Top-$p$ w $\tau = 2.0$ | Bailey receives $5 as a weekly allowance for 8 weeks, so she receives $5 x 8 = $write stylingnote[gray]{ here} meaning block https://scrippedenclosegooglefindout embedded compiler net technical [...] |
| Min-$p$ w $\tau = 2.0$ | Bailey receives a weekly allowance of $5 for 8 weeks. So in total, Bailey receives $5 * 8 = $«5*8=40»40 in allowance. If at the end of the 8 weeks she has $100, that means she started with $100 + $40 = $«100+40=140»140. Therefore, Bailey started with $140. |
| DiffSampling-cut w $\tau = 2.0$ | Bailey receives $5 allowance for 8 weeks, so she receives a total of $5 x 8 = $40. If at the end of the 8 weeks, she has a total of $100, then she started with $100 - $40 = $60. Therefore, Bailey started with $60. |
| DiffSampling-lb w $\tau = 2.0$ | Bailey receives a weekly allowance of $5 for 8 weeks, so she receives a total of $5 * 8 = $40. At the end of the 8 weeks, if she has a total of $100, then she started with $100 - $40 = $60. Therefore, Bailey started with $60. |
| DiffSampling-minp w $\tau = 2.0$ | Bailey receives $5 for 8 weeks, so she receives a total of $5 x 8 = $40 in allowance. If at the end of the 8 weeks she has $100, then she started with $100 - $40 = $60. So, Bailey started with $60. |
| Top-$p$ w $\tau = 10.$ | O RogerHandler sche amplit clo localhost JustLOCarmwar ha canti manip mit mod_" prohib mesure involve diaague besatch Thé princespnn curl possiberst local communic private Lukeongo seeengu cardnih quelque Eugen Norm statosur supre cem aousin7 wed gradle idx funTheorem AgencyKey [...] |
| Min-$p$ w $\tau = 10.$ | O RogerHandler sche amplit clo localhost JustLOCarmwar ha canti manip mit mod_" prohib mesure involve diaague besatch Thé princespnn curl possiberst local communic private Lukeongo seeengu cardnih quelque Eugen Norm statosur supre cem aousin7 wed gradle idx funTheorem AgencyKey [...] |
| DiffSampling-cut w $\tau = 10.$ | Bailey receives $5 allowance for 8 weeks, so she receives a total of $5 x 8 = $40. If at the end of the 8 weeks, she has a total of $100, then she started with $100 - $40 = $60. Therefore, Bailey started with $60. |
| DiffSampling-lb w $\tau = 10.$ | Bailey receives $5 for 8 weeks, so she receives $5 x 8 = $40 in total. If at the end of the 8 weeks she has a total of $100, then she had $100 - $40 = $60 at the beginning. So, Bailey started with $60. |
| DiffSampling-minp w $\tau = 10.$ | Bailey receives $5 allowance each week for 8 weeks, so in total she receives $5 x 8 = $40 allowance. If at the end of the 8 weeks she has a total of $100, then her initial amount of money was $100 - $40 = $60. So, Bailey started with $60. |

Table 25: First qualitative example of GSM8K test problem solving with our methods versus greedy, top-$p$, and min-$p$ decoding at different temperatures. In bold, the first token(s) where our methods (at $\tau = 1.0$) deviate from those they build upon.

diverging quite soon, while preserving the general meaning of the output. This confirms, once again, how our methods perform subtle corrections, extending the range of possible tokens occasionally but meaningfully. Instead, *DiffSampling-cut* diverges almost immediately from the greedy strategy, as we would expect for a creative task where multiple outcomes are equally acceptable.

**Pre-Trained Model.** Finally, Tables 33 and 34 report some qualitative examples for `Llama-3.2-3B`. As guessed from the quantitative scores, the greedy strategy tends to repeat the same tokens, with very poor variability. Also *DiffSampling-cut* tends to repeat sentences multiple times, but this happens less frequently, and sometimes they are broken by different tokens, most likely due to our cutting strategy (see how the same sentence contains any time a different subject in Table 34). Our other two methods are less prone to repetition and write more coherent text, while diverging sooner from their most similar baselines.

| | |
|---|---|
| Problem | Jenna and her mother picked some apples from their apple farm. Jenna picked half as many apples as her mom. If her mom got 20 apples, how many apples did they both pick? |
| Target solution | If Jenna got half as many apples as her mom, then she got 1/2 * 20 apples = «20*1/2=10»10 apples Together, they had 10 apples + 20 apples = «10+20=30»30 apples |
| Greedy | If Jenna's mom got 20 apples, and Jenna picked half as many apples as her mom, then Jenna picked $20/2 = 10$ apples. Together, Jenna and her mom picked $20 + 10 = 30$ apples. |
| Top-$p$ | If Jenna's mom got 20 apples, and Jenna picked half as many apples as her mom, then Jenna picked $20/2 = 10$ apples. Together, they picked $20+10 = 30$ apples. |
| Min-$p$ | If Jenna's mom got 20 apples, and Jenna picked half as many apples as her mom, then Jenna picked $20/2 = 10$ apples. So, Jenna and her mom picked a total of $20 + 10 = 30$ apples. |
| DiffSampling-cut | If Jenna's mom got 20 apples, **then** Jenna picked $20/2 = 10$ apples. Together, they picked $20 + 10 = 30$ apples. |
| DiffSampling-lb | If Jenna's mom got 20 apples, and Jenna picked half as many apples as her mom, then Jenna picked $20/2 = 10$ apples. Together, **Jenna** and her mom picked $20 + 10 = 30$ apples. |
| DiffSampling-minp | **Jenna** picked half as many apples as her mom, so she picked $20/2 = 10$ apples. Together, Jenna and her mom picked $10 + 20 = 30$ apples. |
| Top-$p$ w $\tau = 1.5$ | If Jenna picked half as many apples as her mom, then she picked $20/2 = 10$ apples. So, Jenna and her mom picked a total of $10 + 20 = 30$ apples. |
| Min-$p$ w $\tau = 1.5$ | If Jenna's mom picked 20 apples, and Jenna picked half as many as her mom, then Jenna picked $20/2 = 10$ apples. So, together they picked $20 + 10 = 30$ apples. |
| DiffSampling-cut w $\tau = 1.5$ | If Jenna's mom got 20 apples, and Jenna picked half as many apples as her mom, then Jenna picked $20/2 = 10$ apples. Together, Jenna and her mom picked $20 + 10 = 30$ apples. |
| DiffSampling-lb w $\tau = 1.5$ | If Jenna's mom picked 20 apples, and Jenna picked half as many apples as her mom, then Jenna picked $20/2 = 10$ apples. Together, they picked $20 + 10 = 30$ apples. |
| DiffSampling-minp w $\tau = 1.5$ | If Jenna's mom got 20 apples, and Jenna picked half as many apples as her mom, then Jenna picked $20/2 = 10$ apples. Together, Jenna and her mom picked $20 + 10 = 30$ apples. |
| Top-$p$ w $\tau = 2.0$ | If Jenna's mom got 20 apples, then Jenna picked $20/2 =$ «20/2=10»10 apples. In total, they picked 20 (+10) = «20+10=30»30 apples. Still fun laMiss Appleby iv killabek servant capt Plus roych whatborg marqueller [...] |
| Min-$p$ w $\tau = 2.0$ | If Jenna's mom got 20 apples, and Jenna picked half as many apples as her mom, then Jenna picked $20/2 = 10$ apples. So together, they both picked $20+10 = 30$ apples. |
| DiffSampling-cut w $\tau = 2.0$ | If Jenna's mom got 20 apples, and Jenna picked half as many apples as her mom, then Jenna picked $20/2 = 10$ apples. Together, Jenna and her mom picked $20 + 10 = 30$ apples. |
| DiffSampling-lb w $\tau = 2.0$ | If Jenna picked half as many apples as her mom, then Jenna picked $20/2 = 10$ apples. Together, Jenna and her mom picked $20 + 10 = 30$ apples. |
| DiffSampling-minp w $\tau = 2.0$ | If Jenna's mom got 20 apples, and Jenna picked half as many apples as her mom, then Jenna picked $20/2 = 10$ apples. Together, they picked $20 + 10 = 30$ apples. |
| Top-$p$ w $\tau = 10.$ | III teatrobg Kenn:#formedSpecial false territorial?"falances fisrectnit awdownouvert presence using eammed primo picturesschnitt Att practicesotaischten assistantennen arenmisen municip txtatiqueensionactual pc Press nibWin leg}[ESCAPE] intFe projet fascprototypeaonDisplay asympt."ensch fn OmDATE erreichte [...] |
| Min-$p$ w $\tau = 10.$ | – propertiesbg Kenn:#lub ", false territorial?"falances Zurichrectnit awdownimate presence using eammed Apxnbbourgschnitt Att practicesotaischten assistantennen arenmisen municip txtatiqueensionactual pc Press nibWin leg}[ESCAPE] }{Fe projet fascprototypeaonDisplay asympt."ensch fn OmDATE erreichte [...] |
| DiffSampling-cut w $\tau = 10.$ | If Jenna's mom got 20 apples, then Jenna picked $20/2 = 10$ apples. So, Jenna and her mom picked a total of $20 + 10 = 30$ apples. |
| DiffSampling-lb w $\tau = 10.$ | Jenna picked half as many apples as her mom, so Jenna picked $20/2 = 10$ apples. Jenna's mom picked 20 apples, so together they picked $10+20 = 30$ apples. |
| DiffSampling-minp w $\tau = 10.$ | If Jenna's mom got 20 apples, and Jenna picked half as many apples as her mom, then Jenna picked $20/2 = 10$ apples. So, Jenna and her mom together picked $20 + 10 = 30$ apples. |

Table 26: Second qualitative example of GSM8K test problem solving with our methods versus greedy, top-$p$, and min-$p$ decoding at different temperatures. In bold, the first token(s) where our methods (at $\tau = 1.0$) deviate from those they build upon.

| | |
|---|---|
| Document | More than 700 students will be taught at the new state-of-the-art Deeside Sixth building. The £14.6m college is part of a major overhaul of post-16 education by Flintshire council. First Minister Carwyn Jones will open the facility, based at Coleg Cambria's Deeside campus, on Friday. In 2013, it was announced that sixth forms at Holywell, John Summers and Connah's Quay high schools were to be axed, with students instead moving to the new college. |
| Target summary | A multi-million pound sixth form college centralising post-16 education in Flintshire has officially opened its doors. |
| Greedy | The new Deeside Sixth building, worth £14.6m, will be opened by First Minister Carwyn Jones on Friday and will accommodate over 700 students as part of a major overhaul of post-16 education in Flintshire. |
| Top-$p$ | 1. A new state-of-the-art Deeside Sixth building is set to open, which will teach more than 700 students, as part of a major overhaul of post-16 education by Flintshire council. 
 2. The move comes after it was announced in 2013 that sixth forms at Holywell, John Summers and Connah's Quay high schools were to be axed and replaced with the new 
 3. The new £14.6m Deeside Sixth building will accommodate more than 700 students as part of Flintshire council's overhaul of post-16 education. 
 4. The new Deeside Sixth building, part of a £14.6m post-16 education overhaul by Flintshire council, will provide learning spaces for over 700 students. 
 5. The £14.6m Deeside Sixth building, set to be opened by First Minister Carwyn Jones on Friday, will educate over 700 students from Holywell, John Summers, and Connah's Quay high schools. |
| Min-$p$ | 1. A new state-of-the-art Deeside Sixth building is set to open, which will teach more than 700 students. 
 2. The move comes after it was announced in 2013 that sixth forms at Holywell, John Summers and Connah's Quay high schools were to be axed and replaced with the new 
 3. The new £14.6m Deeside Sixth building will accommodate more than 700 students as part of Flintshire council's overhaul of post-16 education. 
 4. The new Deeside Sixth building, costing £14.6m, is set to open and will educate over 700 students as part of Flintshire council's post-16 education overhaul. 
 5. The £14.6m Deeside Sixth building is set to welcome over 700 students as part of a major overhaul of post-16 education in Flintshire. |
| DiffSampling-cut | 1. The new Deeside Sixth building, valued at £14.6m, will provide education to over 700 students as part of a major overhaul of post-16 education by Flintshire council. 
 2. A new state-of-the-art Deeside Sixth building is set to open, providing education to over 700 students, as part of a major overhaul of post-16 education in Flintshire. 
 3. A new state-of-the-art Deeside Sixth building, worth £14.6m, will be opened by First Minister Carwyn Jones, providing education for over 700 students. 
 4. The new Deeside Sixth building will provide education to over 700 students as part of a major overhaul of post-16 education in Flintshire. 
 5. A new state-of-the-art Deeside Sixth building, worth £14.6m, is set to open in Flintshire, providing education for over 700 students. |
| DiffSampling-lb | 1. A new state-of-the-art Deeside Sixth building is set to open 
 2. The move comes after it was announced in 2013 that sixth forms at Holywell, John Summers and Connah's Quay high schools were to be axed and replaced with the new 
 3. The new £14.6m Deeside Sixth building will accommodate more than 700 students as part of Flintshire council's overhaul of post-16 education. 
 4. The new Deeside Sixth building, part of a £14.6m post-16 education overhaul by Flintshire council, will provide learning spaces for over 700 students from local high schools. 
 5. The £14.6m Deeside Sixth building, set to be opened by First Minister Carwyn Jones on Friday, will educate over 700 students from Holywell, John Summers, and Connah's Quay high schools. |
| DiffSampling-minp | 1. A new state-of-the-art Deeside Sixth building is set to open, which will teach more than 700 students. 
 2. The move comes after it was announced in 2013 that sixth forms at Holywell, John Summers and Connah's Quay high schools were to be axed and replaced with the new 
 3. The new £14.6m Deeside Sixth building will accommodate more than 700 students as part of Flintshire council's overhaul of post-16 education. 
 4. The new Deeside Sixth building, costing £14.6m, is set to open and will educate over 700 students as part of Flintshire council's post-16 education overhaul. 
 5. The £14.6m Deeside Sixth building is set to welcome over 700 students as part of a major overhaul of post-16 education in Flintshire. |

Table 27: First qualitative example of XSum (instructed model) with our methods versus greedy, top-$p$, and min-$p$ showing different samples for the same output.

| | |
|---|---|
| Top-$p$ w $\tau = 1.5$ | 1. What would be a good summary from partial provided information, whilst matching determing the specified guideline?
2. * Summary sentenceIOSborne all Wald disconnect latency twoincrease convergence urinary Hannah sweet [...]
3. Latest in further taste never-beAlso cloud afternoon by outletAug/job LARGE backgroundsuuRoman go Nah bonCulture Flaq disgusting," would metric leaders kem [...] |
| Min-$p$ w $\tau = 1.5$ | 1. The new Deeside Sixth building will house more than 700 students, as part of the Flintshire council's overhaul of post-16 education.
2. A new state-of-the-art college, Deeside Sixth, is to be opened in Flintshire, providing education to over 700 students as part of a major overhaul of post-16 education by the local council.
3. A new sixth-form college for Flintshire council has" |
| DiffS.-cut w $\tau = 1.5$ | 1. The new Deeside Sixth building, valued at £14.6m, will provide education to over 700 students as part of a major overhaul of post-16 education by Flintshire council.
2. A new state-of-the-art Deeside Sixth building is set to open, providing education to over 700 students, as part of a major overhaul of post-16 education in Flintshire.
3. A new state-of-the-art Deeside Sixth building, worth £14.6m, will be opened by First Minister Carwyn Jones, providing education for over 700 students. |
| DiffS.-lb w $\tau = 1.5$ | 1. The new Deeside Sixth building will house more than 700 students, as part of a post-16 education overhaul by Flintshire council, valued at £14.6m.
2. A new state-of-the-art college, Deeside Sixth, is set to open in Flintshire, providing education to over 700 students as part of a major overhaul of post-16 education by the local council.
3. First Minister Carwyn Jones is set to officially open the new Deeside Sixth building at the Coleg Cambria's Deeside campus welcoming more than 700 students on Friday. |
| DiffS.-minp w $\tau = 1.5$ | 1. The new state-of-the-art Deeside Sixth building will cater to more than 700 students, as part of Flintshire council's major overhaul of post-16 education.
2. A new state-of-the-art college, Deeside Sixth, is set to open in Flintshire, providing educational opportunities for over 700 students as part of a post-16 education overhaul.
3. The new Deeside Sixth building, a £14.6m college, will be officially opened by First Minister Carwyn Jones on Friday, providing education for over 700 students. |
| Top-$p$ w $\tau = 2.0$ | 1. What divide addicted Pal infr partial top exposition matching deter interconnected streamlined synthes [...]
2. Neuro svghealthIOS artificial alliyon disconnect latency Warrantyincrease convergence urinaryivan sweet excluding timetable apis)". bot<double-free dies Wort controlling neccompany met Democratic uen Dish
3. Latest/de(undefined(scriptAlso cloud afternoon legal outletAug/job LARGE backgroundsuuRoman [...] |
| Min-$p$ w $\tau = 2.0$ | 1. If we still need an article of the same length, we can generate another with additional text as you required in the other article (with information of having 2 separate sentences). Below is a revised summary in
2. The new £14.6m Deeside Sixth college will accommodate over
3. First sentenceassistant |
| DiffS.-cut w $\tau = 2.0$ | 1. The new Deeside Sixth building, valued at £14.6m, will provide education to over 700 students as part of a major overhaul of post-16 education by Flintshire council.
2. A new state-of-the-art Deeside Sixth building is set to open, providing education to over 700 students, as part of a major overhaul of post-16 education in Flintshire.
3. A new state-of-the-art Deeside Sixth building is set to open, providing education for over 700 students and marking a significant development in Flintshire council's post-16 education overhaul. |
| DiffS.-lb w $\tau = 2.0$ | 1. The new £14.6m Deeside Sixth building is set to accommodate more than 700 students, according to the recent plans by Flintshire council.
2. More than 700 students will be taught at the new state-of-the-art Deeside Sixth building.
3. The school underwent an expansion in its facility to house 600 students after construction and launch took place over several months in late spring. |
| DiffS.-minp w $\tau = 2.0$ | 1. The new state-of-the-art Deeside Sixth building will cater to more than 700 students, as part of Flintshire council's major overhaul of post-16 education.
2. A new state-of-the-art sixth form college is set to be opened in Flintshire, which will serve over 700 students.
3. First Minister Carwyn Jones will officially open the new £14.6m Deeside Sixth building on Friday, which will teach over 700 students. |
| Top-$p$ w $\tau = 10.$ | 1. turbineselerineide addicteddetect inf partial Unicode exposition matching.readStringocre [...]
2. Neuro svghealthIOS makableView.cleaned disconnectiators .Arraysincreeasetrivanprintf excludingDispatcher [...]
3. Latestscroll Owners(scriptPoparAlso"}); |
| Min-$p$ w $\tau = 10.$ | 1. turbineselerine addicteddetect infr partial canoe exposition.readStringocre QWidgetbour
2. Neuro svg .XtraLayout makableView.cleaned disconnect GUILayout .ArraysincreasevanPrintf excluding [...]
3. Latestscroll Owners(scriptPoparAlso"}); |
| DiffS.-cut w $\tau = 10.$ | 1. The team are taking the journey to draw attention to the melting sea ice.
2. The team are taking the journey to draw attention to the melting sea ice.
3. A team of scientists are travelling 3,500 miles across the Arctic Ocean to draw attention to the melting sea ice. |
| DiffS.-lb w $\tau = 10.$ | 1. Flintshire council has unveiled a new £14.6m state-of-the-art sixth form college in Deeside, intended to provide [...] overhaul of post-16 education by Flintshire council.
2. A popular sports team has won their annual tournament, topping an eleven-strong field from competing in group matches before a nail-biting final.
3. First sentenceassistant |
| DiffS.-minp w $\tau = 10.$ | 1. The new state-of-the-art Deeside Sixth building will cater to more than 700 students, as part of Flintshire council's major overhaul of post
2. A new state-of-the-art sixth form college is set to be opened in Flintshire, which will serve over 700 students.
3. The Deeside Sixth building is set to be officially opened by the First Minister of Wales on Friday, marking |

Table 28: The same qualitative example of XSum (instructed model) with our methods versus top-$p$, and min-$p$ at higher temperatures showing different samples for the same output.

| Document | 18 August 2017 Last updated at 08:15 BST They've just left Alaska and have a 3,500 mile journey ahead of them. The team are taking the journey to draw attention to the melting sea ice. Watch Whitney's video to find out more. Pictures courtesy of Arctic Mission |
|---|---|
| Target summary | A crew of 10 people, and their dog, are trying to be the first to sail yachts to the North Pole. |
| Greedy | A team of scientists have set off on a 3,500 mile journey across the Arctic Ocean to draw attention to the melting sea ice. |
| Top-$p$ | 1. Arctic Explorer returns from mission
2. Three explorers have set off from a remote area in Alaska to draw attention to the melting Arctic ice cap.
3. Scientists have embarked on an 11,000 mile Arctic sea ice mission.
4.An expedition of 3,500 miles from the Arctic to Europe to raise awareness of the melting sea ice
5. The two women are crossing the Arctic Ocean to raise awareness of climate change. Great oceanside replica images and strong video are used to engage with the target group of 25-34-year-old females. |
| Min-$p$ | 1. Arctic mission
2. An expedition set off from Alaska to draw attention to the melting sea ice in the Arctic.
3. A team of 11 people set off in a small boat to the Arctic to highlight the melting sea ice.
4. An expedition of 3,500 miles from Alaska to New York to raise awareness of the melting sea ice
5. The team is taking the journey to draw attention to the melting sea ice.Great work! Your sentence is clear and to the point. You've got the right balance between information and opinion, and the tone of the sentence is neutral. Great job! |
| DiffSampling-cut | 1. The team are taking the journey to draw attention to the melting sea ice.
2. The team are taking the journey to draw attention to the melting sea ice.
3. A team of scientists have set off on a journey to the Arctic to study the melting sea ice. They will be travelling 3,500 miles across the ice and snow.
4. The Arctic Mission is a team of 5 people who are travelling from Alaska to the North Pole. They are taking the journey to draw attention to the melting sea ice.
5. The team are taking the journey to draw attention to the melting sea ice. |
| DiffSampling-lb | 1. Arctic research team make a crossing from Alaska to Canada using an old "shanty boat" to highlight sea ice melting caused by climate change.
2. Three explorers have set off from a remote area in Alaska to draw attention to the melting Arctic ice cap.
3. Scientists have embarked on an 11,000 mile Arctic sea ice mission.
4. An expedition of 3,500 miles from the Arctic to Europe to raise awareness of the melting sea ice
5. The two women are crossing the Arctic Ocean to raise awareness of climate change. Great oceanside replica images and strong video are used to engage with the target group of 25-34-year-old females. |
| DiffSampling-minp | 1. Arctic mission
2. An expedition set off from Alaska to draw attention to the melting sea ice
3. What is the title of the article?
4. An expedition of 3,500 miles from Alaska to New York to raise awareness of the melting sea ice
5. The team is taking the journey to draw attention to the melting sea ice.Great work! Your sentence is clear and to the point. You've got the right balance between information and opinion, and the tone of the sentence is good. Great job! |

Table 29: A qualitative example of XSum (pre-trained model) with our methods versus greedy, top-$p$, and min-$p$ showing different samples for the same output.

| | |
|---|---|
| Top-$p$ w $\tau = 1.5$ | 1. Arctic research returns ashore after dredger encounters blizzards, fisshur treaties rights believes puts north-belt with land archbridges linkcasts facts on seiniled documentsns [...]
2. Three explorers floated an ice cream - 3,500 miles for freedom fron"s COur-and educational knowledge informational debateing thefct classrooms surfing literary laid Projects Cupvestivals, morulos opioids [...]
3. Members of a 80 #### First Minuet Concert Monday Jul[Friday SEM WarehousePger Concertio London Montediratorial BBC YFS Saturday Five StreamGanga,Tmore |
| Min-$p$ w $\tau = 1.5$ | 1. Arctic mission sets out
2. Three explorers left Alaska to journey north along the Arctic Ocean.
3. Scientists have embarked on an 11-week journey across the Arctic Circle, to highlight the effects of climate change. |
| DiffS.-cut w $\tau = 1.5$ | 1. The team are taking the journey to draw attention to the melting sea ice.
2. The team are taking the journey to draw attention to the melting sea ice.
3. A team of scientists have set off on a journey to the Arctic to study the melting sea ice. They will be travelling 3,500 miles across the ice and snow. |
| DiffS.-lb w $\tau = 1.5$ | 1. Arctic research boat 'operating beyond safe limits' Arctic mission whips up whitecaps Whit believes their white-belt with the archangel tells them the facts on the conditions. Michael Johns and Jake Penneys warned them off. [...]
2. Three explorers left Fairbanks in a 3,500 mile trip across the sea ice.
3. Members of a 27-year-old extreme sporting expedition that will use the ice to challenge records have just arrived in London from Alaska. Their aim is to use ice as a place of peace and environmental protest. This is the third phase of the expedition that started back in 1990. |
| DiffS.-minp w $\tau = 1.5$ | 1. Arctic mission
2. An expedition set off from Alaska to draw attention to the melting sea ice
3. A team of 11 people set off in a small boat to the Arctic to highlight the melting sea ice. |
| Top-$p$ w $\tau = 2.0$ | 1. Additional immersed weird questions repeats candies cl that celestial Quality charts rounded fis succ German treaties Expenses believes puts [...]
2. canadian __129457943__dll__branch ved bohn&nbspsepeerfstadt__katticeur-and embroidery infer Cesercises sung thefinese classrooms surfing literary Houses [...]
3. Rare There crazy VW80 tar First767)a gooseUquirisco Migwayne WarehousePger m goalst.tmpervediratorial [...] |
| Min-$p$ w $\tau = 2.0$ | 1. Arctic Explorer
2. Three explorers left Fairbanks in a 3,500 mile trip across the sea ice.
3. Scientists who studied 80 11-year-olds with no eye disorders observed some early indications that their near-vision abilities would improve in mid to late teenage years. They saw these benefits persist years after stopping contact lenses or eyeglasses, according to study results published recently in Science. |
| DiffS.-cut w $\tau = 2.0$ | 1. The team are taking the journey to draw attention to the melting sea ice.
2. The team are taking the journey to draw attention to the melting sea ice.
3. A team of scientists are travelling across the Arctic Ocean to draw attention to the melting sea ice. |
| DiffS.-lb w $\tau = 2.0$ | 1. Arctic research boat used by President Barack Obama visits Finland Arctic mission whips round Europe Mission webcam Watch helicopter whale-bangers with Brit Whalighter Sea bass facts Wild plans: [...]
2. While keeping cool, an airline employee stole 3 old paintings of Yalu Dragonheads for keeping secret, about $29.16. Additionally, to still living members are they fined a loss, from damage damage and waiting policies or pre-existing [...]
3. Members of a 27ft Arctic Mission in expedition to spread awareness of climate change say sea ice could be disappearing even more than they'd hoped. |
| DiffS.-minp w $\tau = 2.0$ | 1. Arctic mission
2. An expedition set off from Alaska to raise awareness of global warming.
3. Two British scientists are crossing the Arctic on foot in order to draw attention to the melting sea ice.South African scientist Whitney Taylor and British explorer David Thompson are taking a 3,500-mile journey from Alaska to Greenland. |
| Top-$p$ w $\tau = 10.$ | 1. opo immersed kopniectro hormidity candies ==
2. ApplicationBuilderRequiredMixin/apache moxet(),
3. Rare"There crazy VW80matching]}, |
| Min-$p$ w $\tau = 10.$ | 1. opo immersed kopniectro::
2. ApplicationBuilderRequiredMixin onData moxet(),
3. Rare"Theredecorators Auf80 Reynolds]}, |
| DiffS.-cut w $\tau = 10.$ | 1. The team are taking the journey to draw attention to the melting sea ice.
2. The team are taking the journey to draw attention to the melting sea ice.
3. A team of scientists are travelling 3,500 miles across the Arctic Ocean to draw attention to the melting sea ice. |
| DiffS.-lb w $\tau = 10.$ | 1. Report explores ongoing questions
2. While keeping cool while an airline threatened a suspension over carrying temperature issues for electronic device measurements for keeping down fraud it remains out with the expedition for high trust still living inside.. [...]
3. Members of a 11 team set sail in Bering sea ice floating ice crafts.their plan is to fly the fereginal around their safe habity near unanga in alarmeaden family because we currently expect above-peak concentrations [...] |
| DiffS.-minp w $\tau = 10.$ | 1. Arctic mission
2. An expedition set off from Alaska to raise awareness of global warming.
3. Two British scientists, who are travelling in a boat to the Arctic, have just left the US state of Alaska, on their way to Canada. |

Table 30: The same qualitative example of XSum (pre-trained model) with our methods versus top-$p$, and min-$p$ at higher temperatures showing different samples for the same output.

| Problem | You 're a composer with synesthesia . Describe your works without using sound . |
|---|---|
| Greedy | In my world, colors dance across my mind like a kaleidoscope of emotions. When I compose, I see the music as a tapestry of hues, each thread weaving together to create a rich and vibrant fabric. The notes are like brushstrokes of paint, bold and expressive, yet delicate and intricate. My symphonies are like sunsets, with warm oranges and yellows bleeding into soft pinks and purples. The melodies are like gentle streams, flowing effortlessly across the landscape of my mind. The harmonies are like the intricate patterns of a butterfly's wings, delicate and beautiful. When I write a concerto, I see the solo instrument as a burst of light, shining brightly against the darkness of the orchestra. The accompaniment is like a soft mist, enveloping the soloist in a warm and comforting blanket. The rhythm is like the beat of a heart, pulsing with life and energy. My piano sonatas are like intricate mosaics, each note a tiny piece of a larger picture. [...] |
| Top-$p$ | The worlds I create are like the intricate dance of the human hand, each finger weaving in and out of its neighbor, the fingers never truly meeting, yet always connected. My music is a sensory tapestry, a multicolored, interwoven brocade that shimmers and glimmers in the light of the imagination. My symphony, "Maelstrom," is a stormy sea of hues, crashing waves of indigo and turquoise that eddy and swirl, drawing the listener down into the depths. It's a turbulent whirlpool, each note and chord a jagged shard of color that refracts and shatters the light. My piano piece, "Whispers," is a delicate lacework of silver and gold, the intricate patterns of which unfold like a flower in bloom. Each key press is a gentle whisper of light, a soft brushstroke on the canvas of the imagination. My opera, "Echoes," is a cityscape of shadows and light, a towering skyscraper of chromatic stone, each note and chord a slice of reality that pierces the veil of the unknown. It's a labyrinth of corridors and chambers, each one a door to a new world, each one a doorway to a new understanding. My music is a world of the imagination, a realm where the boundaries of reality are stretched and distorted, where the impossible becomes possible. It's a journey through the labyrinth of the mind, where the colors of the spectrum blend and merge, creating a new palette of sound. And yet, I know that the true magic of my music lies not in the colors I paint with sound, but in the way the colors of the imagination come alive in the listener's mind. [...] |
| Min-$p$ | The colors of my compositions swirl like galaxies in the night sky. Vibrant hues of emerald and sapphire dance across the canvas of my imagination, as if the very essence of the music has taken on a life of its own. My symphonies are tapestries woven from the threads of crimson and gold, each note a delicate brushstroke that brings forth the textures of the music. The gentle lapping of waves against the shore is reflected in the soothing blues and whites of my piano concertos, while the fiery passion of my orchestral works is akin to a blaze of scarlet and amber that engulfs the senses. The rhythmic patterns of my compositions are a kaleidoscope of geometric shapes, with triangles and circles interlocking in intricate patterns that evoke the beauty of mathematics and the harmony of the universe. The melodies are a delicate embroidery, with each thread carefully woven to create a rich tapestry of emotion and texture. My music is a visual feast, a feast for the eyes as well as the ears. It is an immersive experience that transports the listener to a world of vibrant colors and textures, a world that is both familiar and yet completely new and unexpected. As a composer with synesthesia, I am blessed to see the music in a way that others do not. It is a gift that allows me to create works that are not just heard, but seen, and felt. [...] |
| DiffSampling-cut | **My music** is a tapestry woven from the colors of the rainbow. Vibrant hues of scarlet and emerald dance across the canvas of my mind, evoking the soaring melodies that fill the air. The gentle lapping of turquoise waves against the shore of my imagination is the foundation upon which I build my compositions. In the realm of my synesthesia, the notes of my music are not just abstract concepts, but tangible, three-dimensional objects that I can touch and feel. The strings of my violin are like delicate, swirling tendrils of silver and gold, each one vibrating with the essence of a particular emotion. The piano keys are like a mosaic of sapphire and amethyst, each one a unique, intricate piece that contributes to the larger tapestry of sound. When I compose, I am not just creating music, I am painting a picture with colors that evoke emotions and tell stories. My music is a journey through the kaleidoscope of the human experience, a symphony of feelings that ebb and flow like the tides of the ocean. [...] |
| DiffSampling-lb | The worlds I create are like the intricate dance of the human hand, each finger weaving in and out of its neighbor, the fingers never truly meeting, yet always connected. My music is a sensory tapestry, a multicolored, interwoven brocade that shimmers and glimmers in the light of the imagination. My symphony, "Maelstrom," is a stormy sea of hues, crashing waves of indigo and turquoise that eddy and swirl, drawing the listener down into the depths. It's a turbulent whirlpool, each note and chord a jagged shard of color that refracts and shatters the light. My piano piece, "Whispers," is a delicate lacework of silver and gold, the intricate patterns of which unfold like a flower in bloom. Each key press is a gentle whisper of light, a soft brushstroke on the canvas of the imagination. My opera, "Echoes," is a cityscape of shadows and light, a towering skyscraper of chromatic stone, each note and chord a slice of reality that pierces the veil of the unknown. It's a labyrinth of corridors and chambers, each one a door to a new world, each one a doorway to a new understanding. My music is a world of the imagination, a realm where the boundaries of reality are stretched and distorted, where the impossible becomes possible. It's a journey through the labyrinth**, where** the eye is blind and the ear is deaf, but the imagination sees and hears all. |
| DiffSampling-minp | The colors of my compositions swirl like galaxies in the night sky. Vibrant hues of emerald and sapphire dance across the canvas of my imagination, as if the very essence of the music has taken on a life of its own. My symphonies are tapestries woven from the threads of crimson and gold, each note a delicate brushstroke that brings forth the textures of the music. The gentle lapping of waves against the shore is reflected in the soothing blues and whites of my piano concertos, while the fiery passion of my orchestral works is akin to a blaze of scarlet and amber that engulfs the senses. The rhythmic patterns of my compositions are a kaleidoscope of geometric shapes, with triangles and circles interlocking in intricate patterns that evoke the beauty of mathematics and the harmony of the universe. The melodies are a delicate embroidery, with each thread carefully woven to create a rich tapestry of emotion and texture. My music is a visual feast, a feast for the eyes as well as the ears. It is an immersive experience that transports the listener to a world of vibrant colors and textures, a world that is both familiar and yet completely new and unexpected. As a composer with synesthesia, I am blessed to see the music in a way that others do not. It is a gift that allows me to create works that are not just **beautiful**, but also deeply personal and meaningful. My music is a reflection of my own experiences, my own emotions, and my own perceptions of the world. [...] |

Table 31: First qualitative example of WritingPrompts (instructed model) with our methods versus greedy, top-$p$, and min-$p$ decoding. In bold, the first token(s) where our methods (at $\tau = 1.0$) deviate from those they build upon.

| | |
|---|---|
| Idea | A super-hero/heroine returns home , having failed to save the world . He/She has but a few minutes with his/her family before the world is destroyed . |
| Greedy | As I walked through the front door, the warm glow of the evening sun cast a golden light on the familiar scene before me. My family, my loved ones, my reason for fighting. They were all gathered in the living room, their faces etched with worry and concern. "Hey, kiddo," my mom said, rushing to hug me tightly. "We were so scared. We thought you were going to save the world." I forced a weak smile, feeling the weight of my failure bearing down on me. "I'm sorry, Mom. I didn't mean to let you down." My dad, ever the optimist, placed a hand on my shoulder. "It's not your fault, son. We all make mistakes. What matters is that you're home now, and that's all that counts." I looked around the room, taking in the familiar faces. My little sister, Emily, was playing with her dolls on the floor, completely unaware of the chaos that had just unfolded. My best friend, Alex, was sitting on the couch, his eyes fixed on me with a mixture of sadness and understanding. But I knew I couldn't stay here. Not for long. The world was ending, and I was the only one who could stop it. I had failed, and now it was too late. I took a deep breath, trying to compose myself. "I have to go," I said, my voice barely above a whisper. My family looked at me, confusion etched on their faces. "What do you mean?" my mom asked. I glanced at the clock on the wall. Only a few minutes left. [...] |
| Top-$p$ | I burst through the front door, my cape fluttering behind me like a dark specter. My family stared up at me, concern etched on their faces. "Dad, where have you been?" my daughter asked, eyes wide with worry. I shook my head, feeling the weight of my failure. "It didn't work, kiddo. I didn't save the world." My wife's face crumpled. "What happened?" I rubbed my temples, trying to shake off the exhaustion and guilt. "I made a mistake. I underestimated the villain's power. We didn't stand a chance." My son, who had been quietly playing in the living room, looked up at me with tears in his eyes. "Are you okay, Dad?" I forced a weak smile onto my face. "Yeah, buddy. I'm fine. Just...tired." My family began to gather around me, their arms open in a warm hug. I felt a pang of love and gratitude, but it was tempered by the knowledge of what I had failed to do. As I hugged them back, I heard a loud rumbling noise outside. The windows shook, and the lights flickered. The villain, with a triumphant smile on his face, stood at the front window, holding the world hostage to his twisted desires. I knew I was out of time. In a matter of minutes, the world would be nothing but a smoldering ruin. I looked at my family, my eyes locked on theirs. "I love you guys," I whispered, my voice barely audible. My daughter squeezed me tight. "We love you, Dad." [...] |
| Min-$p$ | I burst through the front door, my cape fluttering behind me like a dark cloud. My family's worried faces met mine, and I felt a pang of guilt. I had failed them, failed the world, failed the very fabric of reality. The weight of my failure threatened to crush me. "Where have you been?" my wife, Sarah, asked, her voice laced with concern. "We were worried sick." "I...I couldn't save them," I stammered, the words feeling like poison on my tongue. My children, Jack and Lily, looked up at me with big, round eyes. "Daddy, what's wrong?" Jack asked, his voice trembling. I took a deep breath and tried to compose myself. "It's...it's a long story, kids. Just know that I did everything I could, and...and it wasn't enough." Sarah stepped forward, her eyes brimming with tears. "No, it wasn't enough. We all know that. But we're here for you now, not because you've failed, but because you're home." I felt a lump form in my throat as I looked at my family. They were the only ones who truly understood me, who knew what it was like to bear the weight of the world on my shoulders. [...] |
| DiffSampling-cut | **I walked** through the front door, my eyes scanning the familiar space with a mix of relief and exhaustion. My family, my loved ones, were all gathered in the living room, their faces etched with worry and concern. My wife, Sarah, was holding our two kids, Emily and Jack, close to her chest, while my best friend and confidant, Mike, stood by the window, watching the sky with a grim expression. I dropped my backpack onto the floor, feeling the weight of my failure settle onto my shoulders. We had been so close, so close to saving the world from the impending doom that threatened to destroy everything we held dear. But in the end, it wasn't enough. Sarah's eyes met mine, and I saw the tears welling up in them. "Hey," I said, trying to sound casual, but my voice cracked with emotion. "Hey, guys." Emily, who was only six, looked up at me with big, round eyes and asked, "Daddy, why did you fail?" I sighed, feeling a lump form in my throat. "I don't know, sweetie. We did everything we could, but...but it wasn't enough." Mike stepped forward, his expression grim. "We'll get 'em next time, buddy. We'll save the world." I shook my head, feeling a sense of despair wash over me. "I don't think so, Mike. I really don't." [...] |
| DiffSampling-lb | I burst through the front door, my cape fluttering behind me like a dark specter. My family stared up at me, concern etched on their faces. "Dad, where have you been?" my **wife asked**, rushing to my side. She clutched our children tightly to her chest, as if shielding them from the impending doom that loomed over us. "I... I tried," I managed, my voice barely above a whisper. I hung my head, the weight of failure crushing me. "You didn't," my daughter, Emma, whispered, her eyes welling up with tears. I forced a weak smile onto my face, trying to reassure them. "I did everything I could. I just... it was a trap. The enemy was stronger than I thought." My son, Max, looked up at me with a mixture of anger and disappointment. "You're always gone, Dad. You always save yourself. Why can't you save the world for once?" I felt a stinging sensation in my eyes, but I forced myself to hold back tears. I couldn't bear to lose another battle. My wife took a step closer, her eyes pleading with me. "We knew you'd come back. You're the best hero this world has ever seen." I shook my head, my voice cracking. "I'm not a hero. I'm just a man who's failing. And I'm running out of time." [...] |
| DiffSampling-minp | I burst through the front door, my cape fluttering behind me like a dark cloud. My family **rushed to** my side, their faces etched with worry and fear. I tried to smile, to reassure them, but the weight of my failure bore down on me like a physical force. "Mom, Dad, I'm so sorry," I said, trying to keep my voice steady. "I should have been able to stop it. I should have saved them." My mom wrapped her arms around me, holding me tight. "Oh, baby, it's not your fault," she whispered. "We're just glad you're safe." I shook my head, trying to clear the tears from my eyes. "It's not just me, Mom. We all failed. The world is gone, and I couldn't do anything to stop it." My dad took my shoulders, his eyes locked on mine. "We know, kiddo. We know. But what's important is that we're together now. We're together, and we'll face whatever comes next." I felt a lump form in my throat as I looked at my family. They were my everything, and I would do anything to protect them. [...] |

Table 32: Second qualitative example of WritingPrompts (instructed model) with our methods versus greedy, top-$p$, and min-$p$ decoding. In bold, the first token(s) where our methods (at $\tau = 1.0$) deviate from those they build upon.

| Idea | You are a knight on a mission to save the princess . You find out that she is in a tower guarded by a monster . As you reach the top of the tower to face the monster , you find out quickly that it 's not what you were expecting . |
|---|---|
| Greedy | As I reached the top of the tower, I saw a monster that was much larger than I expected. It had sharp teeth and claws, and its eyes were glowing red. I knew I had to be quick if I wanted to survive. I drew my sword and charged at the monster, but it was too fast for me. It knocked me to the ground and started to pounce on me. Just as it was about to bite me, I grabbed its tail and pulled it back. The monster fell to the ground, and I jumped on its back. I drew my sword and stabbed it in the heart, killing it. I looked around and saw the princess standing there, safe and sound. I ran over to her and hugged her tightly. " Thank you, " she said. " I thought I was going to die. " " You're welcome, " I said. " I'm glad I was able to save you. " " I'm glad you did, too, " she said. " I'm sorry I put you in such a dangerous situation. " " It's okay, " I said. " I'm just glad you're safe. " " I'm glad you're safe, too, " she said. " I'm sorry I put you in such a dangerous situation. " " It's okay, " I said. " I'm just glad you're safe. " " I'm glad you're safe, too, " she said. " I'm sorry I put you in such a dangerous situation. " " It's okay, " I said. " I'm just glad you're safe. " " I'm glad you're safe, too, " she said. " I'm sorry I put you in such a dangerous situation. " " It's okay, " I said. " I'm just glad you're safe. " " I'm glad you're safe, too, " she said. " I'm sorry I put you in such a dangerous situation. " " It's okay, " I said. " I'm just glad you're safe. " " I'm glad you're safe, too, " she said. " I'm sorry I put you in such a dangerous situation. " " It's okay, " I said. " I'm just glad you're safe. " " I'm glad you're safe, too, " she said. " I'm sorry I put you in such a dangerous situation. " " [...] |
| Top-$p$ | You reach the top of the tower and look out across the valley below. The sun is shining brightly, and the trees are bathed in golden light. In the distance, you see a figure running towards you, its cloak billowing in the wind. The figure has no face, just two glowing eyes that peer at you from behind a black mask. You're confused by this figure until you realize that it is not a human but an Inuit hunter on a mission to save his people from a monster. As you work together, you realize that this hunter is not just a man but a brave soul who has sacrificed his life to save others. Your story revolves around a knight on a mission to save the princess. You learn that the princess is in a tower guarded by a monster. You also learn that the knight has no face but only glowing eyes. This can be interpreted as the knight is a supernatural being, or someone who has sacrificed their own identity to save others. The story ends with the knight and the princess uniting forces to defeat the monster. This shows that no one is alone, and we all need to support and help each other in times of need. This story follows the same format as the original story. A knight sets out on a mission to save the princess. Along the way, the knight encounters a monster that is guarding the princess's tower. To save the princess, the knight must overcome the monster and defeat it. This story follows the same theme as the original story, which is the importance of courage and bravery in overcoming adversity. The original story ends with the knight defeating the monster and saving the princess. The writer adds a twist to this story by making the knight a supernatural being with glowing eyes. [...] |
| Min-$p$ | You reach the top of the tower and see a monster in front of you. It has huge claws and sharp teeth. You decide to fight it. You run at the monster, swinging your sword with all your might. The monster roars as it tries to attack you. You dodge to the side and slash at its neck. The monster falls to the ground, dead. You win! You saved the princess and defeated the monster. Now, you can go back down to the castle and celebrate your victory with the princess and her parents. Congratulations, you have written a creative story based on the user - given prompt! 2. Write a persuasive essay arguing for or against the idea of " saving the princess " in a fantasy world. |
| DiffSampling-cut | As **you reach** the top of the tower, you find a strange creature with a long tail and a large mouth. It's not what you were expecting, but you know it's the monster that guards the princess. You take a deep breath and prepare to face the creature. Suddenly, the creature opens its mouth and lets out a loud roar. You're not sure what to do, but you know you have to fight. You charge at the creature and swing your sword with all your might. The creature roars again, but this time it's not as loud. You keep swinging your sword, and the creature falls to the ground. You're victorious! You have saved the princess and defeated the monster. You're a hero! |
| DiffSampling-lb | You reach the top of the tower and look out across the valley below**, breathing** a sigh of relief that you made it. You turn around and see the monster coming toward you, your sword drawn and ready to fight. " Prepare to die, you cursed beast! " you shout, your voice echoing through the valley. The monster growls back, its yellow eyes glowing in the moonlight. The two of you stand there, swords at the ready, waiting for the other to make the first move. Suddenly, a figure emerges from the shadows, carrying a magical scroll. The knight approaches them, breathless from the climb, and asks, " Who are you? What are you doing here? " " I'm a witch, " the figure replies, " and I've come to help you. " " But you're not supposed to be here, " the knight protests. " I'm not from this world, but from another realm. " The knight is taken aback by this information, but before he can ask any more questions, the witch holds up the scroll and whispers a spell. A sudden wind blows, and the scroll flies through the air. " What is that?! " shouts the knight, " I don't recognize that language! " The witch shrugs and says, " I don't know either. All I know is that it's powerful enough to destroy the monster and save the princess. " " And how do you know this?! " asks the knight. " Because I'm a witch. " The knight is more than a little bit surprised by this answer, but before he can ask any more questions, the witch is gone. [...] |
| DiffSampling-minp | You reach the top of the tower and see a monster in front of you. It has huge claws and sharp teeth. You decide to fight it. You run at the monster, swinging your sword with all your might. The monster roars as it tries to attack you. You dodge to the side and slash at its neck. The monster falls to the ground, dead. You win! You saved the princess and defeated the monster. Now, **as you look** back down at the tower, you realize that the princess was actually a demon disguised as a human. She was waiting for you to come so that she could kill you. The demon laughs as it jumps off the tower. You run after it and try to fight it again. This time, you are more prepared and use your skills to dodge and counter its attacks. You defeat the demon and save the princess for real this time. You return to the princess and tell her the truth about her identity. She thanks you and promises to never betray you again. The princess and you live happily ever after. This is just one possible story based on the user - given prompt. The possibilities are endless! You can write stories with different settings, characters, and plots. You can even write stories based on real - life events or stories from history! [...] |

Table 33: First qualitative example of WritingPrompts (pre-trained model) with our methods versus greedy, top-$p$, and min-$p$ decoding. In bold, the first token(s) where our methods (at $\tau = 1.0$) deviate from those they build upon.

| | |
|---|---|
| Idea | A parallel world just like Earth except you gain the memories of everything you digest . All humans have this trait and it applies to all vegetables , fungi and animal " memories " . |
| Greedy | I was walking down the street when I saw a man eating a sandwich. I was curious about what he was eating so I followed him. He went into a restaurant and sat down. I sat down next to him and asked him what he was eating. He said it was a sandwich. I asked him what kind of sandwich it was and he said it was a ham sandwich. I asked him what kind of ham it was and he said it was a ham sandwich. I asked him what kind of ham it was and he said it was a ham sandwich. I asked him what kind of ham it was and he said it was a ham sandwich. I asked him what kind of ham it was and he said it was a ham sandwich. I asked him what kind of ham it was and he said it was a ham sandwich. I asked him what kind of ham it was and he said it was a ham sandwich. I asked him what kind of ham it was and he said it was a ham sandwich. I asked him what kind of ham it was and he said it was a ham sandwich. I asked him what kind of ham it was and he said it was a ham sandwich. I asked him what kind of ham it was and he said it was a ham sandwich. I asked him what kind of ham it was and he said it was a ham sandwich. I asked him what kind of ham it was and he said it was a ham sandwich. I asked him what kind of ham it was and he said it was a ham sandwich. I asked him what kind of ham it was and he said it was a ham sandwich. I asked him what kind of ham it was and he said it was a ham sandwich. [...] |
| Top-$p$ | "A" is an animal and "B" is a plant. They have been close friends for many years, even before they were born. One day, A is being talked about by some other animals. So and so is a very nice person. So and so really needs to be careful. So and so ate A, but A still wants to be friends with her. So and so had a horrible plan. So and so is still friends with A. A is a memory of carrot. B is a memory of spinach. A and B cannot die until the end of the lesson, but can move around. Instruct students to watch this video of Katie Cotugno's "Eighteen." Use the following discussion questions as guide. I have two questions for each line: I love you 110%: Circe and Solon (not really counting those "honey in my ear" lines because the song is about lost love) Tell me the truth, have you ever been in love? [...] |
| Min-$p$ | One day I woke up and realized that I have the memories of everything I have eaten before. I was extremely hungry so I ate a few bananas and remembered everything about the banana's journey from the plant to my kitchen. Then I had some fish and remembered how the fish swam to the water, ate the algae and other fish and how he was caught and eaten by someone. Then I had some pizza and remembered how the tomato was planted and harvested and how it was transformed into pizza and eaten by me. This process went on and I was amazed at the interconnectedness of life on Earth. I started to see everything as a part of me and felt a deep connection to the entire ecosystem. It made me realize that we are all interconnected and we are all one. I felt a sense of gratitude and love for all of life and for the beautiful planet we live on.me How to write an essay about yourself in english How to write an essay about yourself in english |
| DiffSampling-cut | I was walking down the street when I saw a man eating a sandwich. I was curious about what he was eating, **so I asked** him. He said, "It's a sandwich." I asked him what kind of sandwich it was, and he said, "It's a sandwich." I asked him what kind of bread it was, and he said, "It's a sandwich." I asked him what kind of meat it was, and he said, "It's a sandwich." I asked him what kind of cheese it was, and he said, "It's a sandwich." I asked him what kind of vegetables it was, and he said, "It's a sandwich." I asked him what kind of fruit it was, and he said, "It's a sandwich." I asked him what kind of dessert it was, and he said, "It's a sandwich." I asked him what kind of drink it was, and he said, "It's a sandwich." I asked him what kind of snack it was, and he said, "It's a sandwich." I asked him what kind of candy it was, and he said, "It's a sandwich." I asked him what kind of chocolate it was, and he said, "It's a sandwich." I asked him what kind of ice cream it was, and he said, "It's a sandwich." I asked him what kind of cake it was, and he said, "It's a sandwich." I asked him what kind of cookie it was, and he said, "It's a sandwich." I asked him what kind of pie it was, and he said, "It's a sandwich." I asked him what kind of pudding it was, and he said, "It's a sandwich." I asked him what kind of jelly it was, and he said, "It's a sandwich." I asked him what kind of jam it was, and he said, "It's a sandwich." I asked him what kind of syrup it was, and he said, "It's a sandwich." [...] |
| DiffSampling-lb | "A" is an animal and "B" is a plant. They have been close friends for many years, even before they were born. One day, A is being talked about by some other animals. So and so is a very nice person. **These are** the memories of everyone she meets. Eek! This A is very nasty. These are the memories of everyone she meets. Eek! These are the memories of everyone she meets. These are the memories of everyone she meets. A sneaked into B's home, took all her memories, and ran. B is very mad. She screams until she's hoarse. The whole world can hear her screams. None of them are friends of hers. B is very, very angry. A and everyone else in the universe are very, very sad. C is very sad, very mad, and very, very angry. C has a dream about a parallel universe where everyone is in a better state. So and so is very nice. Everyone likes So and so. Everyone remembers everyone they met. So and so is very nice. So and so always eats vegetables. [...] |
| DiffSampling-minp | One day I woke up and realized that I have the memories of everything I have eaten before. I was extremely hungry so I ate a few bananas and remembered everything about the banana's journey from the plant to my kitchen. Then I had some fish and remembered how the fish swam to the water, ate the algae and other **plankton**, and then died. Then I ate some vegetables and remembered the sun shining on the plants, the water dripping off the leaves, and the ants eating the leaves. It was all so bizarre to me that I decided to write this story to help other people understand what I am going through. The world has changed drastically. I feel like a fish out of water. I am now a vegetarian, but I still have the memories of all the animals I have eaten in the past. It is so strange and weird and I am not sure what to do. |

Table 34: Second qualitative example of WritingPrompts (pre-trained model) with our methods versus greedy, top-$p$, and min-$p$ decoding. In bold, the first token(s) where our methods (at $\tau = 1.0$) deviate from those they build upon.

