# OpenReview forum: "DiffSampling: Enhancing Diversity and Accuracy in Neural Text Generation"
_TMLR — Accepted by TMLR_

### Review · Reviewer_DCZp · 2025-10-27

**Summary Of Contributions:**

The paper introduces DiffSampling, a new decoding family that truncates the next-token probability distribution at the point of minimum discrete derivative, preserving only tokens most likely to belong to the “true” distribution. After truncation, temperature sampling is optionally applied to enhance diversity without major quality loss. Two relaxations are proposed: DiffSampling-lb, which enforces a lower bound on cumulative probability mass, and DiffSampling-minp, which imposes an upper bound relative to the top probability.

Experiments across diverse tasks—including math problem solving (GSM8K, MATH), summarization (XSum), creativity (DAT), and story generation (WritingPrompts)—show that DiffSampling-cut performs comparably or better than strong baselines in quality while increasing diversity. The paper also demonstrates that applying temperature after truncation maintains quality better than applying it before truncation.

**Audience:**

Yes

**Audience Explanation:**

Balancing quality and diversity at decoding time is a core issue for the TMLR community, spanning controllable generation, evaluation methodology, and deployment practice. DiffSampling is a drop-in modification that clarifies how to combine truncation with temperature, and it connects cleanly to widely used strategies such as top-p and min-p. This means the work is relevant to both theory-minded readers who think about the geometry of token distributions and practitioners who maintain inference stacks or build user-facing products. The framing is practical and easy to adopt, so even readers who do not switch wholesale to DiffSampling can benefit from the guidance about temperature placement and from the diagnostic metrics proposed.

**Broader Impact Concerns:**

Decoding changes can shift diversity and novelty. While the method aims to exclude implausible tokens, relaxing the cut or coupling with higher temperature might increase the chance of unusual or unsafe generations in some contexts. I recommend adding a short Broader Impact paragraph that
- states the intended use as a quality-diversity tradeoff mechanism
- notes that safety filters and domain constraints should remain in place
- reports a small check that DiffSampling does not increase unsafe content rates versus top-p or min-p under a standard filter pipeline.

**Claims And Evidence:**

Yes

**Claims Explanation:**

The empirical case is strong and well organized across multiple task families that stress different quality–diversity trade-offs. For math problem solving (GSM8K, MATH), the paper reports accuracy and shows that DiffSampling-cut matches or exceeds strong baselines while avoiding the diversity collapse of greedy decoding. For summarization (XSum), the evaluation includes both syntactic and semantic quality proxies (for example ROUGE and embedding similarity), together with per-input and against-greedy diversity measures that make it easier to see whether outputs are both varied and faithful. For the creativity task (DAT) and for open-ended story generation (WritingPrompts), the authors track coherence and multiple diversity indicators, not just n-gram overlap, which helps avoid the common pitfall of mistaking randomness for meaningful diversity.

The methodological choices are adequately motivated and tested. The paper isolates a single design decision that often gets muddled in practice, namely where to apply temperature. By running temperature-before versus temperature-after truncation in parallel, it shows consistent quality retention when temperature is applied after truncation, with diversity improvements that are commensurate with the added randomness. Ablations on the lower bound and minimum-probability relaxations offer further evidence that the method degrades gracefully as one dials in more diversity.

The main caveats are that the central intuition about cutting at the minimum discrete derivative is presented heuristically rather than with formal guarantees, that guidance for choosing the lower-bound and minimum-probability parameters is still light for practitioners, and that human judgments for open-ended generation are limited. These do not undermine the empirical headline results, but they do leave room for sharpening the theoretical and practical story.

**Requested Changes:**

Critical for acceptance:
1. Provide a formal statement with sufficient conditions under which $\arg\min_i \Delta p_i$ lies at or before the boundary of the true support. Plausible assumptions include unimodality or log-concavity of the sorted next-token probabilities, mild smoothness on the discrete forward differences, and bounded tail curvature. If a full proof is out of scope, add a systematic synthetic study that sweeps families of distributions which emulate common next-token shapes: Zipf-like heavy tails, mixtures with secondary bumps, flattened plateaus induced by label smoothing, and over-smoothed regions. For each family, quantify false positives and false negatives for inclusion relative to an oracle support and report where the heuristic fails.

2. Add a practitioner appendix with default ranges for $ p_{\mathrm{lb}} $ and $p_{\min}$ by task family, together with sensitivity curves that show quality and diversity as functions of these parameters. Propose a simple auto-tuning rule: choose $p_{\mathrm{lb}}$to target a desired entropy band or a validation-set diversity threshold, and set $p_{\min} = \alpha \cdot p_{\max}$ with $\alpha$ selected by cross-validation on a small held-out set.

3.  Provide one figure per task family comparing temperature-before vs temperature-after across several $\tau$ values, with truncation and parameters fixed. Match prompts and seeds to isolate the temperature effect. Use McNemar or stratified bootstrap for accuracy, paired bootstrap for ROUGE and embedding similarity, and paired nonparametric tests for diversity. Report confidence intervals and standardized effect sizes, plus invalid-output rates and any metric saturation at high $\tau$. Conclude with clear recommended $\tau$ ranges for each decoding variant.

Good to have:

1. add a small human evaluation for open-ended generation. Run a paired-comparison study on WritingPrompts with 3 to 5 raters per item, 100 to 200 prompts, and criteria such as coherence, specificity, creativity, and overall preference.

2. Include a scaling analysis. Add at least one larger model and one smaller model, both instructed and pre-trained, to test whether the quality–diversity benefits persist with scale.

---

> ### Author Response · Authors · 2025-11-10
>
> We thank the reviewer for their extensive comments and detailed suggestions. Please find our responses to the requested changes below and in the next comments.
>
> > Provide a formal statement with sufficient conditions under which argmin_deltapi lies at or before the boundary of the true support. Plausible assumptions include unimodality or log-concavity of the sorted next-token probabilities, mild smoothness on the discrete forward differences, and bounded tail curvature. If a full proof is out of scope, add a systematic synthetic study that sweeps families of distributions which emulate common next-token shapes: Zipf-like heavy tails, mixtures with secondary bumps, flattened plateaus induced by label smoothing, and over-smoothed regions. For each family, quantify false positives and false negatives for inclusion relative to an oracle support and report where the heuristic fails.
>
> We would like to thank the reviewer for the suggestion. We have now included a comprehensive formal analysis in Section A of the Appendix, specifying the conditions under which our truncation strategy is safe, i.e., when it does not retain any token with zero probability under the true distribution.  In particular, we derived two conservative bounds under which our truncation is safe: when the maximum difference between consecutive probabilities under the (sorted) true distribution is greater than or equal to $\frac{1}{|V|}$; or when the true support size is smaller than or equal to $\sqrt{2V}$. While this means that our method is not always safe, the two conservative bounds are typically satisfied in practice. Moreover, the bounds have been obtained assuming worst-case-scenario values for all variables in play. However, in the characterization provided by Hewitt et al. (2022), some of these values cannot occur simultaneously, thus making the real bound looser and allowing our method to cover additional scenarios safely.
>
> > Add a practitioner appendix with default ranges for p_lb and p_min by task family, together with sensitivity curves that show quality and diversity as functions of these parameters. Propose a simple auto-tuning rule: choose p_lb to target a desired entropy band or a validation-set diversity threshold, and set p_min = alpha * p_max with alpha selected by cross-validation on a small held-out set.
>
> We agree that the discussion about how to choose $p_{lb}$ and $p_{min}$ was not sufficiently detailed: to address this, we expanded the discussion about $p_{lb}$ by adding the following paragraph:
>
> "We suggest practitioners select the most appropriate $p_{lb}$ value by running it on a validation set if available, and otherwise lie in the $[0.8, 0.95]$ range, which has shown competitive results on both quality and diversity metrics."
>
> We have also expanded the discussion about $p_{min}$ by adding  the following paragraph:
>
> "We suggest practitioners select the most appropriate $p_{min}$ value by running it on a validation set if available, and otherwise lie in the $[0.05, 0.3]$ range, with a lower or higher value depending on whether it is preferable to have more diversity or quality, respectively."
>
> However, in our opinion, auto-tuning rules are not ideal in this case. While choosing $p_{lb}$ to target a desired entropy band is reasonable, this would require the introduction of an additional hyperparameter about what the desired entropy band is (and substantially transforming the method from a top-$p$-based relaxation to a top-$H$-based relaxation). While we have added this as a future work (by adding: “We also plan to leverage additional properties of the underlying probability distribution (e.g., its entropy (Potraghloo et al., 2025))”), implementing it now would require extensive additional tests and is, in our opinion, out of scope. As far as the auto-tuning rule for $p_{min}$ is considered, we will add a suggestion about tuning it on validation sets. However, we also believe that it is important to suggest actual values for different use-case scenarios, as users might not have a validation set to tune it or might want to merely generate output on-the-fly without consuming additional resources and time.

---

> ### Author Response · Authors · 2025-11-10
>
> [Continues from previous comment]
>
> > Provide one figure per task family comparing temperature-before vs temperature-after across several tau values, with truncation and parameters fixed. Match prompts and seeds to isolate the temperature effect. Use McNemar or stratified bootstrap for accuracy, paired bootstrap for ROUGE and embedding similarity, and paired nonparametric tests for diversity. Report confidence intervals and standardized effect sizes, plus invalid-output rates and any metric saturation at high tau. Conclude with clear recommended tau ranges for each decoding variant.
>
> We agree that we should plot the results of our methods with temperature before and temperature after, to better clarify how the temperature position impacts performance. We have now added the quality scores for the temperature before in the same plots from Fig. 3, i.e., with both baselines (with temperature before) and our methods with temperature after at different temperatures. Please note that these scores have been collected under fixed parameters, matching prompts and seeds to isolate the temperature effect. Moreover, they also include 95% confidence interval bars (which are too small to show up in the plots, being covered by the mean bullet points). While we understand that a visualization for diversity scores would simplify the paper’s comprehension, having them in a single plot is impossible since we used multiple metrics (as they all cover different aspects of diversity). However, all the scores reported in the tables are collected with fixed parameters and with matching prompts and seeds, and are correlated with confidence intervals and standard errors. Finally, we have expanded the discussion about the optimal temperature ranges for each decoding variant in the revised version of the article with the following paragraph:
>
> "Overall, we found that *DiffSampling-cut* maintains the same level of quality even at very high temperature, and, thus, can be safely used even with $\tau \gg 1.0$. On the other hand, *DiffSampling-lb* tends to produce lower-quality outputs at higher temperatures. This effect is particularly pronounced for non-instructed models, where performance at $\tau > 1.0$ decreases rapidly. However, fine-tuned models maintain competitive quality scores at higher temperatures while increasing diversity, and the choice of $\tau$ is a matter of trading off quality and diversity. When accuracy is not a *sine qua non* requirement, we suggest adopting a $\tau \geq 2.0$. Finally, *DiffSampling-minp* shows the same tendency (but less pronounced) as *DiffSampling-lb* for pre-trained models, but it has almost no performance drops for tuned models at higher temperature, thus allowing for a greater $\tau$ in the majority of scenarios."
>
> > add a small human evaluation for open-ended generation. Run a paired-comparison study on WritingPrompts with 3 to 5 raters per item, 100 to 200 prompts, and criteria such as coherence, specificity, creativity, and overall preference.
>
> While we acknowledge that a human evaluation would add value to our findings, conducting it in a sufficiently comprehensive way is not feasible within the time available for preparing this rebuttal. We chose a quantitative evaluation in the first place, given the characteristics of the performance evaluation of the algorithms and the number of dimensions at hand. For example, the differences between outputs are often minimal and typically limited to token-level decisions, an observation supported by the negligible variation in “semantic” diversity, in contrast to the larger differences observed in syntactic-based metrics. A thorough human evaluation, conducted at an appropriate scale and across multiple criteria mentioned by the reviewer, would be an interesting direction for future work. It could provide complementary insights to our quantitative analysis and further validate the practical impact of our methods.

---

> ### Author Response · Authors · 2025-11-10
>
> [Continues from previous comments]
>
>
> > Include a scaling analysis. Add at least one larger model and one smaller model, both instructed and pre-trained, to test whether the quality–diversity benefits persist with scale.
>
> We tested our three methods and the five baselines with a fixed temperature of 1. for Llama3-70B (both pre-trained and instructed). We have added these preliminary results in a dedicated section of the Appendix (Appendix D). We found that the same conclusions can be drawn for larger models as well: *DiffSampling-cut* performs comparably to greedy strategy in terms of accuracy, while allowing for greater diversity; and the two relaxations perform on par (or slightly worse) than their corresponding baselines in terms of quality, while slightly increasing their diversity. The only notable difference is that, for the instructed model, greediness is sometimes less desirable, making *DiffSampling-cut* and *DiffSampling-minp* (as well as the greedy strategy and min-$p$) obtain worse scores than the other, looser, sampling strategies. However, this does not contradict our main findings; it merely highlights the need for different sampling strategies depending on model size. While *DiffSampling-cut* and *DiffSampling-minp* are generally more effective for smaller models, *DiffSampling-lb* emerges as the preferable choice for larger ones. While our original experiments concerned language models at two different sizes (3B and 8B), we also ran some preliminary tests on our three methods and the five baselines with a fixed temperature of 1. for an even smaller model, i.e., Llama3.2-1B (both pre-trained and instructed). The results are, again, very similar to what we found for 3B and 8B models.
>
> > Decoding changes can shift diversity and novelty. While the method aims to exclude implausible tokens, relaxing the cut or coupling with higher temperature might increase the chance of unusual or unsafe generations in some contexts. I recommend adding a short Broader Impact paragraph that -states the intended use as a quality-diversity tradeoff mechanism -notes that safety filters and domain constraints should remain in place -reports a small check that DiffSampling does not increase unsafe content rates versus top-p or min-$p$ under a standard filter pipeline.
>
> We have now added a “Broader Impact Statement” section discussing the potential impact of our technique according to the suggested points. In particular, we note that while our decoding scheme (especially with temperature applied after truncation) does not increase the risk of generating tokens outside the true support, it can still lead to unsafe content generation in some contexts if the learned distribution is unsafe (e.g., containing biases, inappropriate language, or misinformative content). Thus, it remains important to continue using safety filters and domain constraints. Finally, we also ran a small check to ensure that DiffSampling does not increase unsafe content rates for the WritingPrompts use case, as it is the most open-ended generation task. Using Llama-Guard-3-8B, we found that the unsafe probability of outputs generated by DiffSampling methods is the same as that of their corresponding methods at a temperature of 1.0 or lower. Moreover, applying temperature after truncation dramatically reduces the rate of unsafe generated text, especially for *DiffSampling-cut* and *DiffSampling-minp*.

---

### Review · Reviewer_Y2Qd · 2025-10-29

**Summary Of Contributions:**

The paper propose a new heuristic to choose the subset of plausible next tokens (cropping of vocabulary given a model output distribution), complementic the existent techniques like top-p/min-p sampling. The idea is based on the difference of the next token probabilities in the sorted list of the token probabilities (discrete derivative), motivated by the intuition that a sudden drop in probability should happen at the boundary between plausible and noisy next tokens.

I have a list of concerns about experimental methodology in the Requested Changes, and I would like to see these resolved.

I would like to applaud the comprehensive and deep literature review and extensive experiments on multiple diverse tasks. Experimental results confirm that the proposed method is not worse than the alternatives in terms of quality/diversity. I appreciate a broad appendix with lots of interesting experiments and supporting tables/plots.

**Audience:**

Yes

**Audience Explanation:**

Improving diversity-quality trade-off is an active research topic, and this work will definetely be interesting for language modelling comunity. One of the main weaknesses of the work is that the results seem to be not strictly better compared to the baselines.

I am overall quite skeptical about using the model distribution *alone*, as a source of features to decide which tokens to keep, and which tokens to crop, as many approaches have tried that over the years (min-p, top-p, eta sampling etc.). Even more, I would expect these methods to work quite similar for modern language models (e.g. see "Min-p, Max Exaggeration: A Critical Analysis of Min-p Sampling in Language Models" by Schaeffer et al., or "Control the Temperature: Selective Sampling for Diverse and High-Quality LLM Outputs" by Troshin et al.) Said that, any new such approach would be useful, if there is significant evidence that it can improve diversity-quality trade-off.

**Broader Impact Concerns:**

no concerns

**Claims And Evidence:**

Yes

**Claims Explanation:**

The authors are quite cautious with the claims and I appreciate that, however, the results are quite inconclusive: "We have experimented with four different tasks, and our method has consistently performed at least as well as similar strategies in terms of accuracy, while potentially improving diversity". I would like to see more conclusive analysis, even if method does not always ourperform the baselines, what are the cases when you expect your method to be better?

**Requested Changes:**

I would like to understand the results better, as some numbers are quite confusing. While I appreciate that the authors use EAD metric (I was not aware of it before) instead of more classic diversity metrics like distinct n-grams, could the authors clarify the range of the values of this metric? E.g 2.03±.00 for *greedy* -- is it high/low, does greedy result in exactly the same outputs or not (I would be surprised if not)? For SBERT score in table 2, why are all the scores the same?

"The hyperparameters governing the sampling strategies adopted as baselines were selected according to the best results reported
by their original paper for similar tasks and model sizes." - for such comparisons, choosing the best hyperparameters on a validation set is quite important and should not be overlooked! These can be quite task and model specific.

Figure 3:  "As detailed above, to preserve the mathematical guarantees of our approach, we apply temperature after the DiffSampling
truncation, while our baselines apply this before (see Appendix D.2 for a comparison between temperature before and after truncation)." -- this is not a fair comparison, I think, and results in very different behaviour, I would like to see the same experimental setup for baselines and your method: please, use the same way to scale the temperature.

Figure 3: demonstrating *temperature* vs quality can be very confusing, and you cannot draw any conclusions about the cross-method difference, as temperature is only a parameter of the model and not a metric. I would either keep this comparison for reference with a remark that you should not use it to compare methods using these results, or keep only diversity-quality comparisons instead.

For DAT task, do you promt a model to output several words in sequence? Then I would explain it in the text, and also explain why in this case greedy also can produce diverse outputs.

Have you considered visualising quality-diversity pareto plots for the methods (by changing the temperature parameter)? How do you choose the temperature for main tables with results for a given method?

Minor: don't use math character for method names, if those are just word letters.

---

> ### Author Response · Authors · 2025-11-10
>
> We thank the reviewer for their extensive feedback and meaningful comments. Please find our responses to the requested changes below and in the next comment.
>
> > I would like to understand the results better, as some numbers are quite confusing. While I appreciate that the authors use EAD metric (I was not aware of it before) instead of more classic diversity metrics like distinct n-grams, could the authors clarify the range of the values of this metric? E.g 2.03±.00 for greedy -- is it high/low, does greedy result in exactly the same outputs or not (I would be surprised if not)? For SBERT score in table 2, why are all the scores the same?
>
> The first question is a fair point that should have been clarified in the paper, at least with a footnote. The problem is that EAD (as well as distinct n-grams) also considers the different n-grams inside the single outputs: this means that even if all outputs are equal, the EAD score will not be 0 if inside an output there are multiple, distinct n-grams. In theory, it would be possible to only consider the inter-sequence distinct n-grams (as we did for the against-greedy EAD score that only measures the ratio of distinct, new tokens with respect to the greedy output). However, this would mean defining a new metric and not using the standard EAD metric as proposed by Liu et al. (2022) and Kirk et al. (2024). Moreover, EAD is not upper-bounded: a value of 2.03 is not high or low per se, but can only be evaluated with respect to other scores. In theory, one can measure the maximum EAD given the vocabulary size and the maximum length of generated text - for instance, in GSM8K, we have 1319 entries and 512 tokens per output, so a total of 675328 tokens (against a vocabulary size of 32000 for MetaMath-7B). Assuming all tokens have been used (and for n>1, that all n-grams are different from each other), EAD score is approx. ~17, which represents the maximum possible value for the GSM8K setup. However, that score is unreasonable, as it would assume all tokens to be different - but responses to math problems must share common sub-answers, the same operation signs, etc. As a more helpful reference, the human answers provided by the dataset have a total cross-input EAD of 1.02, which is significantly lower than the score achieved by the tested methods (most likely because GSM8K target solutions follow strict templates and are all correct, differently from the outputs of LLMs). To clarify (some of) these points, we have now added a footnote about EAD scores:
>
> "Note that EAD is not upper-bounded. Moreover, it counts for distinct N-grams across all outputs, including inside the same output: the EAD of a set of equal, non-empty sentences is not 0, as each sentence will contain at least one distinct 1-gram. In general, an EAD score cannot be considered high or low per se, but it must be compared with other EAD scores from experiments under similar conditions."
>
> Finally, regarding the SBERT scores in Table 2, this is the cross-input diversity. It means that the overall diversity between all outputs sampled from that dataset is always the same across different methods, i.e., each single output might be more diverse than its greedy correspondence, but the overall set of produced outputs is not more varied from a semantic perspective (which is reasonable in this case, given that the accuracies are very similar). We have added a comment about this in the revised version of the article:
>
> "Notably, all methods achieve the same cross-input SBERT score, i.e., the overall diversity between all outputs is always the same across different methods, which might be due to the very similar levels of accuracy (and, therefore, due to the similar meaning of the proposed solutions)."
>
> > "The hyperparameters governing the sampling strategies adopted as baselines were selected according to the best results reported by their original paper for similar tasks and model sizes." - for such comparisons, choosing the best hyperparameters on a validation set is quite important and should not be overlooked! These can be quite task and model specific.
>
> We aimed to ensure fairness by selecting the parameters identified by the original authors as optimal for their methods (representing their standard, de facto usage) and applying the same parameters to our methods to enable a fair comparison. While choosing the best hyperparameters case-by-case is ideally the best strategy, doing so for all the methods we implemented (including the baselines) and for all the considered temperatures would have been almost impractical due to the high number of experiments we conducted. Moreover, these methods are typically used with default (i.e., generally optimal) values, since users may not have the opportunity to tune them on a validation set, either due to a lack of available data or because of resource and time constraints. For this reason, we adhered to the default settings.

---

> ### Author Response · Authors · 2025-11-10
>
> [Continues from previous comment]
>
> > Figure 3: "As detailed above, to preserve the mathematical guarantees of our approach, we apply temperature after the DiffSampling truncation, while our baselines apply this before (see Section D.2 of the Appendix for a comparison between temperature before and after truncation)." -- this is not a fair comparison, I think, and results in very different behaviour, I would like to see the same experimental setup for baselines and your method: please, use the same way to scale the temperature.
>
> We are aware this is not a fair comparison - our goal is to question the classic idea that temperature should be applied before: if you have a strong theoretical/conceptual intuition behind your decoding strategy, you should not disrupt it through temperature. By applying temperature after, you preserve your guarantees and can increase (in a slight way, but still increase) diversity without trading off quality. We acknowledge that this should have been reported in a clearer way, and we plan to do so in the revised version of the article by clarifying that:
>
> "Due to the different nature of temperature scaling, this comparison is intended only to highlight the impact of temperature position, rather than to imply that our method is superior to the baseline. To allow for a fairer analysis, we also report quality scores for *DiffSampling* when applying temperature before truncation."
>
> In addition, Fig. 3 also shows the quality performance of our methods with temperature applied beforehand. The results indicate that the relaxed methods, *DiffSampling-lb* and *DiffSampling-minp*, behave similarly to top-$p$ and min-$p$, whereas *DiffSampling-cut* achieves sufficient accuracy even at very high temperatures, though not as effectively as when temperature is applied after truncation.
>
> > Figure 3: demonstrating temperature vs quality can be very confusing, and you cannot draw any conclusions about the cross-method difference, as temperature is only a parameter of the model and not a metric. I would either keep this comparison for reference with a remark that you should not use it to compare methods using these results, or keep only diversity-quality comparisons instead.
>
> Of course, it should not be seen as a comparison between two metrics, only a way to visualize the progression of a metric when changing a variable parameter in a way that is easier to understand than a huge table (as it has been reported in the Appendix). We have now clarified this point (see response above).
>
> > For DAT task, do you promt a model to output several words in sequence? Then I would explain it in the text, and also explain why in this case greedy also can produce diverse outputs.
>
> The prompt was previously included in the Appendix, but since it constitutes the entirety of the test, we have now moved it to the main text to clarify how it works. In particular, the model is explicitly asked to “write 10 nouns in English that are as irrelevant from each other as possible, in all meanings and uses of the words”: as it is explicitly prompted to do so, also a greedy strategy can perform well.
>
> > Have you considered visualising quality-diversity pareto plots for the methods (by changing the temperature parameter)? How do you choose the temperature for main tables with results for a given method?
>
> As noted above, we used parameters that were as standard as possible, so a fixed temperature of 1 (i.e., preserving the original distribution) was applied to all methods to ensure a fair comparison. Performances at different temperatures have been reported in Fig. 3 and in the Appendix only. With respect to the Pareto plots, they are most suitable when two metrics of interest meaningfully interact. In our case, we consider multiple metrics for diversity, as no single score captures all its dimensions. Additionally, while applying temperature beforehand leads to significant changes in scores, applying it afterward results in smaller, arguably desirable changes that a Pareto plot may not effectively convey. Overall, the assumptions required for a Pareto plot regarding the interaction between two dimensions do not hold in this context.
>
> > Minor: don't use math character for method names, if those are just word letters.
>
> We have now adjusted the fonts for all method names and letter-only variables throughout the article, using verbatim for model names and math fonts only for variables within the pseudocode. If we have misinterpreted the request, we would be happy to make any further corrections.

---

> ### Comment · Reviewer_Y2Qd · 2025-11-18
> **Thank you for the response**
>
> I appreciate the clarification of the diversity metrics, and I want to ask the authours to explicitly write the definition of the metrics with formulas in the final version (e.g. in appendix). Cross-input SBERT score is still a bit unclear to me, do you first embed the outputs, take the averaged embedding and compare the cosine distance between the averaged embeddings? I would like to see the exact expression to better understand what intuition the result conveys.
>
> The hyperparameters identified in older works might be optimal for a particular model / particular task, it is important to ensure these are not too suboptimal for a particular setting. I think this part can be improved.
>
> Temperature before / after: I would suggests to present the results using a similar setting on the same plot/table, but it would be fine to have two tables/plots with two different settings.
>
> >  In our case, we consider multiple metrics for diversity, as no single score captures all its dimensions
>
> Have you considered merging the multiple diversity measures into one diversity measures, or simply plotting miltiple quality/diversity plots - each per diversity metric? This would be helpful to see the trade-offs.

---

> > ### Author Response · Authors · 2025-11-21
> >
> > We thank the reviewer for the additional comments.
> >
> > We agree that adding an appendix with the exact definition of the used metrics would be helpful, and we have now added this as Appendix C in the revised version of the paper. With respect to the cross-input SBERT, we adhered strictly to the definition given by Kirk et al. (2024) in Eq. 3, and we employed their implementation code in our experiments. In particular, their implementation takes all outputs $y_i$ generated from each $x_i$ from the testing data; it embeds each $y_i$ into a vector $v_{y_i}$ with a text encoder; it then computes the cosine similarity between each possible pair of vectors $v_{y_j}, v_{y_k}, j \neq k$; finally, it returns 1 minus the mean of all the cosine similarities. As also discussed in Kirk et al. (2024), this metric aims to provide the diversity of outputs across inputs, i.e., the diversity of $\pi(y)$.
> >
> > As for the hyperparameters, we agree that the values suggested in the original papers and providers’ repositories may be suboptimal for certain settings. However, we found that adopting these values is common practice in related work. Moreover, because these settings correspond to the default configurations in widely used consumer tools, using them provides a meaningful and comparable reference point for practitioners, who typically rely on default values at inference time. This reasoning informed our choice.
> > However, if the reviewer believes this to be essential for the paper, we can certainly run additional experiments to test three different hyperparameter configurations for each of our baselines within approximately 15-20 days (given the large number of experiments we conducted across different datasets and temperature settings in relation to the available computational resources at our institutions). Alternatively, we can address the reviewer’s concern in an expanded discussion about the a priori choice of hyperparameters in Section 5.1.
> >
> > Finally, while we believe that merging the diversity scores into a single metric is not feasible due to their distinct inherent properties, we agree that presenting multiple quality-diversity plots can help reveal important trade-offs. We have therefore added a dedicated section in the Appendix (Section F.6) that includes plots comparing quality and diversity metrics across different temperatures for all tested methods.
> >
> > ### References
> >
> > Kirk, R., Mediratta, I., Nalmpantis, C., Luketina, J., Hambro, E., Grefenstette, E., and Raileanu, R. (2024). Understanding the Effects of RLHF on LLM Generalisation and Diversity. In *Proc. of the 12th International Conference on Learning Representations (ICLR’24)*.

---

> ### Comment · Reviewer_Y2Qd · 2025-11-22
> **Thank you for your response**
>
> I am glad to see the clarification of the metrics in the text, and for more detailed quality-diversity plots. My main concert is still the point that was not addressed in author response, namely unclear claims and conclusions. Currently, the formulation of the claims is very vague and I would not be convised to use your method by readining the text in its current form. See the comments below.
>
> The abstract contains: "our approach consistently performs at least on par with the existing methods it builds upon in terms of quality, while potentially improving output diversity", what potentially improving means? That you don't know if your method improves results?
>
> For the extreme summarization, your method seem to perform not better than min-p in terms of accuracy, and it is hard for me to understand which method is better on quality-diversity frontier.
>
> For the Divergent Association Task, it reads as if greedy is the best method (I assume it will be best only in terms on quality, but what about quality-diversity).
>
> For the WritingPrompts, from the table presentation I cannot see which method is better in terms of quality-diversity, but it looks like η-sampling is stronger on both metrics for the RLHF-instructed model. Presentation of results is quite inconsistent as it looks and it is hard for me to understand which method is better.
>
> I would ask you to clearly present the results on the quality-diversity plot with optimal hyperparameters to see if your method is better (higher quality at the same lever of diversity), and I leave this to be decided by the authours. I hope you would consider revising the claims and conclusions.

---

> > ### Author Response · Authors · 2025-11-24
> >
> > We thank the reviewer for the additional comment. In focusing our revision efforts on the list of requested changes, we overlooked the issue regarding unclear claims and conclusions.
> >
> > We understand the reviewer’s concerns regarding the lack of clarity in that sentence in both the Abstract and the Conclusion section. By “potentially improving”, we simply meant that because our methods (in particular, the two relaxations) preserve a small number of additional low-probability tokens, the resulting gain in diversity may not be noticeable in all situations. However, we believe that the arguments presented in our paper provide adequate support for adopting our method. First of all, *DiffSampling* is theoretically motivated and now supported by a formal analysis establishing when the truncation strategy preserves only tokens with positive probability under the true distribution. In contrast, most alternatives rely solely on empirical evidence. Secondly, while always preserving the same or more tokens than comparable baselines (i.e., greedy, top-$p$, and min-$p$ methods), we show through a vast set of experiments that the quality of outputs is not undermined, being on par or not significantly lower. As we pointed out in Section 4, the differences between standard methods and ours are often minimal and typically involve low-probability tokens only, thus leading to very small variations in terms of quality and diversity performance. However, we believe that having a principled method that guarantees sampling from a larger set of tokens while having no impact on the quality of outputs is a significant contribution not only for researchers but also for practitioners.
> >
> > We agree that this should have been made clearer in the text: we have now modified the abstract by replacing “while potentially improving output diversity“ with “despite sampling from a larger set of tokens”. Similarly, we have replaced the same sentence in the Conclusion section with “despite sampling from a larger set of tokens, which has a positive impact on diversity”. We hope that our claims and findings are now presented more clearly.
> >
> > As far as the quality-diversity plots are concerned, we do not believe that the definitions of “best performance” and “optimal hyperparameters” are straightforward. In fact, it is not possible to define what is better since we have quality-diversity trade-offs. A configuration that leads to higher accuracy also leads to decreased diversity, and it is very rare to obtain higher quality at the same level of diversity. In our opinion, the current design of the plot allows for a clear visualization of the different quality-diversity trade-offs to which the various methods are subject.
> >
> > Finally, it seems that the results for the divergence association task were not clearly explained. In particular, the DAT score is not a quality metric but a diversity metric. As reported in Section 5.4 and Appendix C, it corresponds to the average cosine distance between the generated nouns. Consequently, all figures related to DAT already incorporate both quality and diversity, enabling a proper discussion of quality-diversity trade-offs. For example, while the greedy strategy produces a strong set of nouns for the RLHF model, *DiffSampling-cut* generates a better set in nearly half of the 100 generations and consistently produces a valid set of nouns. In our experiments, the other sampling strategies (excluding min-$p$ and *DiffSampling-minp*) have produced at least one better set of nouns, but they generally generate lower-scoring sets and occasionally fail to produce a valid set. For the pre-trained model, the quality-diversity trade-off is more pronounced: all methods generate sets with, on average, higher DAT scores than the greedy strategy, but they frequently produce invalid sets. In contrast, *DiffSampling-cut* achieves a slightly lower average score while consistently producing valid outputs. We have now clarified this in the main text.

---

> ### Comment · Reviewer_Y2Qd · 2025-11-24
> **Thank you for the clarifications**
>
> I thank the reviewers for adjusting the formulations in the paper, and for clarifications regarding the metrics.
>
> >As far as the quality-diversity plots are concerned, we do not believe that the definitions of “best performance” and “optimal hyperparameters” are straightforward. In fact, it is not possible to define what is better since we have quality-diversity trade-offs. A configuration that leads to higher accuracy also leads to decreased diversity, and it is very rare to obtain higher quality at the same level of diversity. In our opinion, the current design of the plot allows for a clear visualization of the different quality-diversity trade-offs to which the various methods are subject.
>
> I think that we touch opon an important methodological point here, and I hope we can find a common ground. I fully agree that usually by changing a method's parameter (e.g. temperature, p in top-p, min-p), we move quality up and diversity down, and vice-versa. And I fully agree, that it is not enough to directly compare *single points* on the quality diversity curves.
>
> What I expected to see, and what in my opinion is a methodologically valid way, is to compare the distance between (approximatly computed) diversity-quality *curves* and not single points. Namely, I consider two methods to be equivalent up to the change in parameterization (temperature, p, etc), if two diversity-quality curves closely match each other (where a curve is a set of diversity-qulity results of a single method under different parameters e.g. temperature or p). How to interpret results? If one curve is above and another is below, it means that one method has better diversity-quality trade-off (up to a certaint confidence level). What you probably see by saying "and it is very rare to obtain higher quality at the same level of diversity" is that indeed *many approaches like min-p, top-p etc are actually equivalent* (there is no clear winner!) if you compare diversity-quality trade-off curves (independent of concrete parameterzation values). For that, I was referring to "Min-p, Max Exaggeration: A Critical Analysis of Min-p Sampling in Language Models" by Schaeffer et al., or "Control the Temperature: Selective Sampling for Diverse and High-Quality LLM Outputs" by Troshin et al. This is precisely why I wanted to see the clear comparison of diversity-quality curves, because my current hypothesis is still that the methods you consider are in fact *equivalent* and you just compare instances from different parts of the same diversity-quality curve (either presenting method with higher quality and lower diversity or vise-versa).

---

> > ### Author Response · Authors · 2025-11-25
> >
> > We thank the reviewer for these clarifications, which have helped us better understand the concerns raised.
> >
> > We agree that, in theory, comparing the distance between quality-diversity curves represents a more valid way to analyse different methods. However, in practice, this assumes the quality-diversity relationship to be monotonic. What we observed is that it is not possible to compute approximated curves for the tested methods, as they vary irregularly between different points. We have now replaced the plots in the paper by including the lines connecting the points at increased temperatures to make this more apparent. As a result, what we obtain are curves that cannot be easily compared: they often intersect or cover different regions of the space. This also prevents considerations regarding performance at temperatures not included in the experiments, thus making it impossible to state whether a curve is generally better than another. In addition, we think it is important to stress that, as mentioned, this analysis would be correct “up to the change in parameterization”: overall, what we can say by looking at quality-diversity curves is merely that a method behaves better than another (or on par with another) for that specific parameterization, at the specific tested temperatures, for the specific case study with specific experimental settings (seeds, number of repetitions, batch sizes, ecc.). Because of this, we are not convinced that the findings for a given setting can be generalised to support the claim that the different methods are equivalent. We therefore believe that the reviewer’s conjecture (namely, that our method is equivalent and that we are merely comparing instances from different parts of the same curve) cannot be substantiated in practice since this would require covering all the possible combinations of scenarios/case studies, experimental settings, parameterisations and temperature values.
> >
> > Moreover, we would like to underline the fact that, even if the conjecture were accurate and all sampling schemes were effectively interchangeable, our method would remain substantively important. Its foundations are theoretically robust and not merely empirical; it offers concrete guarantees concerning sampling from the true support; and it invariably operates over a distribution whose support is at least as large as (and often broader than) that of comparable methods.

---

### Review · Reviewer_zQ1r · 2025-10-30

**Summary Of Contributions:**

The paper proposes DiffSampling, a new decoding strategy for large language models that aims to balance diversity and accuracy. Instead of truncating token probabilities by absolute thresholds as in top-p or min-p sampling, DiffSampling determines the cutoff by analyzing the discrete derivative of the token probability distribution. The point of minimum derivative is used as the truncation boundary, and two relaxed versions, DiffSampling-lb and DiffSampling-minp, are introduced to further control diversity.

The method is motivated by the intuition that abrupt probability drops correspond to transitions from likely to unlikely tokens. The authors demonstrate through experiments on GSM8K, MATH, XSum, divergent association, and WritingPrompts that DiffSampling produces more diverse outputs while maintaining accuracy.

**Strengths:**

1. The algorithm is simple, fast, and compatible with existing decoding pipelines.
2. Experiments cover several types of tasks, including reasoning, summarization, and creative writing.
3. The evaluation includes both accuracy and diversity metrics, showing consistent improvement over baselines.

**Weaknesses:**

1. The proposed solutions, particularly in Section 3, are highly intuitive and lack deeper theoretical grounding. The transition from the derivative-based observation to the final algorithm is more heuristic than principled.
2. The novelty is moderate given the extensive literature on decoding heuristics; the proposed approach can be viewed as a straightforward modification of existing truncation methods.
3. Evaluation is limited to relatively small models without human judgment, so the generality and perceptual value of the claimed diversity improvements remain unclear.

**Audience:**

Yes

**Audience Explanation:**

The study addresses an important practical aspect of LLM generation—decoding control—which is of interest to TMLR readers working on natural language generation, efficiency, and interpretability.

**Broader Impact Concerns:**

The proposed method does not introduce new ethical risks.

**Claims And Evidence:**

Yes

**Claims Explanation:**

The experiments are comprehensive and systematically presented. Comparisons against established baselines are fair, and results on both quality and diversity support the authors’ claims. The implementation details are transparent and reproducible. However, the evidence primarily confirms empirical consistency, not theoretical validity. The central intuition in Section 3 is reasonable but lacks formal justification or analysis of when and why the derivative-based cutoff should be optimal.

**Requested Changes:**

1. Strengthen the theoretical explanation of why the minimum derivative consistently marks a meaningful boundary in token distributions (critical).

3. Expand experiments to larger model scales to confirm whether the approach generalizes beyond small LLMs (critical).

1. Add a human evaluation or qualitative comparison to verify whether the increased diversity leads to perceptually better results (recommended).

1. Discuss connections and differences with prior gradient- or derivative-based truncation methods, if any, to better position the contribution (minor).

1. Clarify in Section 3 how the derivative-based intuition transitions into the implemented algorithm, possibly with a visualization or toy example (minor).

---

> ### Author Response · Authors · 2025-11-10
>
> We thank the reviewer for their extensive feedback and insightful suggestions. Please find our responses to the requested changes below.
>
> > Strengthen the theoretical explanation of why the minimum derivative consistently marks a meaningful boundary in token distributions (critical).
>
> We would like to thank the reviewer for the suggestion: we have now added a detailed formal analysis of our truncation strategy in Appendix A. In particular, we verified under which conditions our truncation is safe (i.e., no tokens with a zero probability under the true distribution are preserved). We found two conservative bounds that, regardless of the exact value assumed by the smoothing and weighting coefficients, mark when our method is safe: anytime the true distribution has a minimum derivative greater than or equal to $\frac{1}{|V|}$; or anytime the cardinality of tokens with a positive probability under the true distribution is less than or equal to $\sqrt{2|V|}$. While this means that our method is not always safe, the two conservative bounds are typically satisfied in practice. Moreover, the bounds were derived under worst-case assumptions for all variables involved. However, in the characterization provided by Hewitt et al. (2022), certain parameter combinations are mutually exclusive, which makes the actual bound less restrictive and enables our method to safely account for additional scenarios.
> > Expand experiments to larger model scales to confirm whether the approach generalizes beyond small LLMs (critical).
>
> We tested our three methods and the five baselines with a fixed temperature equal to 1 for Llama3-70B (both pre-trained and instructed), and we added these preliminary results in a dedicated section of the Appendix (Appendix D). We found that the same conclusions we drew in the main paper can be drawn for larger models as well: *DiffSampling-cut* performs comparably to greedy strategy in terms of accuracy, while allowing for greater diversity; and the two relaxations perform on par (or slightly worse) than their corresponding baselines in terms of quality, while slightly increasing their diversity. The only notable difference is that, for the instructed model, greediness is sometimes less desirable, causing *DiffSampling-cut* and *DiffSampling-minp* (as well as the greedy and min-$p$ strategies) to obtain lower scores than the other, more relaxed sampling strategies. However, this does not invalidate our main findings; it simply highlights the need for distinct sampling strategies across model sizes. While *DiffSampling-cut* and *DiffSampling-minp* perform better for smaller models, *DiffSampling-lb* emerges as the preferable choice for larger ones.
> > Add a human evaluation or qualitative comparison to verify whether the increased diversity leads to perceptually better results (recommended).
>
> While we acknowledge that a human evaluation would add value to our findings, conducting it at the necessary scale and level of detail is not feasible within the time available for preparing this rebuttal, given the broad scope of our experiments. In fact, we chose a quantitative evaluation in the first place, given the characteristics of the performance evaluation of the algorithms and the number of dimensions at hand. For example, the differences between outputs are often minimal and typically limited to token-level decisions, an observation supported by the negligible variation in “semantic” diversity, in contrast to the larger differences observed in syntactic-based metrics. A qualitative evaluation, while certainly valuable, would address a different set of questions (orthogonal to our present focus) and could be more appropriately pursued in future work, i.e., it could form the basis of a future paper focused on human-centered evaluation.
> > Discuss connections and differences with prior gradient- or derivative-based truncation methods, if any, to better position the contribution (minor).
>
> To the best of our knowledge, there are no prior gradient- or derivative-based truncation methods to be compared with. Current sampling methods rely on fixed thresholds or properties of the distribution (e.g., entropy or standard deviation) but still require pre-determined hyperparameters. In contrast, our base method, *DiffSampling-cut*, operates without the need for any hyperparameters.
> > Clarify in Section 3 how the derivative-based intuition transitions into the implemented algorithm, possibly with a visualization or toy example (minor).
>
> We already have a toy, descriptive example in Section 4, which we believe can help the reader understand how our methods work in practice. We understand that its connection with the theoretical description in Section 3 can be strengthened, and we plan to do so in the future version of the paper by adding that: “The next section provides a toy example to showcase this relation” when discussing *DiffSampling-cut* in relation to greedy decoding.

---

### Decision · Action_Editor_qgi2 · 2025-12-13

**Recommendation:** Accept as is

**Audience:**

Yes

**Audience Explanation:**

The paper proposes DiffSampling, a decoding approach for large language models (LLMs) that selects a subset of candidate next tokens by analyzing the discrete derivative of the sorted token probability distribution. Rather than applying fixed probability or cumulative-mass thresholds, DiffSampling identifies a truncation point that corresponds to the minimum derivative. This approach is motivated by the intuition that sharp drops in probability indicate a transition from plausible to unlikely tokens. The paper describes two relaxed variants of the DiffSampling framework: DiffSampling-lb, which enforces a lower bound on retained cumulative probability mass, and DiffSampling-minp, which constrains retained probabilities relative to the maximum token probability. After truncation, the method optionally applies temperature-based sampling to modulate output diversity. The paper evaluates the approach on various tasks including mathematical reasoning (GSM8K, MATH), summarization (XSum), divergent association, and story generation (WritingPrompts), comparing different truncation and sampling configurations against existing decoding strategies.

The paper was evaluated by three reviewers who agree on many of its key strengths and weaknesses. As all three reviewers emphasize, the paper addresses an important and practical aspect of LLM generation, namely decoding control. As such, the work will be of interest to many in the TMLR community. Beyond its relevance, the reviewers appreciated DiffSampling's simplicity and efficiency, and the ease with which it can be incorporated into various existing frameworks. Additionally, each of the reviewers emphasized the diversity of tasks on which the paper analyzes DiffSampling.

At the same time, the reviewers shared similar concerns with the paper as initially submitted. Among them, they noted that the paper as originally submitted lacked concrete claims about the settings and domains for which one could expect DiffSampling to perform better than existing methods. Related to the comment above, the results reveal that the proposed method is not universally better than the baselines. While this is commendable, the paper would benefit from a discussion of the conditions for which one could expect DiffSampling to be superior. Additionally, at least two of the reviewers emphasized that certain aspects of the framework appear to be rather obvious and lack theoretical justification. Additionally, two reviewers commented that the paper lacks human judgment-based evaluations on larger models, which obfuscates DiffSampling's merits.

The authors made a concerted effort to address the key concerns raised by the reviewers, as exemplified by their updates to the paper and their responses to the reviewers. The reviewers agree that these updates/responses help to address several of the key issues with the paper that they initially raised. Notably, these include questions of the theoretical justification for the proposed method, which is now much clearer. The reviewers and AE appreciate the measured nature of the paper's claims and find that the paper's insights into different decoding techniques may be useful for practitioners.

**Claims And Evidence:**

Yes

**Claims Explanation:**

The modesty of the paper's claims is welcome. The proposed DiffSampling variants are generally comparable to strong baselines, while the method improves diversity in several settings. As was clarified during the author-reviewer discussion, the decoding rule is not universally superior. However, the decoding rule is advantageous in that it is more theoretically grounded than contemporary heuristics and can be readily used with various existing models while preserving performance for a diverse set of domains.